# Accurate Evaluation of Quickest Changepoint Detectors via Non-parametric Survival Analysis

## Abstract

We propose non-parametric estimators for the average run length (ARL) and average detection delay (ADD) in quickest changepoint detection (QCD) under finite and irregular sequence lengths. Although ARL and ADD are widely used as optimality criteria in theoretical and simulation studies, their application to real-world datasets is hindered by limited and irregular sequence lengths. To address this issue, we propose non-parametric estimators for the ARL and ADD, termed *KM-ARL and KM-ADD*, by drawing an analogy between QCD and survival analysis to model detection probabilities under sequence truncation. We derive estimation bias bounds and prove that they are asymptotically unbiased unless extrapolation is required. Experiments on simulated and real-world datasets demonstrate their practical utility, enhancing robustness against limited and irregular sequence lengths, improving interpretability, and facilitating empirical, intuitive model selection. Our Python code are provided in the supplementary material and will be released upon acceptance, offering ready-to-use implementations for practitioners.

## 1 Introduction

We study evaluation metrics for models in online quickest changepoint detection (QCD) with unknown pre- and post-change distributions, where their datasets with changepoint labels are available. QCD has been extensively studied theoretically, with many models proposed and their optimality proven (Tartakovsky, 2019). It also has diverse real-world applications, including statistical process control (Hawkins et al., 2003), industrial quality control (Wadinger et al., 2024), epidemiology (Johnson & Pedersen, 2025), wireless sensor networks (Hadjiliadis et al., 2009), health monitoring (Tan et al., 2023), radar target detection (Xiang et al., 2021), and seismic sensing (Li et al., 2016).

The average run length (ARL) and the average detection delay (ADD) are central to the theoretical and simulation analysis of QCD models (Tartakovsky et al., 2014; Tartakovsky, 2019). The ARL is defined as the average time a detector takes to raise a false alarm, while the ADD refers to the average delay a detector takes to identify a changepoint after it has occurred. These metrics exhibit a tradeoff: reducing ADD typically shortens ARL (making the detector trigger-happy) and vice versa.

Although the ARL and ADD are widely used as optimality criteria in theoretical and simulation studies, their application to real-world datasets is challenging because practical sequences are often limited and irregular in length. Fig. 1(a) shows that naive ARL and ADD estimators yield substantial bias and variance, hindering reliable evaluation of QCD models.

To address this issue, we draw an analogy between QCD and survival analysis (Kleinbaum & Klein, 1996) and propose non-parametric estimators for the ARL and ADD. We adapt the Kaplan-Meier estimator (KME) (Kaplan & Meier, 1958) to the QCD setting to model detection probabilities under sequence truncation. These estimators are non-parametric, requiring no assumptions on the underlying data distribution (e.g., exponential). We derive estimation bias bounds for these estimators, termed *KM-ARL* and *KM-ADD*, by decomposing the estimation bias into *finite-sample bias* and *truncation bias*. We then show that the finite-sample bias decays exponentially with increasing dataset size and that the truncation bias is smaller than that of conventional estimators. Building on these findings, we prove that the KM-ARL and KM-ADD are *asymptotically unbiased* unless extrapolation is required.

To demonstrate practical applicability, we conduct experiments on both simulated and real-world datasets with limited and irregular lengths. The results show that our estimators reduce estimation bias compared to baseline estimators and enhance robustness to limited and irregular sequence lengths, thereby improving interpretability and facilitating empirical, intuitive model selection (Fig. 1(b)). Our code is provided in the supplementary material and will be publicly released upon acceptance, offering off-the-shelf, ready-to-use implementations in Python (Van Rossum & Drake, 2009) for practitioners.

Our contributions are threefold: (1) we propose KM-ARL and KM-ADD, non-parametric estimators for the ARL and ADD, enabling their evaluation on real-world datasets with limited and irregular-length sequences; (2) we derive estimation bias bounds for these estimators and prove that they are asymptotically unbiased unless extrapolation is required; and (3) we demonstrate their practical utility through experiments and provide ready-to-use Python implementations.

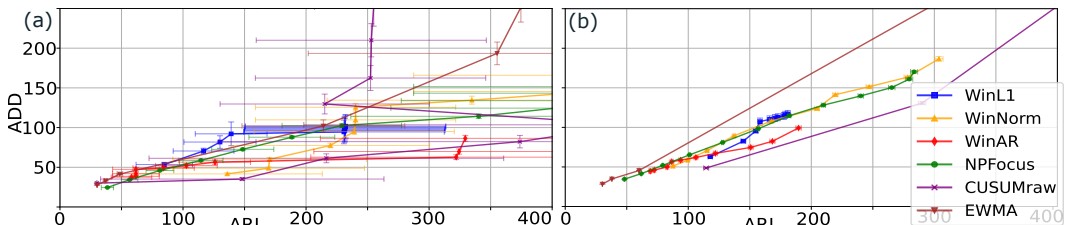

Figure 1: **Evaluation of QCD models on real-world dataset (machine-labeled subset of WISDM Actitracker). (a) LB-ARL & LB-ADD. (b) KM-ARL & KM-ADD.** Error bars represent the standard error of the mean. Our KM-ARL and KM-ADD are more robust to irregular lengths than the conventional estimators (LB-ARL and LB-ADD: see Sec. 3). The figures are shown at the same magnification scale for readability. The complete figure is provided in Fig. 14 in App. C.5. Additional evaluation on a Gaussian process dataset is provided in App. C.3.

## 2 RELATED WORK

We highlight our contribution in the context of prior research on ARL and ADD estimation using survival analysis. To our knowledge, there is no metric that explicitly address irregular sequence lengths in QCD. (Sahki et al., 2020) empirically estimate ARLs and ADDs on truncated sequences of *fixed* length, leveraging *parametric* survival analysis, assuming the survival function decays exponentially. (Bradley et al., 2023) employ a *semi-parametric* survival analysis method, the Cox proportional hazards model (Cox, 1972), which assumes an exponential response of the estimator to covariates. They also use the KME to empirically evaluate intrusion detectors under *fixed* length truncation; however, the ARL and ADD are not directly estimated, and no theoretical analysis is provided. (Lim & Lee, 2025) propose ad hoc estimators for the ARL and ADD on sequences of *fixed* length; however, their theoretical justification has yet to be established, as these estimators diverge when the sequence length $\to \infty$. In contrast, we build our estimators on *non-parametric* survival analysis, eliminating the exponential assumptions, derive bias bounds, and prove asymptotic unbiasedness for the proposed estimators. Moreover, our analysis includes *irregular* sequence lengths, thereby addressing more practical scenarios. We also provide a ready-to-use implementations of our estimators for practitioners. Supplementary related work is given in App. E.

## 3 KAPLAN-MEIER ARL & ADD

We first introduce our notation. For comparison, we then introduce conventional estimators under random length truncation. Finally, we present our proposed estimators.

**Definitions.** $[n]$ denotes $\{1, 2, \ldots, n\}$ with $n \in \mathbb{N}$. We use $P$ as a probability distribution. $X^{(0,t)} = (X^{(0)}, X^{(1)}, \ldots, X^{(t)})$ denote real-valued random variables representing a sequence of length $t + 1$ ($t \in \mathbb{Z}_{\geq 0}$). Each frame $X^{(s)}$ ($s \in \mathbb{Z}_{\geq 0}$) is sampled from the pre-change density $g(X^{(s)} \mid X^{(0,s-1)})$ when $s < \nu$ and from the post-change density $f(X^{(s)} \mid X^{(0,s-1)})$ when $s \geq \nu$, where the changepoint

$\nu \in \mathbb{Z}_{\geq 0} \cup \{\infty\}$ is a random variable independent of the observations. $\nu = \infty$ and $\nu = 0$ indicate that all frames in the sequence are sampled from the pre-change and post-change densities, respectively. Let $T \in \mathbb{Z}_{\geq 0}$ denote a random variable representing the sequence length, which is independent of the observations. Let $\tau : X^{(0,T)} \mapsto \tau(X^{(0,T)}) \in \mathbb{Z}_{\geq 0} \cup \{\infty\}$ be a changepoint detector, where $\tau(X^{(0,T)})$ is the detection point of sequence $X^{(0,T)}$. We set $\tau(X^{(0,T)}) = \infty$ when no change is detected within the input sequence of finite length. The ARL and ADD are defined as

$$\mu_\infty := \mathbb{E}[\tau \mid \nu = \infty]$$
$$\text{and } M_\infty := \mathbb{E}[\Delta\tau \mid \Delta\tau \geq 0, \nu < \infty], \tag{1}$$

respectively, where $\Delta\tau := \tau - \nu$. $T = \infty$ is assumed here, and the expectation is also taken over $\nu$. We consider estimation of the ARL and ADD from a dataset $\{(X_i^{(0,T_i)}, \nu_i)\}_{i=1}^N$ with predicted detection points $\{\tau_i\}_{i=1}^N$ from a detector $\tau$, where $N \in \mathbb{N}$ denotes the dataset size, $X_i^{(0,T_i)}$ is a sequence with a random length $T_i$, $\nu_i$ is the corresponding changepoint, and $\tau_i$ is the detection point of $X_i^{(0,T_i)}$. Let $\langle \cdot \rangle_{i:\mathcal{C}} := \sum_{i:\mathcal{C}} \cdot /|\{i : \mathcal{C}\}|$ denote the empirical expectation under condition $\mathcal{C}$.

**Conventional estimators.** A conventional estimator of ARL under random sequence lengths, adapted from (Qiu, 2013) and referred to as the Less-Biased ARL (LB-ARL) in this paper, follows the definition of $\mu_\infty$: $\hat{\mu}_T^{(\text{LB})} := \langle \tau_i \rangle_{i:\nu_i=\infty,,\tau_i \leq T_i}$, where only sequences satisfying $\tau_i \leq T_i$ are included. Similarly, a conventional ADD estimator, the LB-ADD, can be defined as $\hat{M}_T^{(\text{LB})} := \langle \Delta\tau_i \rangle_{i:\Delta\tau_i \geq 0, \nu_i < \infty, \Delta\tau_i \leq \Delta T_i}$, where $\Delta T_i := T_i - \nu_i$.

A critical drawback of these conventional estimators is that they ignore sequences in which the QCD model fails to raise an alarm before the horizon ($\tau > T$). Although the LB-ARL has been employed in Monte Carlo simulations of the ARL (Qiu, 2013; Lim & Lee, 2025) when $T = \text{const.} < \infty$, it exhibits a more substantial negative bias than our estimator due to truncation, as proven in Sec.4 and demonstrated in Sec.5.

**Key ideas from survival analysis.** To overcome this challenge, we model the detection probability beyond the truncation length, using a non-parametric approach inspired by *survival analysis* (Kleinbaum & Klein, 1996). A central interest in survival analysis is the *survival function* $S(t) := P(\text{event time} > t)$, representing the probability that a patient survives beyond time $t$ ($t \in \mathbb{R}_{\geq 0}$), where the *event time* refers to the time of death. Crucially, the estimation of $S(t)$ on a dataset is performed under *right-censoring*; i.e., the exact event times of several patients are unknown because the patients are lost to follow-up or are still under observation at the end of the study. Thus, we only know the lower bound of the event times of these patients, called *censoring times*. The mean survival time is given by the integral of $S(t)$ over time, i.e., the area under the survival curve, because $\int_0^\infty S(t)dt = \int_0^\infty P(\text{event time} > t)dt = \int_0^\infty \mathbb{E}[\mathbb{1}(\text{event time} > t)]dt = \mathbb{E}[\int_0^\infty \mathbb{1}(\text{event time} > t)dt] = \mathbb{E}[\text{event time}]$, where $\mathbb{1}$ is the indicator function.

**KM-ARL.** To estimate the ARL under irregular sequence lengths, we draw an analogy by regarding a patient as a sequence, the event time as the detection point, the censoring time as the minimum of the changepoint and the sequence length, and the mean survival time as the ARL (see Tab. 1 in App. A for reference). Under this analogy, we estimate the *survival function of detection points* $S^{\text{ARL}}(t) := P(\tau > t \mid \nu = \infty)$ using the Kaplan-Meier estimator (KME) (Kaplan & Meier, 1958), a non-parametric estimator of the survival function: $\hat{S}^{\text{ARL}}(t) = \prod_{j:t_j^{\text{ARL}} \leq t} (1 - \frac{d_j^{\text{ARL}}}{n_j^{\text{ARL}}})$, where $0 < t_1^{\text{ARL}} < t_2^{\text{ARL}} < \cdots < t_{N'}^{\text{ARL}}$ are distinct detection points, with $N'_{\text{ARL}} \in \mathbb{N} (\leq N)$; $d_j^{\text{ARL}} := |\{i \in [N] \mid \tau_i = t_j^{\text{ARL}}\}|$ is the number of sequences with a detection at $t_j^{\text{ARL}}$ ($j \in [N'_{\text{ARL}}]$); and $n_j^{\text{ARL}} := |\{i \in [N] \mid \min\{\tau_i, C_i^{\text{ARL}}\} \geq t_j^{\text{ARL}}\}|$ is the number of sequences neither detected nor censored prior to $t_j^{\text{ARL}}$ ($j \in [N'_{\text{ARL}}]$), with the censoring time of sequence $i$ defined as $C_i^{\text{ARL}} := \min\{\nu_i, T_i\}$. An example of $\hat{S}^{\text{ARL}}(t)$ is shown in App. C.1. We propose a non-parametric estimator of the ARL under irregular sequence lengths, termed KM-ARL, as the integral of $\hat{S}^{\text{ARL}}(t)$ over the range $[0, a]$ for arbitrary $a \in \mathbb{R}_{\geq 0}$:

$$\hat{\mu}_T^{(\text{KM})} := \int_0^a \hat{S}^{\text{ARL}}(t)dt. \tag{2}$$

In practice, we set $a = T_{\max} := \max_i\{\min\{\tau_i, C_i^{\text{ARL}}\}\}$, i.e., the maximum last-observed time, following standard practice in survival analysis (Qi & Wang, 2018; Calkins et al., 2018). For a given dataset, choosing $a > T_{\max}$ is irrelevant because it is extrapolation beyond the observed support. Theoretically ideal choices of $a$ are given in Sec. 4.

**KM-ADD.** For the ADD, we regard a patient as a sequence with $\nu_i < \infty$ and $\tau_i \geq \nu_i$, the event time as the detection delay $\Delta\tau_i(\geq 0)$, the censoring time as the sequence length measured from the changepoint ($C_i^{\text{ADD}} := \Delta T_i = T_i - \nu_i$), and the mean survival time as the ADD (see Tab. 1 in App. A for reference). Under this analogy, we propose a non-parametric estimator of the ADD under irregular lengths, termed KM-ADD, as

$$\hat{M}_T^{(\text{KM})} := \int_0^b \hat{S}^{\text{ADD}}(t)dt, \tag{3}$$

where $\hat{S}^{\text{ADD}}(t) := \prod_{j:t_j^{\text{ADD}} \leq t}(1 - \frac{d_j^{\text{ADD}}}{n_j^{\text{ADD}}})$ is a non-parametric estimate of the *survival function of detection delays* $S^{\text{ADD}}(t) := P(\Delta\tau > t \mid \Delta\tau \geq 0, \nu < \infty)$; $0 \leq t_1^{\text{ADD}} < t_2^{\text{ADD}} < \cdots < t_{N'_{\text{ADD}}}^{\text{ADD}}$ are the distinct detection delays, i.e., the sorted unique values of $\Delta\tau_i \geq 0$, with $N'_{\text{ADD}} \in \mathbb{N}$ ($\leq N$); $d_j^{\text{ADD}} := |\{i \in [N] \mid 0 \leq \Delta\tau_i = t_j^{\text{ADD}}\}|$ is the number of sequences with positive detection delay equal to $t_j^{\text{ADD}}$ ($j \in [N'_{\text{ADD}}]$); and $n_j^{\text{ADD}} := |\{i \in [N] \mid \min\{\Delta\tau_i, C_i^{\text{ADD}}\} \geq t_j^{\text{ADD}}, \Delta\tau_i \geq 0\}|$ is the number of sequences neither detected nor censored prior to $t_j^{\text{ADD}}$ ($j \in [N'_{\text{ADD}}]$). Again, the upper limit $b$ is arbitrary, and we set $b = \Delta T_{\max} := \max_i\{\min\{\Delta\tau_i, C_i^{\text{ADD}}\} \mid \Delta\tau_i \geq 0, \nu_i < \infty\}$ in experiment.

## 4 BIAS ANALYSIS

We derive bias bounds for the KM-ARL and KM-ADD and prove that they are asymptotically unbiased unless extrapolation is required. We first examine the estimation bias of the KM-ARL:

$$\mathcal{B}(\hat{\mu}_T^{(\text{KM})}) := \mathbb{E}[\hat{\mu}_T^{(\text{KM})}] - \mu_\infty, \tag{4}$$

where $\mu_\infty := \mathbb{E}[\tau \mid \nu = \infty] < \infty$ is the true ARL under infinite sequence length (we assume all relevant finiteness hereafter). We decompose this bias into two components: the *finite-sample bias* and the *truncation bias*

$$\mathcal{B}_{\text{FS}}(\hat{\mu}_T^{(\text{KM})}) := \mathbb{E}[\hat{\mu}_T^{(\text{KM})}] - \mu_T^{(\text{KM})} \tag{5}$$

$$\mathcal{B}_{\text{TR}}(\hat{\mu}_T^{(\text{KM})}) := \mu_T^{(\text{KM})} - \mu_\infty^{(\text{KM})} \tag{6}$$

so that $\mathcal{B}(\hat{\mu}_T^{(\text{KM})}) = \mathcal{B}_{\text{FS}}(\hat{\mu}_T^{(\text{KM})}) + \mathcal{B}_{\text{TR}}(\hat{\mu}_T^{(\text{KM})})$, where $\mu_T^{(\text{KM})} := \int_0^a S_{\text{ARL}}(t)dt$ is the KM-ARL with the true survival function of detection points.

**Finite-sample bias.** $\mathcal{B}_{\text{FS}}(\hat{\mu}_T^{(\text{KM})})$ quantifies the error arising from a finite dataset of sequences with finite and irregular lengths and dominates the total bias when the dataset size is small. We derive a bound for $\mathcal{B}_{\text{FS}}(\hat{\mu}_T^{(\text{KM})})$, indicating that it decays exponentially as $N \to \infty$:

**Theorem 4.1** (Finite-sample bias bounds for KM-ARL). *We idealize the time index as continuous without loss of generality, for technical convenience. Let $F^{\text{ARL}}, G^{\text{ARL}}$, and $H^{\text{ARL}}$ be the cumulative distribution functions (CDFs) of $\tau$, $C^{\text{ARL}}$, and $\min\{\tau, C^{\text{ARL}}\}$, respectively. Assume that (i) $F^{\text{ARL}}$ and $G^{\text{ARL}}$ do not have common discontinuities, (ii) $\tau$ and $C^{\text{ARL}}$ are independent, known as the independent censoring (or non-informative censoring) assumption (Ranganathan & Pramesh, 2012), and (iii) $H^{\text{ARL}}$ is continuous. Then, we have for any $a \in \mathbb{R}_{\geq 0}$*

$$-\int_0^a t\, G^{\text{ARL}}(t)\, H^{\text{ARL}}(t)^{N-1}\, dF^{\text{ARL}}(t)$$

$$\leq \mathcal{B}_{\text{FS}}(\hat{\mu}_T^{(\text{KM})}) \leq \int_0^a a\, G^{\text{ARL}}(t)\, H^{\text{ARL}}(t)^{N-1}\, dF^{\text{ARL}}(t). \tag{7}$$

Our proof is based on (Stute & Wang, 1993; Stute, 1994) and given in App. B.1. Assumption (i) is a technical requirement and is valid unless pathological situations are considered. Assumption

(ii) holds for online QCD models that do not look ahead the input sequence, which are the focus in this paper. This is because: detection points $\tau_i$ prior to censoring can be regarded as samples from $P(\tau \mid \nu = \infty)$; $P(\tau \mid \nu = \infty)$, $P(\nu)$, and $P(T)$ are independent; and thus, $P(\tau \mid \nu = \infty)$ and $P(C := \min\{\nu, T\})$ are also independent. In contrast, Assumption (ii) does not hold for offline changepoint detection models because $\tau$ depends on $X^{(0,T)}$, which in turn depends on $\nu$ and $T$. Assumption (iii) is also a technical requirement.

According to Thm. 4.1, the finite-sample bias exhibits the following properties. First, it decays exponentially to zero as $N \to \infty$ if $H^{\mathrm{ARL}}(t) < 1$ for $t \in [0, a]$. Second, it vanishes when $G^{\mathrm{ARL}} = 0$, i.e., in the absence of censoring (no changepoints or horizons) in $t \in [0, a]$, which is desirable because estimation becomes more accurate as censoring decreases. Third, the bound becomes looser for larger $a$ because $G^{\mathrm{ARL}}$ and $H^{\mathrm{ARL}}$ approach 1 monotonically as $t$ increases. However, the truncation bias decreases for larger $a$, leading to a tradeoff between the finite-sample and truncation biases. Finally, we note that empirically verifying the convergence of the finite-sample bias is challenging because, for small sample sizes, estimation variance overshadows the bias.

We can derive a similar bound for the finite-sample bias of the KM-ADD (proof is given in App. B.2), ensuring that it also decays exponentially as $N'' \to \infty$, where $N''$ is the number of sequences with $\nu_i < \infty$ and $\Delta\tau_i \geq 0$:

**Theorem 4.2** (Finite-sample bias bounds for KM-ADD). *We idealize the time index as continuous without loss of generality, for technical convenience. Consider only sequences with $\nu_i < \infty$ and $\Delta\tau_i \geq 0$ in the dataset. Let $F^{\mathrm{ADD}}, G^{\mathrm{ADD}}$, and $H^{\mathrm{ADD}}$ be the CDFs of $\Delta\tau, C^{\mathrm{ADD}}$, and $\min\{\Delta\tau, C^{\mathrm{ADD}}\}$, respectively. Assume that (i) $F^{\mathrm{ADD}}$ and $G^{\mathrm{ADD}}$ do not have common discontinuities, (ii) $\Delta\tau$ and $C^{\mathrm{ADD}}$ are independent, and (iii) $H^{\mathrm{ADD}}$ is continuous. Then, for any $b \in \mathbb{R}_{\geq 0}$,*

$$- \int_0^b t\, G^{\mathrm{ADD}}(t)\, H^{\mathrm{ADD}}(t)^{N''-1}\, dF^{\mathrm{ADD}}(t)$$

$$\leq \mathcal{B}_{\mathrm{FS}}(\hat{M}_T^{(\mathrm{KM})}) \leq \int_0^b b\, G^{\mathrm{ADD}}(t)\, H^{\mathrm{ADD}}(t)^{N''-1}\, dF^{\mathrm{ADD}}(t). \tag{8}$$

Here, we defined the finite-sample bias of the KM-ADD similarly to that of the KM-ARL: $\mathcal{B}_{\mathrm{FS}}(\hat{M}_T^{(\mathrm{KM})}) := \mathbb{E}[\hat{M}_T^{(\mathrm{KM})}] - M_T^{(\mathrm{KM})}$, where $M_T^{(\mathrm{KM})} := \int_0^b S^{\mathrm{ADD}}(t)dt$ is the KM-ADD with the true survival function of detection delays. Assumption (i) and (iii) are technical conditions, as previously noted below Thm. 4.1. Assumption (ii) is the independent censoring assumption for the KM-ADD, which can be justified in online QCD models by the approximation $P(\Delta\tau \mid \Delta\tau \geq 0, \nu < \infty) \approx P(\tau \mid \nu = 0)$ and $P(C := T - \nu \mid \Delta\tau \geq 0, \nu < \infty) \approx P(C := T \mid \nu = 0)$ because $P(\tau \mid \nu = 0)$ and $P(C := T \mid \nu = 0)$ are independent. This approximation implies that the distributions of the detection delay $\tau - \nu$ and the censoring time $T - \nu$ are approximately equal to their respective distributions measured from $t = 0$ rather than from $\nu$.

**Truncation bias.** $\mathcal{B}_{\mathrm{TR}}(\hat{\mu}_T^{(\mathrm{KM})})$ captures the error from sequence truncation, which dominates the total bias when sequence lengths are short. It vanishes as $a$ increases because $\mathcal{B}_{\mathrm{TR}}(\hat{\mu}_T^{(\mathrm{KM})}) = \int_0^a S^{\mathrm{ARL}}(t)dt - \mathbb{E}[\tau \mid \nu < \infty] = -\int_a^\infty S^{\mathrm{ARL}}(t) \to 0$, where we used $\mathbb{E}[\tau \mid \nu < \infty] = \int_0^\infty S^{\mathrm{ARL}}(t)dt$. The convergence rate depends on the underlying distributions. In Thm. 4.3 below, we derive a bound for $\mathcal{B}_{\mathrm{TR}}(\hat{\mu}_T^{(\mathrm{KM})})$, indicating that it is non-positive and minor than that of the conventional LB-ARL. In Sec. 5, we will empirically justify our result (Fig. 2). Proof is given in App. B.3. Note that given an evaluation dataset, we can easily specify $T_{\max}^*$ defined below (App. F).

**Theorem 4.3** (Truncation bias bound for KM-ARL). *Define the truncation bias of the LB-ARL as $\mathcal{B}_{\mathrm{TR}}(\hat{\mu}_T^{(\mathrm{LB})}) := \mu_T^{(\mathrm{LB})} - \mu_\infty$, where $\mu_T^{(\mathrm{LB})} := \mathbb{E}[\tau \mid \nu = \infty, \tau \leq T]$ is the true ARL under random lengths. For $a = T_{\max}^*$, where $T_{\max}^* := \inf\{T \mid \mathrm{CDF}(T) = 1\}$ is the least upper bound for the support of the CDF of $T$, we have $\mathcal{B}_{\mathrm{TR}}(\hat{\mu}_T^{(\mathrm{LB})}) \leq \mathcal{B}_{\mathrm{TR}}(\hat{\mu}_T^{(\mathrm{KM})}) \leq 0$.*

A similar bound holds for the KM-ADD. Proof is given in App. B.4.

**Theorem 4.4** (Truncation bias bound for KM-ADD). *Define the truncation bias of the LB-ADD as $\mathcal{B}_{\mathrm{TR}}(\hat{M}_T^{(\mathrm{LB})}) := M_T^{(\mathrm{LB})} - M_\infty$, where $M_T^{(\mathrm{LB})} := \mathbb{E}[\Delta\tau \mid \nu < \infty, 0 \leq \Delta\tau \leq \Delta T]$ is the true ADD under random sequence lengths. For $b = \Delta T_{\max}^*$, where $\Delta T_{\max}^* := \inf\{\Delta T := T - \nu \mid$*

$\mathrm{CDF}(\Delta T) = 1\}$, *we have* $\mathcal{B}_{\mathrm{TR}}(\hat{M}_T^{(\mathrm{LB})}) \leq \mathcal{B}_{\mathrm{TR}}(\hat{M}_T^{(\mathrm{KM})}) \leq 0$, *under the independent censoring assumption in Thm. 4.2.*

**Total Estimation Bias.** Finally, we examine the total estimation bias of the KM-ARL: $\mathcal{B}(\hat{\mu}_T^{(\mathrm{KM})}) = \mathcal{B}_{\mathrm{FS}}(\hat{\mu}_T^{(\mathrm{KM})}) + \mathcal{B}_{\mathrm{TR}}(\hat{\mu}_T^{(\mathrm{KM})})$ (a similar discussion below holds for the KM-ADD). From Thm. 4.1 and $\mathcal{B}_{\mathrm{TR}}(\hat{\mu}_T^{(\mathrm{KM})}) = -\int_a^\infty S^{\mathrm{ARL}}(t)$, we have for any $a \in \mathbb{R}_{\geq 0}$

$$-\int_0^a t\, G^{\mathrm{ARL}}(t)\, H^{\mathrm{ARL}}(t)^{N-1}\, dF^{\mathrm{ARL}}(t) - \int_a^\infty S^{\mathrm{ARL}}(t)$$

$$\leq \mathcal{B}(\hat{\mu}_T^{(\mathrm{KM})}) \leq \int_0^a a\, G^{\mathrm{ARL}}(t)\, H^{\mathrm{ARL}}(t)^{N-1}\, dF^{\mathrm{ARL}}(t) - \int_a^\infty S^{\mathrm{ARL}}(t). \tag{9}$$

Define $t_F := \inf\{t \in \mathbb{R}_{\geq 0} \mid F^{\mathrm{ARL}}(t) = 1\}$ and $t_H := \inf\{t \in \mathbb{R}_{\geq 0} \mid H^{\mathrm{ARL}}(t) = 1\}$, and set $a$ to $t_F$. Then, since $S^{\mathrm{ARL}} = 1 - F^{\mathrm{ARL}}$, we have $\int_a^\infty S^{\mathrm{ARL}}(t) = 0$; i.e., the truncation bias vanishes. Therefore,

$$-\int_0^{t_F} t\, G^{\mathrm{ARL}}(t)\, H^{\mathrm{ARL}}(t)^{N-1}\, dF^{\mathrm{ARL}}(t)$$

$$\leq \mathcal{B}(\hat{\mu}_T^{(\mathrm{KM})}) \leq \int_0^{t_F} t_F\, G^{\mathrm{ARL}}(t)\, H^{\mathrm{ARL}}(t)^{N-1}\, dF^{\mathrm{ARL}}(t). \tag{10}$$

If $t_F < t_H$, the bound decays exponentially as $N \to \infty$ because $H^{\mathrm{ARL}}(t) < 1$ for $t \in [0, t_F]$; i.e., *the KM-ARL is asymptotically unbiased if $t_F < t_H$.* Otherwise, there are non-vanishing terms $-\int_{t_H}^{t_F} t\, G^{\mathrm{ARL}}(t)\, dF^{\mathrm{ARL}}(t)$ and $\int_{t_H}^{t_F} t_F\, G^{\mathrm{ARL}}(t)\, dF^{\mathrm{ARL}}(t)$ in the lower and upper bounds, respectively. This is reasonable because $t_F \geq t_H$ implies that detection points may occur beyond censoring times with non-zero probability; i.e., not all detection points are observable. Consequently, estimating the ARL without bias or additional assumptions is impossible, necessitating extrapolation.

## 5 EXPERIMENT

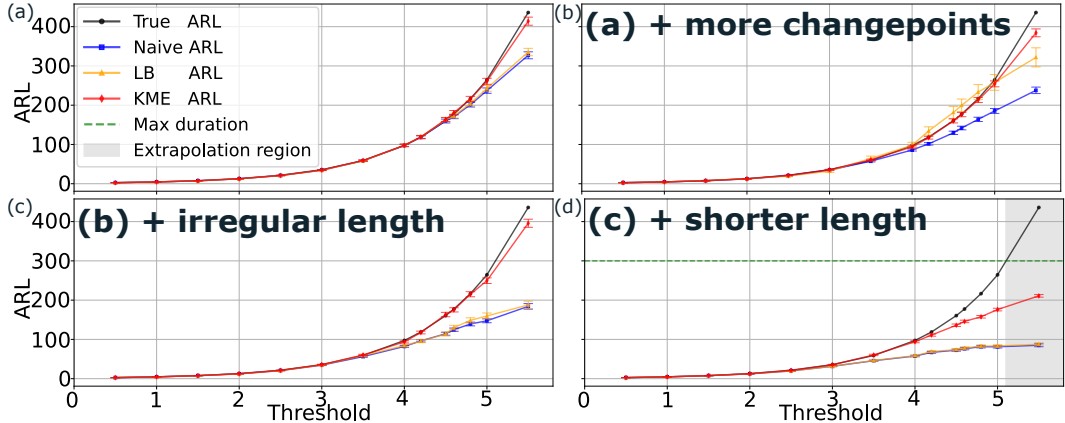

Figure 2: **Threshold of detector vs. ARL.** The KM-ARL provides more accurate estimates of the true ARL even when sequences contain changepoints and have limited and irregular lengths. The Gaussian process dataset contains 1000 sequences. The GSR with ground-truth statistics is evaluated under various thresholds. Changepoints are sampled uniformly. Error bars represent the standard error of the mean. **(a) Sequence length is** 1000, **with** 10% **of sequences containing a changepoint. (b) Sequence length is** 1000, **with** 90% **of sequences containing a changepoint. (c) Sequence lengths vary irregularly in the range** [100, 1000], **with** 90% **of sequences containing a changepoint. (d) Sequence lengths vary irregularly in the range** [30, 300], **with** 90% **of sequences containing a changepoint.** In (d), $\mathrm{ARL} > 300$ (gray area) is an extrapolation region (excluding the true ARL).

To demonstrate the practical relevance of our KM-ARL and KM-ADD for sequences with limited and irregular lengths, we use both simulated and real-world datasets. Our results show that these metrics

reduce estimation bias compared to baseline estimators, enhance robustness to such sequences, and thereby improve interpretability and facilitate empirical, intuitive model selection. Our code is given in the supplementary material and will be released upon acceptance.

**Datasets.** We use two simulation datasets, the Gaussian and Poisson processes, and one real-world dataset, the WISDM Actitracker (Kwapisz et al., 2011). The pre-change and post-change Gaussian processes have the mean of $0$ and $0.1$, respectively, with preserving the variance equal to $0.1$. We use two types of changepoint distributions: geometric and uniform. Several sequences have no changepoint, and the number of with-change sequences depends on the changepoint distributions. We also simulate irregular length by randomly truncating the sequences. The setup and experimental results for Poisson processes are provided in App. C.4 due to page limitations. The results are similar to those obtained for the Gaussian processes. The WISDM Actitracker is a large, real-world dataset for smartphone-based human activity recognition (walking, jogging, stairs, sitting, standing, and lying down) collected by the WISDM Lab at Fordham University, offering both user-labeled and machine-labeled data. The labels specify activities and their temporal intervals. We provide the results for the machine-labeled subset in the main text, and the results for the user-labeled subset are provided in Fig. 15 in App. C.5. The machine-labeled subset contains 51,326 sequences, and the sequence lengths exhibit substantial irregularity from 1 to 54,401 after our preprocesses. See App. D.3 for more statistical information of the WISDM Actitracker dataset. We remove the sequences with length 1, as they are not informative when computing performance metrics of QCD models. The preprocesses are detailed in our code and App. D.2.

### 5.1 ESTIMATION ON SIMULATION DATASET

**Detection models.** The QCD model used for the simulation datasets is the generalized Shiryaev-Roberts (GSR) procedure with ground-truth statistics. The GSR raise an alarm when the following statistic hits a pre-defined threshold: $R(t) = \omega \prod_{s=0}^{t} \mathcal{L}(s) + \sum_{k=0}^{t} \prod_{s=k}^{t} \mathcal{L}(s)$, where the likelihood ratio is denoted by $\mathcal{L}(t) = f\left(X^{(t)} \mid X^{(0,t-1)}\right) / g\left(X^{(t)} \mid X^{(0,t-1)}\right)$, and $\omega$ controls the strength of warm start and is set to $0$ in our experiments. Additionally, we provide the results for the cumulative sum (CUSUM) procedure in App. C.6.

**True ARLs & ADDs.** For the experiments on the Gaussian and Poisson process datasets, we simulate the ground-truth ARLs and ADDs by generating sequences of effectively infinite length. Their error bars are omitted because all errors are sufficiently small, with relative errors $\lesssim 10^{-3}$.

**Naive ARL.** We introduce another baseline metric, referred to as the Naive ARL. While the LB-ARL use only sequences with $\nu_i = \infty$ and $\tau_i \le T_i$ for computing the ARL, the Naive ARL utilizes sequences with $\tau_i < \nu_i$ and $\tau_i \le T_i$: $\hat{\mu}_T^{(NV)} := \langle \tau_i \rangle_{i:\tau_i < \nu_i, \tau_i \le T_i}$. The number of sequences used for the Naive ARL is larger than that of the LB-ARL because the condition $\tau_i < \nu_i \wedge \tau_i \le T_i$ includes $\nu_i < \infty \wedge \tau_i \le T_i$; however, the Naive ARL has a non-vanishing bias because $\mathbb{E}[\hat{\mu}_T^{(NV)}] = \mathbb{E}[\tau \mid \tau < \nu, \tau < T]$ under minor assumptions, which is not equal to $\mathbb{E}[\tau \mid \nu = \infty, \tau < T]$ or $\mathbb{E}[\tau \mid \nu = \infty]$.

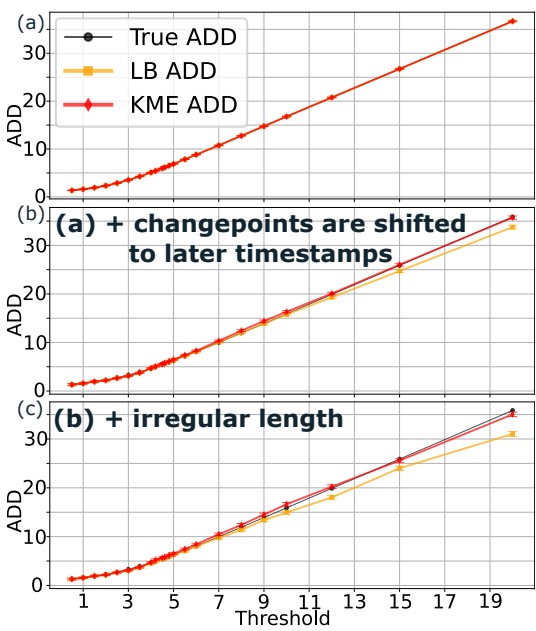

Figure 3: **Threshold of detector vs. ADD.** The KM-ADD provides more accurate estimates of the true ADD even when sequence lengths are limited and irregular. The Gaussian process dataset has 10000 sequences. Changepoints are sampled from the geometric distribution with the success probability $p \in [0, 1]$. A smaller $p$ leads to more sparse and delayed changepoints. Other settings are identical to that of Fig. 2. **(a) Sequence length is** 100, **with** $p = 0.25$. **(b) Sequence length is** 100, **with** $p = 0.001$. The changepoints are shifted to later timestamps, decreasing the chance of detection due to the length limit. **(c) Sequence lengths vary irregularly in the range** $[10, 100]$, **with** $p = 0.001$.

**Result on ARL.** Fig. 2 presents the true ARL, Naive ARL, LB-ARL, and KM-ARL across various GSR thresholds, demonstrating that the KM-ARL provides more accurate estimates of the true ARL even when sequences contain changepoints and have limited and irregular lengths. Fig. 2(a) simulates an ideal light-censoring scenario, where most detection points occur before changepoints or truncation. The true ARL is less than $T_{\max}$, the sequence length $T$ is constant (=1000), and only 10% of the sequences contain a changepoint. Fig. 2(b) simulates a heavier-censoring scenario, where 90% of the sequences contain a changepoint. Fig. 2(c) simulates an even more heavily censored scenario, where, in addition to Fig. 2(b), sequence lengths are irregular and randomly sampled from $[100, 1000]$. Our KM-ARL remains robust under this condition, utilizing with-change and truncated sequences, unlike the Naive ARL and the LB-ARL. Fig. 2(d) simulates a challenging and realistic scenario, where, in addition to Fig. 2(c), $T_{\max}$ is reduced to 300, and sequence lengths are sampled from $[30, 300]$. The gray area indicates ARL $> T_{\max} = 300$, where no data are observed, i.e., an extrapolation region (excluding the true ARL). Bias becomes non-negligible in ARL $\gtrsim T_{\max}$ due to truncation. In this region, no metric can reliably estimate the ARL without bias or additional assumptions, as noted Sec. 4. To further mitigate truncation bias, one may combine our estimators with parametric extrapolation methods for the survival function, such as those proposed in (Sahki et al., 2020).

**Result on ADD.** Fig. 3 presents the true ADD, LB-ADD, and KM-ADD across various GSR thresholds, demonstrating that the KM-ADD provides more accurate estimates of the true ADD even when sequences have limited and irregular lengths. Fig. 3(a) simulates an ideal light-censoring scenario, where most detection points occur before truncation. $T$ is constant and set to 100, and $\Delta T_{\max}$ is also set to 100, which is greater than the true ADD. Fig. 3(b) simulates a heavier-censoring scenario, where changepoints are shifted to later timestamps, reducing the chance of detection before truncation. Fig. 3(c) simulates a challenging and realistic scenario, where, in addition to Fig. 3(b), sequence length are irregular and sampled from $[10, 100]$, further reducing the chance of detection before truncation. The KM-ADD still remains robust, utilizing truncated sequences, unlike the conventional LB-ADD.

**ARL-ADD tradeoff curve.** Fig. 4 shows ARL-ADD tradeoff curves, which are theoretically related to the optimality of QCD models and are used for model evaluation in practice. Fig. 4 demonstrates that the KM-ARL and KM-ADD can more accurately estimate the ARL-ADD curve than the others. Fig. 4(a) simulates a light-censored scenario, where most detection points occur before censoring. The sequence length $T$ is constant and set to $T = T_{\max} = 500$. Fig. 4(b) simulates a challenging and realistic scenario, where censoring is heavier. Sequence lengths are irregular and sampled from $[50, 500]$. The KM-ARL and KM-ADD remains more robust than the baselines. Again, bias is non-negligible in the region ARL $\approx T_{\max} = 500$, where extrapolation region is nearby.

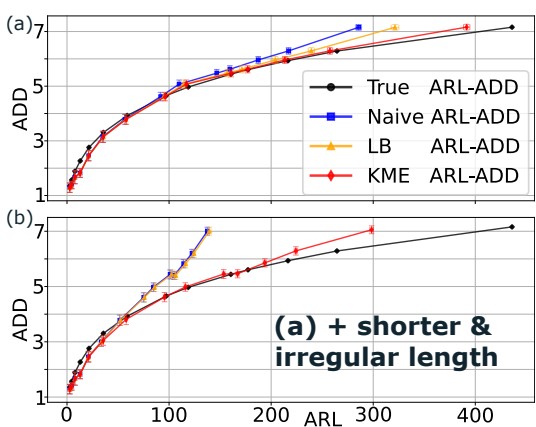

Figure 4: **ARL-ADD tradeoff curves.** The KM-ARL and KM-ADD provide more accurate estimates of the true ARL-ADD curve even when sequences contain changepoints and have limited and irregular lengths. The setup is identical to that of Fig. 2, except that the dataset size is 10000. 50% of sequences contain a changepoint. The LB-ADD is used for the Naive ARL's ADD. **(a) Sequence length is** 1000. **(b) Sequence lengths vary irregularly in the range** $[50, 500]$. The complete ablation study is given in App. C.2.

## 5.2 EVALUATION OF MODELS ON REAL-WORLD DATASET

We further evaluate six QCD models on a real-world, challenging dataset: the WISDM Actitracker (Kwapisz et al., 2011), which contains 51,326 sequences with substantial irregularity of lengths from 1 to 54,401. The computational costs for KM-ARL and KM-ADD are negligible in our experiments compared with the time required to run the QCD algorithms.

**QCD models.** We use the following online QCD models, including window-based, frame-based, parametric, and non-parametric models: Window L1 (Bai, 1995), Window Normal (Lavielle, 1999; Lavielle & Teyssiere, 2006), Window AR (Bai, 2000), non-parametric focused changepoint detection (NP-FOCuS) (Romano et al., 2024a), CUSUM (Page, 1954), and exponentially weighted moving average (EWMA) (Roberts, 1959). The window size and burn-in interval are fixed at 30, ensuring much smaller than the average length. Most other hyperparameters left at default values. See App. D.1 for details of the models and their hyperparameters.

**Result.** The ARL-ADD tradeoff curves are presented in Fig. 1, demonstrating that our KM-ARL and KM-ADD are more robust even when censoring is significantly heavy. Fig. 1(a) shows that LB-ARL and LB-ADD become unstable and exhibit high variance, particularly at large QCD thresholds. This instability arises because the detector often fails to raise an alarm within the sequence length, and the limited number of sequences further amplifies the empirical variance. In contrast, Fig. 1(b) shows that our KME-based estimators do not suffer from this issue. These metrics reduce variance compared to baseline estimators because they are calculated from a constant number of sequences, regardless of the threshold, which enhances their robustness against censoring. Therefore, the KM-ARL and KM-ADD improves interpretability and facilitating empirical, intuitive model selection (see App. D.4 for more details of the variance computation).

## 6 DISCUSSION, LIMITATIONS, AND FUTURE WORK

**To further reduce estimation bias.** For further bias reduction, one may use the bootstrap method (Efron & Tibshirani, 1994) to mitigate the finite-sample bias, although this effect is typically overshadowed by the truncation bias. To address the truncation bias beyond extrapolation, a parametric estimator can be combined with our estimators, as in (Sahki et al., 2020); however, note that the estimation accuracy deteriorates when the parametric assumption is invalid.

**Changepoint isolation and multiple changepoint detection.** Our work assumes a single changepoint type; however, changepoint isolation (classification) (Tartakovsky et al., 2014) is often required in practice. We conjecture that our survival analysis–based approach can be extended to this challenging setting. A promising direction is to use survival models under *competitive risks* (Morita, 2021), treating different changepoint types as distinct causes of death. Similarly, while our method does not currently support multiple changepoint detection (Niu et al., 2016), it may leverage *multi-state models under competing risks* (Therneau et al., 2020; Beyersmann et al., 2011; Mills, 2011), which are well established in survival analysis.

**Probability of false alarms.** The probability of false alarms (PFA) is another standard metric in QCD theory, commonly used to establish Bayesian optimality (Tartakovsky, 2019). However, it is also affected by right-censoring in real-world scenarios. We hypothesized that multi-state models under competing risks, such as the Aalen–Johansen estimator (AJE) (Aalen & Johansen, 1978), could alleviate this issue. Our preliminary experiments, however, failed to accurately estimate PFAs, possibly because the AJE assumes no event ties, an assumption often violated when continuous-time sequences are discretized into frames. This finding suggests that more careful approaches, such as those from discrete-time survival analysis (Tutz et al., 2016), are required.

**Dependent censoring.** In Sec. 4, we assume independent censoring for KM-ADD, supported by the distributional approximation therein. This assumption can potentially be relaxed by leveraging extensive research on dependent censoring in survival analysis (Hsu & Taylor, 2010; Lin et al., 2023; Crommen et al., 2025). Doing so would not only eliminate the assumption but also extend our bias analysis to offline QCD, where censoring depends on detection. See App. F for more details.

**Heavy censoring.** We do not recommend using KM-ARL or KM-ADD when datasets are severely imbalanced between pre-change and post-change sequences because this leads to significantly heavy censoring and inflated finite-sample and truncation bias. While the extent of acceptable censoring depends on the dataset and underlying distributions, Malmquist (2025) reports that, for 30, 50, 100, or 150 subjects (sequences), censoring rates up to 90% are tolerable if censoring is uniform. However, for dependent censoring occurring just before event time (detection), estimation bias increases sharply

(see App. F). Nonetheless, several studies have proposed modifications to the KME to address heavy censoring (Shafiq et al., 2007; Zare & Mahmoodi, 2013), which can be integrated with our KM-ARL and KM-ADD.

## REPRODUCIBILITY STATEMENT

For reproducibility, we provide the code for our experiments and proposed estimators in the Supplementary Materials, which will be publicly available upon acceptance. Complete descriptions of data processing steps for all datasets are given in Sec. 5, App. D, and in our code. Assumptions, definitions, and full proofs of our theoretical results appear in Sec. 3, Sec. 4, and App. B.

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

APPENDIX

# Contents

## A QCD-SURVIVAL ANALYSIS CORRESPONDENCE

**Proposed Correspondence.** We summarize our key idea, an analogy between QCD and survival analysis in Tab. 1.

| ARL | ADD | Survival analysis |
|---|---|---|
| detection time $\tau_i$ | detection delay $\tau_i - \nu_i$ | event time |
| $\min\{\nu_i, T_i\}$ | $T_i - \nu_i$ | censoring time $C_i$ |
| KM-ARL $\hat{\mu}_T^{(\mathrm{KM})}$ | KM-ADD $\hat{M}_T^{(\mathrm{KM})}$ | restricted mean survival time |

Table 1: QCD-survival analysis analogy.

**Illustration.** For clarity, we illustrate the original KME, KM-ARL, and the computation of $d_j$ and $n_j$ in Fig. 5.

### A.1 BACKGROUND OF KME IN SURVIVAL ANALYSIS

#### A.1.1 INTRODUCTION

We provide the motivation and intuition for the original KME in survival analysis Kaplan & Meier (1958); Kleinbaum & Klein (1996). Let $\tau$ be a nonnegative random variable representing a lifetime, and let

$$S(t) := P(\tau > t) \tag{11}$$

denote the survival function. In the ideal (textbook) setting without censoring, and with i.i.d. samples $\tau_1, \ldots, \tau_N$, a naive nonparametric estimator of $S(t)$ is the following empirical survival function

$$\hat{S}_{\mathrm{emp}}(t) = \frac{1}{N} \sum_{i=1}^{N} \mathbb{1}\{\tau_i > t\}. \tag{12}$$

However, in practice, we often do *not* observe all $\tau_i$ fully. Instead, some subjects drop out, are lost to follow-up, or the study ends before they fail. This leads to *right-censoring*, where we observe

$$Y_i = \min(\tau_i, C_i), \qquad \Delta_i = \mathbb{1}\{\tau_i \leq C_i\}, \tag{13}$$

with $C_i$ a censoring time. Then $Y_i$ is either the failure time (if $\Delta_i = 1$) or a censoring time (if $\Delta_i = 0$), where $\Delta_i$ is the event flag.

If we treat censored observations as if they were failures, we underestimate survival; if we drop them entirely, we discard information about the period in which we do know they survived. The KME is motivated precisely by this need:

- use all information available prior to censoring;
- do not make parametric assumptions (nonparametric);
- remain interpretable as a product of empirical conditional survival probabilities.

We derive the KME formula in the following to clarify its motivation.

#### A.1.2 DECOMPOSITION VIA CONDITIONAL SURVIVAL PROBABILITIES

A key idea is to write $S(t)$ as a product of conditional survival probabilities at each distinct failure time (or event time, i.e., the time of death). Let

$$t_{(1)} < t_{(2)} < \cdots < t_{(K)} \tag{14}$$

be the distinct *failure* times in the population (not yet the sample). For $t$ between $t_{(j)}$ and $t_{(j+1)}$, we can write

$$S(t) = P(\tau > t) = \prod_{t_{(m)} \leq t} P\big(\tau > t_{(m)} \,\big|\, \tau \geq t_{(m)}\big). \tag{15}$$

This is just repeated application of the chain rule of probability. Intuitively,

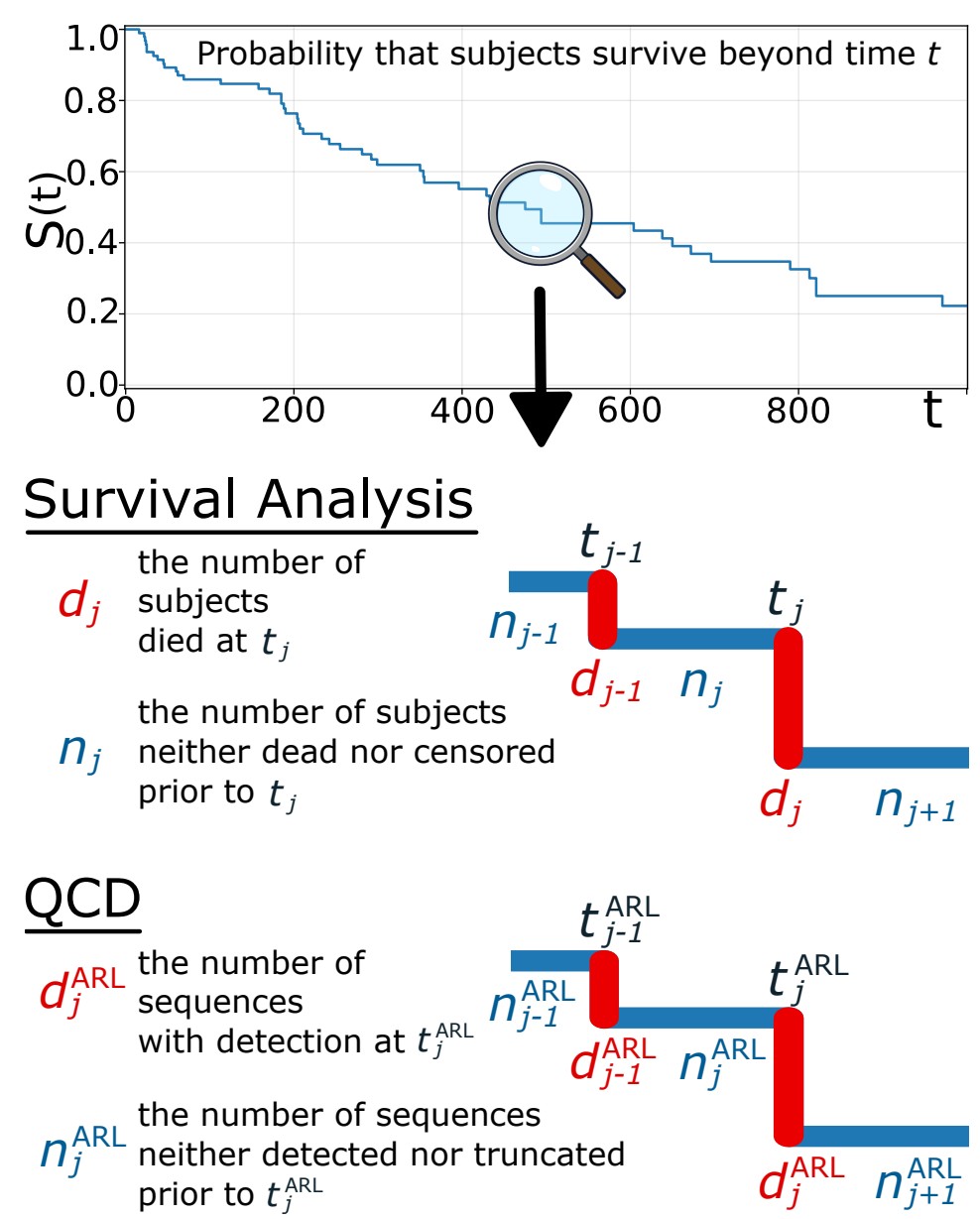

Figure 5: **Original KME, KM-ARL, and computation of** $d_j$ **and** $n_j$**.** we estimate the *survival function of detection points* $S^{\text{ARL}}(t) := P(\tau > t \mid \nu = \infty)$ using the Kaplan-Meier estimator (KME) (Kaplan & Meier, 1958), a non-parametric estimator of the survival function: $\hat{S}^{\text{ARL}}(t) = \prod_{j:t_j^{\text{ARL}} \leq t}(1 - \frac{d_j^{\text{ARL}}}{n_j^{\text{ARL}}})$, where $0 < t_1^{\text{ARL}} < t_2^{\text{ARL}} < \cdots < t_{N'}^{\text{ARL}}$ are distinct detection points, with $N'_{\text{ARL}} \in \mathbb{N} (\leq N)$; $d_j^{\text{ARL}} := |\{i \in [N] \mid \tau_i = t_j^{\text{ARL}}\}|$ is the number of sequences with a detection at $t_j^{\text{ARL}}$ $(j \in [N'_{\text{ARL}}])$; and $n_j^{\text{ARL}} := |\{i \in [N] \mid \min\{\tau_i, C_i^{\text{ARL}}\} \geq t_j^{\text{ARL}}\}|$ is the number of sequences neither detected nor censored prior to $t_j^{\text{ARL}}$ $(j \in [N'_{\text{ARL}}])$, with the censoring time of sequence $i$ defined as $C_i^{\text{ARL}} := \min\{\nu_i, T_i\}$. We propose a non-parametric estimator of the ARL under irregular sequence lengths, termed KM-ARL, as the integral of $\hat{S}^{\text{ARL}}(t)$ over the range $[0, a]$ for arbitrary $a \in \mathbb{R}_{\geq 0}$: $\hat{\mu}_T^{(\text{KM})} := \int_0^a \hat{S}^{\text{ARL}}(t)dt$.

- at each time $t_{(m)}$, we look at those who are still alive (have "survived so far");
- we multiply by the probability that they survive *beyond* $t_{(m)}$;
- the overall survival up to time $t$ is the product of these step-by-step survival probabilities.

The KM estimator simply replaces each conditional probability $P(\tau > t_{(m)} \mid \tau \geq t_{(m)})$ by a natural empirical estimate using the sample with censoring.

### A.1.3 RISK SET $n_j$ AND NUMBER OF EVENTS $d_j$

Suppose we have $N$ subjects, and we observe $(Y_i, \Delta_i)$ for $i = 1, \ldots, N$. Let

- $t_{(1)} < \cdots < t_{(J)}$ be the distinct observed *failure* times in the sample (i.e., times where at least one $\Delta_i = 1$ and $Y_i = t_{(j)}$),
- $d_j$ be the number of failures at time $t_{(j)}$,
- $n_j$ be the size of the *risk set* at time $t_{(j)}$, i.e., the number of subjects who are known to be alive and uncensored just *before* $t_{(j)}$.

**Intuition for the risk set $n_j$.** The risk set $n_j$ at time $t_{(j)}$ collects all individuals for whom failure at $t_{(j)}$ is still a possibility:

- subjects who failed earlier ($Y_i < t_{(j)}$ and $\Delta_i = 1$) are removed: they are already dead;
- subjects who were censored earlier ($Y_i < t_{(j)}$ and $\Delta_i = 0$) are removed: we no longer observe them, and we cannot know whether they would have failed at $t_{(j)}$ or not;
- subjects whose observed time $Y_i$ is at least $t_{(j)}$ remain in the risk set: we know they survived at least up to just before $t_{(j)}$.

Thus,

$$n_j = \sum_{i=1}^{N} \mathbb{1}\{Y_i \geq t_{(j)}\}. \tag{16}$$

This definition uses all partial information: even if some $Y_i$ correspond to future censoring events, until they are censored, they contribute to the information that the subject has survived up to that time.

**Intuition for the number of failures $d_j$.** At $t_{(j)}$, we also count the number of failures:

$$d_j = \sum_{i=1}^{N} \mathbb{1}\{Y_i = t_{(j)}, \ \Delta_i = 1\}. \tag{17}$$

Only *failures* contribute to $d_j$. Censoring at time $t_{(j)}$ does not count as a failure; it simply removes subjects from future risk sets (for times after $t_{(j)}$). This aligns with the goal of estimating the distribution of failure times, not censoring times.

### A.1.4 WHY $d_j/n_j$? CONDITIONAL PROBABILITY VIEWPOINT

Consider the underlying quantity

$$P(\tau = t_{(j)} \mid \tau \geq t_{(j)}) \tag{18}$$

as the probability that a subject who has survived up to $t_{(j)}$ fails exactly at $t_{(j)}$. In a discrete-time picture, this is analogous to a "hazard probability" at $t_{(j)}$. Given $n_j$ subjects in the risk set at $t_{(j)}$, and $d_j$ of them failing at $t_{(j)}$, it is natural to estimate this conditional probability by the empirical proportion

$$\hat{P}(\tau = t_{(j)} \mid \tau \geq t_{(j)}) = \frac{d_j}{n_j}. \tag{19}$$

Consequently, the conditional survival probability at $t_{(j)}$ is estimated by

$$\hat{P}(\tau > t_{(j)} \mid \tau \geq t_{(j)}) = 1 - \frac{d_j}{n_j}. \tag{20}$$

Note that censored individuals are *included* in $n_j$ as long as they are not yet censored at $t_{(j)}$, but never in $d_j$. This reflects the idea that:

- up to the censoring time, their "survival experience" is fully observed and contributes to the risk set $n_j$;

- at the censoring time, we stop knowing what happens afterward, so they drop from future risk sets (but are not treated as failures $d_j$).

### A.1.5 KME AS PRODUCT-LIMIT ESTIMATOR

Putting everything together, the KM estimator of $S(t)$ is defined as

$$\hat{S}_{\mathrm{KM}}(t) = \prod_{t_{(j)} \leq t} \left( 1 - \frac{d_j}{n_j} \right). \tag{21}$$

This is exactly the product of estimated conditional survival probabilities:

$$\hat{S}_{\mathrm{KM}}(t) = \prod_{t_{(j)} \leq t} \hat{P}(\tau > t_{(j)} \mid \tau \geq t_{(j)}). \tag{22}$$

Intuitively:

- At the beginning ($t < t_{(1)}$), survival is 1: no one has failed.

- At each failure time $t_{(j)}$, we shrink the survival curve by multiplying by $1 - d_j/n_j$.

- Between failure times, $\hat{S}_{\mathrm{KM}}(t)$ is constant, yielding a right-continuous step function.

This construction uses only the ordering of failure and censoring times and assumes (in the standard theory) that censoring is non-informative (independent of the failure mechanism, given the observed history). Under these conditions, the KM estimator can be seen as the nonparametric maximum likelihood estimator of $S(t)$ under right-censoring.

In summary, the roles of $d_j$ and $n_j$ are:

- $n_j$ is the size of the risk set just before $t_{(j)}$. It collects all individuals whose failure at $t_{(j)}$ is still observable: they have not failed and have not been censored yet. This ensures that $d_j/n_j$ is a valid empirical conditional probability.

- $d_j$ is the number of failures at $t_{(j)}$. It quantifies how much the survival curve should drop at that time, relative to the current risk set.

- The ratio $d_j/n_j$ is the natural empirical estimator of the conditional failure probability at $t_{(j)}$. Its complement $1 - d_j/n_j$ is the empirical conditional survival probability.

- The KM estimator is then the product over time of these conditional survival probabilities, giving a nonparametric estimator of the entire survival function that properly accounts for right-censoring.

## B PROOFS

Our proofs for the finite-sample bounds are based on (Stute, 1994). In our proofs for the finite-sample bounds, we idealize the time index as continuous without loss of generality, for technical convenience. Define $\nu_1, \ldots, \nu_N$ as i.i.d. random variables (the changepoints), independent of the observations of sequence, where $N \in \mathbb{N}$. Assume that $\tau_1, \ldots, \tau_N$ are independent and identically distributed (i.i.d.) random variables (the detection points of a QCD model). Assume that $T_1, \ldots, T_N$ are i.i.d. random variables (the lengths of sequences), independent of the observations of sequences.

## B.1 PROOF OF THEOREM 4.1

Let $C_1^{\text{ARL}}, \ldots, C_N^{\text{ARL}}$ are i.i.d. random variables (the censoring times for the KM-ARL), defined as $C_i^{\text{ARL}} := \min\{\nu_i, T_i\}$ for $i \in [N]$. Assume that $C^{\text{ARL}}$ is independent of $\tau$, called the independent censoring assumption (or non-informative censoring assumption) for the KM-ARL. This assumption holds for online QCD models that do not look ahead the input sequence, which are the focus of this paper. This is because: detection points $\tau_i$ prior to censoring can be regarded as samples from $P(\tau \mid \nu = \infty)$; $P(\tau \mid \nu = \infty)$, $P(\nu)$, and $P(T)$ are independent; and thus, $P(\tau \mid \nu = \infty)$ and $P(C^{\text{ARL}} := \min\{\nu, T\})$ are also independent. Here, \$P\$ denotes a probability or a probability measure, with slight abuse of notation throughout the paper. In contrast, the independent censoring assumption does not hold for offline changepoint detection models because $\tau$ depends on $X^{(0,T)}$, which in turn depends on $\nu$ and $T$. Let $F^{\text{ARL}}$ and $G^{\text{ARL}}$ denote the cumulative distribution functions (CDFs) of $\tau$ and $C^{\text{ARL}}$, respectively. The survival function of detection points is defined as $S^{\text{ARL}}(t) := 1 - F^{\text{ARL}}(t)$, where $t \in \mathbb{R}_{\geq 0}$.

We adopt a different convention of the KME from that in the main text. Consider the random variables

$$Z_i^{\text{ARL}} := \min(\tau_i, C_i^{\text{ARL}}), \tag{23}$$

$$\delta_i^{\text{ARL}} := \mathbb{1}_{\{\tau_i < C_i^{\text{ARL}}\}}, \tag{24}$$

where $\delta_i^{\text{ARL}}$ is the indicator representing whether detection $\tau_i$ has been observed before a changepoint or the end of the sequence. Let $H^{\text{ARL}}$ denote the CDF of $Z^{\text{ARL}}$, and assume that it is continuous on $t \in \mathbb{R}_{\geq 0}$. Let $Z_{1:N}^{\text{ARL}} \leq Z_{2:N}^{\text{ARL}} \leq \cdots \leq Z_{N:N}^{\text{ARL}}$ be the *order statistics* of $Z^{\text{ARL}}$ (i.e., $Z_i^{\text{ARL}}$ are sorted in increasing order) and $\delta_{[i:N]}^{\text{ARL}}$ be the *concomitant* of $Z_{i:N}^{\text{ARL}}$; i.e., $\delta_i^{\text{ARL}}$ are sorted according to $Z_{i:N}^{\text{ARL}}$, meaning that $\delta_{[i:N]}^{\text{ARL}} = \delta_j$ if $Z_{i:N}^{\text{ARL}} = Z_j^{\text{ARL}}$. The estimation of $S^{\text{ARL}}(t)$ is given by

$$1 - \hat{F}_N^{\text{ARL}}(t) := \hat{S}^{\text{ARL}}(t) = \prod_{i=1}^{N}\left(1 - \frac{\delta_{[i:N]}^{\text{ARL}}}{N - i + 1}\right)^{\mathbb{1}(Z_{i:N}^{\text{ARL}} \leq t)}, \tag{25}$$

where $t \in \mathbb{R}_{\geq 0}$ This is equivalent to the convention in the main text, which can be confirmed by canceling factors telescopically. In case of ties, treat a death as if it occurs before a censoring at the same time point. Our KM-ARL is formally defined as

$$\hat{\mu}_T^{(\text{KM})} = \int_0^a t \, d\hat{F}_N^{\text{ARL}}, \tag{26}$$

where $a \in \mathbb{R}_{\geq 0}$ is arbitrary.

**Theorem B.1** (Finite-sample bias bounds for KM-ARL (Thm. 4.1))**.** *We idealize the time index as continuous without loss of generality, for technical convenience. Let $F^{\text{ARL}}, G^{\text{ARL}}$, and $H^{\text{ARL}}$ be the CDFs of $\tau$, $C^{\text{ARL}}$, and $Z^{\text{ARL}}$, respectively. Assume that (i) $F^{\text{ARL}}$ and $G^{\text{ARL}}$ do not have common discontinuities, (ii) $\tau$ and $C^{\text{ARL}}$ are independent, known as the independent censoring (or non-informative censoring), and (iii) $H^{\text{ARL}}$ is continuous. Then, we have for any $a \in \mathbb{R}_{\geq 0}$*

$$-\int_0^a t \, G^{\text{ARL}}(t) \, H^{\text{ARL}}(t)^{N-1} \, F^{\text{ARL}}(dt) \tag{27}$$

$$\leq \mathcal{B}_{\text{FS}}(\hat{\mu}_T^{(\text{KM})}) \leq \tag{28}$$

$$\int_0^a a \, G^{\text{ARL}}(t) \, H^{\text{ARL}}(t)^{N-1} \, F^{\text{ARL}}(dt), \tag{29}$$

Assumption (i) is a technical requirement and is valid unless pathological situations are considered. In most cases, it holds with probability 1 because we consider continuous time. Note that $F^{\text{ARL}}$ and $G^{\text{ARL}}$ may have separate discontinuities. Assumption (ii) holds for online QCD models that do not look ahead the input sequence, which are the focus of this paper. This is because: detection points $\tau_i$ prior to censoring can be regarded as samples from $P(\tau \mid \nu = \infty)$; $P(\tau \mid \nu = \infty)$, $P(\nu)$, and $P(T)$ are independent; and thus, $P(\tau \mid \nu = \infty)$ and $P(C := \min\{\nu, T\})$ are also independent. In contrast, Assumption (ii) does not hold for offline changepoint detection models because $\tau$ depends on $X^{(0,T)}$, which in turn depends on $\nu$ and $T$. Assumption (iii) is also a technical requirement.

*Proof.*

$$\int_0^a t \, dF^{\mathrm{ARL}}(t)$$

$$= \int_0^a \left( \int_0^a \mathbb{1}(t > s) \, ds \right) dF^{\mathrm{ARL}}(t) \tag{30}$$

$$= \int_0^a \left( \int_0^a \mathbb{1}(t > s) \, dF^{\mathrm{ARL}}(t) \right) ds \tag{31}$$

$$= \int_0^a \left( \int_s^a dF^{\mathrm{ARL}}(t) \right) ds \tag{32}$$

$$= \int_0^a \left( F^{\mathrm{ARL}}(a) - F^{\mathrm{ARL}}(s) \right) ds \tag{33}$$

$$= a F^{\mathrm{ARL}}(a) - a + \int_0^a S^{\mathrm{ARL}}(t) \, dt. \tag{34}$$

Similarly, we have

$$\int_0^a t \, d\hat{F}^{\mathrm{ARL}}(t) = a\hat{F}^{\mathrm{ARL}}(a) - a + \int_0^a \hat{S}^{\mathrm{ARL}}(t) \, dt. \tag{35}$$

Thus, the finite-sample bias is given by

$$B_{\mathrm{FS}}\big(\hat{\mu}_T^{(\mathrm{KM})}\big)$$

$$= \mathbb{E}\Big[ \int_0^a \hat{S}^{\mathrm{ARL}}(t) \, dt \Big] - \int_0^a S^{\mathrm{ARL}}(t) \, dt$$

$$= \mathbb{E}\Big[ \int_0^a t \, d\hat{F}^{\mathrm{ARL}}(t) \Big] - \int_0^a t \, dF^{\mathrm{ARL}}(t)$$

$$\quad - a\big( \mathbb{E}[\hat{F}^{\mathrm{ARL}}(a)] - F^{\mathrm{ARL}}(a) \big)$$

$$=: I_1 - a I_2, \tag{36}$$

where we defined

$$I_1 = \mathbb{E}\Big[ \int_0^a t \, d\hat{F}^{\mathrm{ARL}}(t) \Big] - \int_0^a t \, dF^{\mathrm{ARL}}(t) \tag{37}$$

$$I_2 = \mathbb{E}[\hat{F}^{\mathrm{ARL}}(a)] - F^{\mathrm{ARL}}(a). \tag{38}$$

We bound them by invoking the following lemma from (Stute, 1994), adapted to our setting. The full proof of Lem. B.2 is lengthy but given in (Stute, 1994). Refer also to (Stute & Wang, 1993), on which Stute (1994) builds his result.

**Lemma B.2** (Cor. 1.1 in (Stute, 1994)). *For any $F^{\mathrm{ARL}}$, $G^{\mathrm{ARL}}$, $H^{\mathrm{ARL}}$, and $\hat{F}^{\mathrm{ARL}}$ that satisfy the assumptions described in this section, the following inequality holds for any $N \in \mathbb{N}$, where $\varphi(t) \geq 0$ is a Borel measurable function on $t \in \mathbb{R}_{\geq 0}$:*

$$- \int \varphi(t) \, G^{\mathrm{ARL}}(t) \, H^{\mathrm{ARL}}(t)^{N-1} \, dF^{\mathrm{ARL}}(t)$$

$$\leq \mathbb{E}[ \int \varphi(t) d\hat{F}^{\mathrm{ARL}}(t) ] - \int \varphi(t) dF^{\mathrm{ARL}}$$

$$\leq 0. \tag{39}$$

According to Lem. B.2, for $\varphi(t) = \mathbb{1}(t \in [0, a])$, we have

$$- \int_0^a G^{\mathrm{ARL}}(t) \, H^{\mathrm{ARL}}(t)^{N-1} \, dF^{\mathrm{ARL}}(t)$$

$$\leq \mathbb{E}[ \int_0^a d\hat{F}^{\mathrm{ARL}}(t) ] - \int_0^a dF^{\mathrm{ARL}} \leq 0. \tag{40}$$

Since $\mathbb{E}[\int_0^a d\hat{F}^{\mathrm{ARL}}(t)] - \int_0^a dF^{\mathrm{ARL}} = \mathbb{E}[\hat{F}^{\mathrm{ARL}}(a)] - F^{\mathrm{ARL}}(a) = I_2$, we have

$$-\int_0^a G^{\mathrm{ARL}}(t)\, H^{\mathrm{ARL}}(t)^{N-1}\, dF^{\mathrm{ARL}}(t) \le I_2 \le 0. \tag{41}$$

Additionally, for $\varphi(t) = t\, \mathbb{1}(t \in [0,1])$, we have

$$-\int_0^a t\, G^{\mathrm{ARL}}(t)\, H^{\mathrm{ARL}}(t)^{N-1}\, dF^{\mathrm{ARL}}(t)$$

$$\le \mathbb{E}[\int_0^a t d\hat{F}^{\mathrm{ARL}}(t)] - \int_0^a t dF^{\mathrm{ARL}}(= I_1) \le 0. \tag{42}$$

From Eqs. (36), (37), (38), (41), and (42), we have proved

$$-\int_0^a t\, G^{\mathrm{ARL}}(t)\, H^{\mathrm{ARL}}(t)^{N-1}\, dF^{\mathrm{ARL}}(t)$$

$$\le I_1 - aI_2 \ \ (= B_{\mathrm{FS}}(\hat{\mu}_T^{(\mathrm{KM})})) \le$$

$$\int_0^a a\, G^{\mathrm{ARL}}(t)\, H^{\mathrm{ARL}}(t)^{N-1}\, dF^{\mathrm{ARL}}(t) \tag{43}$$

for arbitrary $a \in \mathbb{R}_{\ge 0}$. □

### B.2 PROOF OF THEOREM 4.2

Consider only sequences with $\nu < \infty$ and $\Delta\tau \ge 0$ only, as others do not contribute to the computation of the ADD, defined as $\mathbb{E}[\tau - \nu \mid \tau \ge \nu, \nu < \infty]$. In this section, we use $N \in \mathbb{N}$ as the number of sequences with $\nu_i < \infty$ and $\Delta\tau_i \ge 0$ in the dataset. Assume that $\Delta\tau_1, \ldots, \Delta\tau_N$ are independent and identically distributed (i.i.d.) random variables (the detection delays of a QCD model), defined as $\Delta\tau_i := \tau_i - \nu_i$. Let $C_1^{\mathrm{ADD}}, \ldots, C_N^{\mathrm{ADD}}$ are i.i.d. random variables (the censoring times for the KM-ADD), defined as $C_i^{\mathrm{ADD}} := \Delta T_i = T_i - \nu_i$ for $i \in [N]$. Assume that $C^{\mathrm{ADD}}$ is independent of $\Delta\tau$, called the independent censoring assumption (or non-informative censoring assumption) for the KM-ADD. The independent censoring assumption is justified in online QCD models by the approximate $P(\Delta\tau \mid \Delta\tau \ge 0, \nu < \infty) \approx P(\tau \mid \nu = 0)$ and $P(C := \Delta T \mid \Delta\tau \ge 0, \nu < \infty) \approx P(C := T \mid \nu = 0)$ because $P(\tau \mid \nu = 0)$ and $P(C := T \mid \nu = 0)$ are independent. This indicates that the distributions of detection delay $\tau - \nu$ and censoring time $T - \nu$ are approximately equal to the distributions of them measured from $t = 0$, not $\nu$. Under this approximation, the definition of the ADD becomes $\mathbb{E}[\tau \mid \nu = 0]$, which is called the steady-state ARL (Saccucci & Lucas, 1990; Sasikumr & Sujatha, 2025; Lim & Lee, 2025) and often used in the theory of control charts. For simplicity, we assume the independent censoring in the proof of Thm. 4.2, and define $F^{\mathrm{ADD}}$, $G^{\mathrm{ADD}}$, and $H^{\mathrm{ADD}}$ accordingly. The survival function of detection delays is defined as $S^{\mathrm{ADD}}(t) := 1 - F^{\mathrm{ADD}}(t)$, where $t \in \mathbb{R}_{\ge 0}$.

Consider the random variables

$$Z_i^{\mathrm{ADD}} := \min(\Delta\tau_i, C_i^{\mathrm{ADD}}), \tag{44}$$

$$\delta_i^{\mathrm{ADD}} := \mathbb{1}_{\{\Delta\tau_i < C_i^{\mathrm{ADD}}\}}, \tag{45}$$

where $\delta_i^{\mathrm{ADD}}$ is the indicator representing whether the detection $\tau_i$ has been observed before the end of the sequence. Let $H^{\mathrm{ADD}}$ denote the CDF of $Z^{\mathrm{ADD}}$, and assume that it is continuous on $t \in \mathbb{R}_{\ge 0}$. Let $Z_{1:N}^{\mathrm{ADD}} \le Z_{2:N}^{\mathrm{ADD}} \le \cdots \le Z_{N:N}^{\mathrm{ADD}}$ be the *order statistics* of $Z^{\mathrm{ADD}}$ (i.e., $Z_i^{\mathrm{ADD}}$ are sorted in increasing order) and $\delta_{[i:N]}^{\mathrm{ADD}}$ be the *concomitant* of $Z_{i:N}^{\mathrm{ADD}}$; i.e., $\delta_i^{\mathrm{ADD}}$ are sorted according to $Z_{i:N}^{\mathrm{ADD}}$, meaning that $\delta_{[i:N]}^{\mathrm{ADD}} = \delta_j$ if $Z_{i:N}^{\mathrm{ADD}} = Z_j^{\mathrm{ADD}}$. The *survival function of detection points* is given by

$$1 - \hat{F}_N^{\mathrm{ADD}}(t) := \hat{S}^{\mathrm{ADD}}(t) = \prod_{i=1}^N (1 - \frac{\delta_{[i:N]}^{\mathrm{ADD}}}{N - i + 1})^{\mathbb{1}(Z_{i:N}^{\mathrm{ADD}} \le t)}, \tag{46}$$

where $t \in \mathbb{R}_{\ge 0}$ This is equivalent to the convention in the main text, which can be confirmed by canceling factors telescopically. In case of ties, treat a death as if it occurs before a censoring at the same time point. Our KM-ARL is formally defined as

$$\hat{M}_T^{(\mathrm{KM})} = \int_0^b t\, d\hat{F}_N^{\mathrm{ADD}}, \tag{47}$$

where $a \in \mathbb{R}_{\ge 0}$ is arbitrary.

**Theorem B.3** (Finite-sample bias bounds for KM-ADD (Thm. 4.2)). *We idealize the time index as continuous without loss of generality, for technical convenience. Consider only sequences with $\nu_i < \infty$ and $\Delta\tau_i \geq 0$ in the dataset. Let $F^{\mathrm{ADD}}, G^{\mathrm{ADD}}$, and $H^{\mathrm{ADD}}$ be the CDFs of $\Delta\tau$, $C^{\mathrm{ADD}}$, and $Z^{\mathrm{ADD}}$, respectively. Assume that (i) $F^{\mathrm{ADD}}$ and $G^{\mathrm{ADD}}$ do not have common discontinuities, (ii) $\Delta\tau$ and $C^{\mathrm{ADD}}$ are independent, known as the independent censoring (or non-informative censoring), and (iii) $H^{\mathrm{ADD}}$ is continuous. Then, we have for any $b \in \mathbb{R}_{\geq 0}$*

$$-\int_0^b t\, G^{\mathrm{ADD}}(t)\, H^{\mathrm{ADD}}(t)^{N-1}\, F^{\mathrm{ADD}}(dt) \tag{48}$$

$$\leq \mathcal{B}_{\mathrm{FS}}(\hat{M}_T^{(\mathrm{KM})}) \leq \tag{49}$$

$$\int_0^b b\, G^{\mathrm{ADD}}(t)\, H^{\mathrm{ADD}}(t)^{N-1}\, F^{\mathrm{ADD}}(dt), \tag{50}$$

Assumption (i) is a technical requirement and is valid unless pathological situations are considered. In most cases, it holds with probability 1 because we consider continuous time. Note that $F^{\mathrm{ADD}}$ and $G^{\mathrm{ADD}}$ may have separate discontinuities. Assumption (ii) is the independent censoring assumption discussed above. Assumption (iii) is also a technical requirement.

*Proof.* Thanks to the setup detailed in this section, the proof is identical to that of Thm. 4.1 with relevant replacements, such as $(\cdot)^{\mathrm{ARL}}$ with $(\cdot)^{\mathrm{ADD}}$. $\square$

## B.3 PROOF OF THM. 4.3

**Theorem B.4** (Truncation bias bounds for KM-ARL (Thm. 4.3)). *Define the truncation bias of the LB-ARL as $\mathcal{B}_{\mathrm{TR}}(\hat{\mu}_T^{(\mathrm{LB})}) := \mu_T^{(\mathrm{LB})} - \mu_\infty$, where $\mu_T^{(\mathrm{LB})} := \mathbb{E}[\tau \mid \nu = \infty, \tau \leq T]$ is the true ARL under random sequence lengths. For $a = T_{\max}^*$, we have $\mathcal{B}_{\mathrm{TR}}(\hat{\mu}_T^{(\mathrm{LB})}) \leq \mathcal{B}_{\mathrm{TR}}(\hat{\mu}_T^{(\mathrm{KM})}) \leq 0$.*

*Proof.* Recall that

$$\mathcal{B}_{\mathrm{TR}}(\hat{\mu}_T^{(\mathrm{KM})}) = \mu_T^{(\mathrm{KM})} - \mu_\infty$$

$$= \int_0^{T_{\max}^*} S^{\mathrm{ARL}}(t)dt - \mathbb{E}[\tau \mid \nu = \infty] \tag{51}$$

$$\mathcal{B}_{\mathrm{TR}}(\hat{\mu}_T^{(\mathrm{LB})}) = \mu_T^{(\mathrm{LB})} - \mu_\infty$$

$$= \mathbb{E}[\tau \mid \nu = \infty, \tau \leq T] - \mathbb{E}[\tau \mid \nu = \infty]. \tag{52}$$

Since

$$\int_0^\infty S^{\mathrm{ARL}}(t)dt \tag{53}$$

$$= \int_0^\infty P(\tau > t \mid \nu = \infty)dt \tag{54}$$

$$= \int_0^\infty \mathbb{E}[\mathbb{1}(\tau > t) \mid \nu = \infty]dt \tag{55}$$

$$= \mathbb{E}\Big[\int_0^\infty \mathbb{1}(\tau > t)dt \mid \nu = \infty\Big] \tag{56}$$

$$= \mathbb{E}[\tau \mid \nu = \infty], \tag{57}$$

we have $\mathcal{B}_{\mathrm{TR}}(\hat{\mu}_T^{(\mathrm{KM})}) = -\int_{T_{\max}^*}^{\infty} S^{\mathrm{ARL}}(t) \leq 0$. Below, we show that $\mathcal{B}_{\mathrm{TR}}(\hat{\mu}_T^{(\mathrm{LB})}) \leq \mathcal{B}_{\mathrm{TR}}(\hat{\mu}_T^{(\mathrm{KM})})$. Since

$$\mu_T^{(\mathrm{KM})}$$

$$= \int_0^{T_{\max}^*} S^{\mathrm{ARL}}(t)dt \tag{58}$$

$$= \int_0^{T_{\max}^*} P(\tau > t \mid \nu = \infty)dt \tag{59}$$

$$= \int_0^{T_{\max}^*} \mathbb{E}[\mathbb{1}(\tau > t) \mid \nu = \infty]dt \tag{60}$$

$$= \mathbb{E}[\int_0^{T_{\max}^*} dt \mid \nu = \infty] \tag{61}$$

$$= P(\tau \leq T_{\max}^* \mid \nu = \infty)$$
$$\times \mathbb{E}[\int_0^{T_{\max}^*} \mathbb{1}(\tau > t)dt \mid \nu = \infty, \tau \leq T_{\max}^*]$$
$$+ P(\tau > T_{\max}^* \mid \nu = \infty)$$
$$\times \mathbb{E}[\int_0^{T_{\max}^*} \mathbb{1}(\tau > t)dt \mid \nu = \infty, \tau > T_{\max}^*] \tag{62}$$

$$= P(\tau \leq T_{\max}^* \mid \nu = \infty)\mathbb{E}[\tau \mid \nu = \infty, \tau \leq T_{\max}^*]$$
$$+ P(\tau > T_{\max}^* \mid \nu = \infty)\mathbb{E}[T_{\max}^* \mid \nu = \infty, \tau > T_{\max}^*] \tag{63}$$

$$= P(\tau \leq T_{\max}^* \mid \nu = \infty)\mathbb{E}[\tau \mid \nu = \infty, \tau \leq T_{\max}^*]$$
$$+ P(\tau > T_{\max}^* \mid \nu = \infty)T_{\max}^* \tag{64}$$

$$= (1 - P(\tau > T_{\max}^* \mid \nu = \infty))\mathbb{E}[\tau \mid \nu = \infty, \tau \leq T_{\max}^*]$$
$$+ P(\tau > T_{\max}^* \mid \nu = \infty)T_{\max}^* \tag{65}$$

$$= \mathbb{E}[\tau \mid \nu = \infty, \tau \leq T_{\max}^*]$$
$$+ P(\tau > T_{\max}^* \mid \nu = \infty)$$
$$\times (T_{\max}^* - \mathbb{E}[\tau \mid \nu = \infty, \tau \leq T_{\max}^*]) \tag{66}$$

we have

$$\mathcal{B}_{\mathrm{TR}}(\hat{\mu}_T^{(\mathrm{KM})}) - \mathcal{B}_{\mathrm{TR}}(\hat{\mu}_T^{(\mathrm{LB})})$$

$$= \mu_T^{(\mathrm{KM})} - \mu_T^{(\mathrm{LB})} \tag{67}$$

$$= \mathbb{E}[\tau \mid \nu = \infty, \tau \leq T_{\max}^*]$$
$$+ P(\tau > T_{\max}^* \mid \nu = \infty)$$
$$\times (T_{\max}^* - \mathbb{E}[\tau \mid \nu = \infty, \tau \leq T_{\max}^*])$$
$$- \mathbb{E}[\tau \mid \nu = \infty, \tau \leq T] \tag{68}$$

$$= \mathbb{E}[\tau \mid \nu = \infty, \tau \leq T_{\max}^*] - \mathbb{E}[\tau \mid \nu = \infty, \tau \leq T]$$
$$+ P(\tau > T_{\max}^* \mid \nu = \infty)$$
$$\times (T_{\max}^* - \mathbb{E}[\tau \mid \nu = \infty, \tau \leq T_{\max}^*]). \tag{69}$$

Finally, we can see that $\mathbb{E}[\tau \mid \nu = \infty, \tau \leq T_{\max}^*] - \mathbb{E}[\tau \mid \nu = \infty, \tau \leq T] \geq 0$ (Lem. B.5), and $T_{\max}^* - \mathbb{E}[\tau \mid \nu = \infty, \tau \leq T_{\max}^*] \geq 0$; hence, $\mathcal{B}_{\mathrm{TR}}(\hat{\mu}_T^{(\mathrm{KM})}) - \mathcal{B}_{\mathrm{TR}}(\hat{\mu}_T^{(\mathrm{LB})}) \geq 0$. □

**Lemma B.5.** $\mathbb{E}[\tau \mid \nu = \infty, \tau \leq T_{\max}^*] - \mathbb{E}[\tau \mid \nu = \infty, \tau \leq T] \geq 0.$

*Proof.* Without loss of generality, this inequality is equivalent to $\mathbb{E}[\tau \mid 0 \leq \tau \leq T_{\max}^*] - \mathbb{E}[\tau \mid 0 \leq \tau \leq T] \geq 0$, where $T \leq T_{\max}^*(\in \mathbb{R})$ holds a.s., and $\tau$ and $T$ are independent. For notational simplicity, we will show that

$$\mathbb{E}_{X,Y}[X|X \leq y_{\max}] - \mathbb{E}_{X,Y}[X|X \leq Y] \geq 0, \tag{70}$$

where $X$ is a non-negative random variable, $Y$ is a non-negative random variable with $0 \leq Y \leq y_{\max}$ a.s., $y_{\max}(\geq 0)$ is a real constant, and $X$ and $Y$ are independent.

Write $F_X$ for the CDF of $X$, and let

$$g(x) := P(Y \geq x), \quad 0 \leq x \leq y_{\max}. \tag{71}$$

Because $Y \in [0, y_{\max}]$, $g$ is non-increasing in $x$, while $g(x) = 0$ for $x > y_{\max}$. Thus, we have

$$P(X \leq Y) = \mathbb{E}\left[\mathbb{1}(X \leq Y)\right] = \int_0^\infty P(Y \geq x)dF_X(x) = \int_0^{y_{\max}} g(x)dF_X(x), \tag{72}$$

where we used the independence of $X$ and $Y$. Additionally, we have

$$\mathbb{E}\left[X\mathbb{1}(X \leq Y)\right] = \int_0^\infty xP(Y \geq x)dF_X(x) = \int_0^{y_{\max}} xg(x)dF_X(x). \tag{73}$$

Therefore,

$$\mathbb{E}[X \mid X \leq Y] = \frac{\mathbb{E}\left[X\mathbb{1}(X \leq Y)\right]}{P(X \leq Y)} = \frac{\int_0^{y_{\max}} xg(x)dF_X(x)}{\int_0^{y_{\max}} g(x)dF_X(x)}. \tag{74}$$

On the other hand, we can similarly derive

$$\mathbb{E}\left[X \mid X \leq y_{\max}\right] = \frac{\int_0^{y_{\max}} xdF_X(x)}{\int_0^{y_{\max}} dF_X(x)}. \tag{75}$$

Next, let $\nu$ be the finite measure on $[0, y_{\max}]$ given by $d\nu(x) = dF_X(x)$, and define another measure $\mu$ by

$$d\mu(x) = g(x)d\nu(x), \quad 0 \leq x \leq y_{\max}. \tag{76}$$

Note that $g(x)$ is non-increasing in $x$ and that $x$ is increasing in $x$. Thus, Eqs. (74) and (75) can be rewritten as

$$\mathbb{E}[X \mid X \leq Y] = \frac{\int xd\mu(x)}{\int d\mu(x)}, \quad \mathbb{E}[X \mid X \leq y_{\max}] = \frac{\int xd\nu(x)}{\int d\nu(x)}. \tag{77}$$

Consider the covariance under the normalized measure proportional to $\nu$:

$$\mathrm{Cov}_\nu(x, g(x)) = \mathbb{E}_\nu[Xg(X)] - \mathbb{E}_\nu[X]\,\mathbb{E}_\nu[g(X)] \leq 0, \tag{78}$$

because $x$ is increasing and $g(x)$ is decreasing (Lem. B.6). Therefore,

$$\left(\frac{\int xg(x)d\nu(x)}{\int d\nu(x)}\right) \leq \left(\frac{\int xd\nu(x)}{\int d\nu(x)}\right)\left(\frac{\int g(x)d\nu(x)}{\int d\nu(x)}\right). \tag{79}$$

This is equivalent to

$$\frac{\int xg(x)d\nu(x)}{\int g(x)d\nu(x)} \leq \frac{\int xd\nu(x)}{\int d\nu(x)}. \tag{80}$$

In other words,

$$\mathbb{E}[X \mid X \leq Y] \leq \mathbb{E}\left[X \mid X \leq y_{\max}\right]. \tag{81}$$

This concludes the proof. $\qquad\square$

**Lemma B.6.** *Let $X$ be a real random variable, $f$ an increasing functions, and $h$ and non-increasing function. Then, it holds that $\mathrm{Cov}(f(X), h(X)) \leq 0$.*

*Proof.* Take two i.i.d. copies, $X$ and $X'$, with the same distribution. Let us consider the random quantity

$$\left(f(X) - f(X')\right)\left(h(X) - h(X')\right). \tag{82}$$

Expanding its expectation, we have

$$\mathbb{E}\left[\left(f(X) - f(X')\right)\left(h(X) - h(X')\right)\right] = \mathbb{E}[f(X)h(X)] + \mathbb{E}\left[f(X')h(X')\right]$$
$$- \mathbb{E}\left[f(X)h(X')\right] - \mathbb{E}\left[f(X')h(X)\right] \tag{83}$$
$$= 2\mathbb{E}[f(X)h(X)] - 2\mathbb{E}[f(X)]\mathbb{E}[h(X)] \tag{84}$$
$$= 2\mathrm{Cov}(f(X), h(X)). \tag{85}$$

Thus, we have the identity

$$\mathrm{Cov}(f(X), h(X)) = \frac{1}{2}\mathbb{E}\left[\left(f(X) - f(X')\right)\left(h(X) - h(X')\right)\right]. \tag{86}$$

Now use the fact that: $f$ is increasing, and $h$ is non-increasing. Fix any $x, y$ in the support. Then:

- If $x > y$, then $f(x) > f(y)$ and $h(x) \le h(y)$, so $(f(x) - f(y)) > 0$ and $(h(x) - h(y)) \le 0$, hence $(f(x) - f(y))(h(x) - h(y)) \le 0$.

- If $x < y$, then $f(x) < f(y)$ and $h(x) \ge h(y)$, so $(f(x) - f(y)) < 0$ and $(h(x) - h(y)) \ge 0$, again $(f(x) - f(y))(h(x) - h(y)) \le 0$.

- If $x = y$, the product is 0.

Thus, we have

$$(f(x) - f(y))(h(x) - h(y)) \le 0 \text{ for all } x, y. \tag{87}$$

Therefore, when $X$ and $X'$ are i.i.d.,

$$(f(X) - f(X'))(h(X) - h(X')) \le 0 \text{ a.s.} \tag{88}$$

Taking expectations,

$$\mathbb{E}\left[(f(X) - f(X'))(h(X) - h(X'))\right] \le 0. \tag{89}$$

Plug this back into Eq. (86), we have

$$\mathrm{Cov}(f(X), h(X)) = \frac{1}{2}\mathbb{E}\left[(f(X) - f(X'))(h(X) - h(X'))\right] \le 0. \tag{90}$$

$\square$

### B.4  PROOF OF THEOREM 4.4

**Theorem B.7** (Truncation bias bound for KM-ADD). *Define the truncation bias of the LB-ADD as $\mathcal{B}_{\mathrm{TR}}(\hat{M}_T^{(\mathrm{LB})}) := M_T^{(\mathrm{LB})} - M_\infty$, where $M_T^{(\mathrm{LB})} := \mathbb{E}[\Delta\tau \mid \nu < \infty, 0 \le \Delta\tau \le \Delta T]$ is the true ADD under random sequence lengths. For $b = \Delta T_{\max}^*$, we have $\mathcal{B}_{\mathrm{TR}}(\hat{M}_T^{(\mathrm{LB})}) \le \mathcal{B}_{\mathrm{TR}}(\hat{M}_T^{(\mathrm{KM})}) \le 0$, under the independent censoring assumption in Thm. 4.2.*

*Proof.* Recall that

$$\mathcal{B}_{\mathrm{TR}}(\hat{M}_T^{(\mathrm{KM})}) = M_T^{(\mathrm{KM})} - M_\infty$$
$$= \int_0^{\Delta T_{\max}^*} S^{\mathrm{ADD}}(t)dt - \mathbb{E}[\Delta\tau \mid \nu < \infty, \Delta\tau \ge 0] \tag{91}$$

$$\mathcal{B}_{\mathrm{TR}}(\hat{M}_T^{(\mathrm{LB})}) = M_T^{(\mathrm{LB})} - M_\infty$$
$$= \mathbb{E}[\Delta\tau \mid \nu < \infty, 0 \le \Delta\tau \le \Delta T] - \mathbb{E}[\Delta\tau \mid \nu < \infty, \Delta\tau \ge 0]. \tag{92}$$

First, since

$$\int_0^\infty S^{\mathrm{ADD}}(t)dt \tag{93}$$

$$= \int_0^\infty P(\Delta\tau > t \mid \nu < \infty, \Delta\tau \ge 0)dt \tag{94}$$

$$= \int_0^\infty \mathbb{E}[\mathbb{1}(\Delta\tau > t) \mid \nu < \infty, \Delta\tau \ge 0]dt \tag{95}$$

$$= \mathbb{E}[\int_0^\infty \mathbb{1}(\Delta\tau > t)dt \mid \nu < \infty, \Delta\tau \ge 0] \tag{96}$$

$$= \mathbb{E}[\Delta\tau \mid \nu < \infty, \Delta\tau \ge 0], \tag{97}$$

we have $\mathcal{B}_{\mathrm{TR}}(\hat{M}_T^{(\mathrm{KM})}) = -\int_{\Delta T_{\max}^*}^\infty S^{\mathrm{ADD}}(t) \le 0$.

Second, we will show that $\mathcal{B}_{\mathrm{TR}}(\hat{M}_T^{(\mathrm{LB})}) \leq \mathcal{B}_{\mathrm{TR}}(\hat{M}_T^{(\mathrm{KM})})$. Since

$$M_T^{(\mathrm{KM})}$$

$$= \int_0^{\Delta T_{\max}^*} S^{\mathrm{ADD}}(t)dt \tag{98}$$

$$= \int_0^{\Delta T_{\max}^*} P(\Delta\tau > t \mid \nu < \infty, \Delta\tau \geq 0)dt \tag{99}$$

$$= \int_0^{\Delta T_{\max}^*} \mathbb{E}[\mathbb{1}(\Delta\tau > t) \mid \nu < \infty, \Delta\tau \geq 0]dt \tag{100}$$

$$= \mathbb{E}[\int_0^{\Delta T_{\max}^*} \mathbb{1}(\Delta\tau > t)dt \mid \nu < \infty, \Delta\tau \geq 0] \tag{101}$$

$$= P(\Delta\tau \leq \Delta T_{\max}^* \mid \nu < \infty, \Delta\tau \geq 0)$$
$$\times \mathbb{E}[\int_0^{\Delta T_{\max}^*} \mathbb{1}(\Delta\tau > t)dt \mid \nu < \infty, 0 \leq \Delta\tau \leq \Delta T_{\max}^*]$$
$$+ P(\Delta\tau > \Delta T_{\max}^* \mid \nu < \infty, \Delta\tau \geq 0)$$
$$\times \mathbb{E}[\int_0^{\Delta T_{\max}^*} \mathbb{1}(\Delta\tau > t)dt \mid \nu < \infty, 0 \leq \Delta\tau > \Delta T_{\max}^*] \tag{102}$$

$$= P(\Delta\tau \leq \Delta T_{\max}^* \mid \nu < \infty, \Delta\tau \geq 0)$$
$$\times \mathbb{E}[\Delta\tau \mid \nu < \infty, \Delta 0 \leq \tau \leq \Delta T_{\max}^*]$$
$$+ P(\Delta\tau > \Delta T_{\max}^* \mid \nu < \infty, \Delta\tau \geq 0)$$
$$\times \mathbb{E}[\Delta T_{\max}^* \mid \nu < \infty, 0 \leq \Delta\tau > \Delta T_{\max}^*] \tag{103}$$

$$= P(\Delta\tau \leq \Delta T_{\max}^* \mid \nu < \infty, \Delta\tau \geq 0)$$
$$\times \mathbb{E}[\Delta\tau \mid \nu < \infty, 0 \leq \Delta\tau \leq \Delta T_{\max}^*]$$
$$+ P(\Delta\tau > \Delta T_{\max}^* \mid \nu < \infty)\Delta T_{\max}^* \tag{104}$$

$$= (1 - P(\Delta\tau > \Delta T_{\max}^* \mid \nu < \infty, \Delta\tau \geq 0))$$
$$\times \mathbb{E}[\Delta\tau \mid \nu < \infty, 0 \leq \Delta\tau \leq \Delta T_{\max}^*]$$
$$+ P(\Delta\tau > \Delta T_{\max}^* \mid \nu < \infty, \Delta\tau \geq 0)\Delta T_{\max}^* \tag{105}$$

$$= \mathbb{E}[\Delta\tau \mid \nu < \infty, 0 \leq \Delta\tau \leq \Delta T_{\max}^*]$$
$$+ P(\Delta\tau > \Delta T_{\max}^* \mid \nu < \infty, \Delta\tau \geq 0)$$
$$\times (\Delta T_{\max}^* - \mathbb{E}[\Delta\tau \mid \nu < \infty, 0 \leq \Delta\tau \leq \Delta T_{\max}^*]) \tag{106}$$

we have

$$\mathcal{B}_{\mathrm{TR}}(\hat{M}_T^{(\mathrm{KM})}) - \mathcal{B}_{\mathrm{TR}}(\hat{M}_T^{(\mathrm{LB})})$$

$$= M_T^{(\mathrm{KM})} - M_T^{(\mathrm{LB})} \tag{107}$$

$$= \mathbb{E}[\Delta\tau \mid \nu < \infty, 0 \leq \Delta\tau \leq \Delta T_{\max}^*]$$
$$+ P(\Delta\tau > \Delta T_{\max}^* \mid \nu < \infty, \Delta\tau \geq 0)$$
$$\times (\Delta T_{\max}^* - \mathbb{E}[\Delta\tau \mid \nu < \infty, 0 \leq \Delta\tau \leq \Delta T_{\max}^*])$$
$$- \mathbb{E}[\Delta\tau \mid \nu < \infty, 0 \leq \Delta\tau \leq \Delta T] \tag{108}$$

$$= \mathbb{E}[\Delta\tau \mid \nu < \infty, 0 \leq \Delta\tau \leq \Delta T_{\max}^*]$$
$$- \mathbb{E}[\Delta\tau \mid \nu < \infty, 0 \leq \Delta\tau \leq \Delta T]$$
$$+ P(\Delta\tau > \Delta T_{\max}^* \mid \nu < \infty, \Delta\tau \geq 0)$$
$$\times (\Delta T_{\max}^* - \mathbb{E}[\Delta\tau \mid \nu < \infty, 0 \leq \Delta\tau \leq \Delta T_{\max}^*]). \tag{109}$$

Finally, we can see that $\mathbb{E}[\Delta\tau \mid \nu < \infty, 0 \leq \Delta\tau \leq \Delta T_{\max}^*] - \mathbb{E}[\Delta\tau \mid \nu < \infty, 0 \leq \Delta\tau \leq \Delta T] \geq 0$ (Lem. B.8), and $\Delta T_{\max}^* - \mathbb{E}[\Delta\tau \mid \nu < \infty, 0 \leq \Delta\tau \leq \Delta T_{\max}^*] \geq 0$; hence, $\mathcal{B}_{\mathrm{TR}}(\hat{M}_T^{(\mathrm{KM})}) - \mathcal{B}_{\mathrm{TR}}(\hat{M}_T^{(\mathrm{LB})}) \geq 0$.

$$\square$$

**Lemma B.8.** $\mathbb{E}[\Delta\tau \mid \nu < \infty, 0 \leq \Delta\tau \leq \Delta T^*_{\max}] - \mathbb{E}[\Delta\tau \mid \nu < \infty, 0 \leq \Delta\tau \leq \Delta T] \geq 0$.

*Proof.* Without loss of generality, this inequality is equivalent to $\mathbb{E}[\Delta\tau \mid 0 \leq \Delta\tau \leq \Delta T^*_{\max}] - \mathbb{E}[\Delta\tau \mid 0 \leq \Delta\tau \leq \Delta T] \geq 0$, where $\Delta T \leq \Delta T^*_{\max}(\in \mathbb{R})$ holds a.s. $\Delta\tau$ and $\Delta T(= C^{\mathrm{ADD}})$ are independent because of the independent censoring assumption. This inequality can be rewritten as

$$\mathbb{E}_{X,Y}[X|X \leq y_{\max}] - \mathbb{E}_{X,Y}[X|X \leq Y] \geq 0, \tag{110}$$

where $X$ is a non-negative random variable, $Y$ is a non-negative random variable with $0 \leq Y \leq y_{\max}$ a.s., $y_{\max}(\geq 0)$ is a real constant, and $X$ and $Y$ are independent. Because Ineq. (110) coincides with Ineq. (70), the proof follows the same steps as Lem. B.5. $\qquad\square$

## C SUPPLEMENTARY EXPERIMENTAL RESULTS

### C.1 EXAMPLE OF SURVIVAL CURVE OF DETECTION POINTS

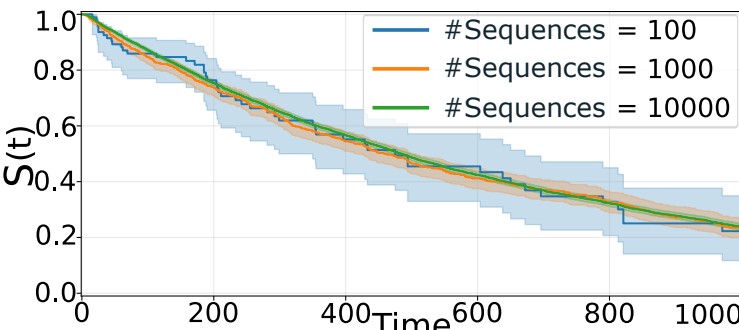

Figure 6: **Estimated survival functions of detection points.** The shaded area represents the standard error of the mean.

Survival functions of detection points are shown in Fig. 6. The experiment uses the Gaussian process dataset at three dataset sizes. The QCD model used is the GSR procedure with ground-truth statistics, evaluated under a given threshold. Changepoint locations are sampled from a geometric distribution with the success probability of $p = 0.001$. Error bars represent the standard error of the mean.

### C.2 COMPLETE ABLATION OF FIG. 4

Fig. 7 presents the complete ablation study corresponding to Fig. 4, demonstrating that KM-ARL and KM-ADD yield more accurate estimates of the true ARL-ADD curve, even under limited and irregular sequence lengths. The gray areas indicate regions of extrapolation (excluding the true ARL), where ARLs cannot be estimated due to the absence of data, unless additional assumptions are imposed on the underlying distribution.

### C.3 GAUSSIAN PROCESS DATASET

We provide additional experimental results on the Gaussian process dataset. Fig. 8 is an example sequence. Fig. 9 shows temporal statistics of a sequence in our Gaussian process dataset. Fig. 10 presents the ARL-ADD tradeoff curves of the QCD models evaluated on a Gaussian process dataset, while we use the WISDM Actitracker dataset in Fig. 1 in the main text. It demonstrates that our KM-ARL and KM-ADD are more robust to irregular lengths than the conventional LB-ARL and LB-ADD. Note that some curves in Fig. 10(a) are non-monotonic because the sequences used to compute the LB-ARL and LB-ADD differ drastically across different thresholds, causing unstable estimates. For example, short sequences are excluded from the computation of the LB metrics when the threshold is high. This issue does not arise for our KME-based metrics, contributing to their robustness to finite and irregular sequence lengths.

### C.4 POISSON PROCESS DATASET

We provide experimental results on the Poisson process dataset. The pre-change and post-change Poisson processes have the mean of 1 and 4, respectively. We use two types of changepoint distributions: geometric and uniform. Fig. 8 is an example sequence from our Poisson process dataset. Fig. 12 shows statistics of a sequence in our Poisson process dataset. Fig. 13 presents the ARL-ADD curve evaluated on the Poisson process dataset, demonstrating that the KM-ARL and KM-ADD provide more accurate estimates of the true ARL-ADD curve, even when sequence lengths are limited and irregular.

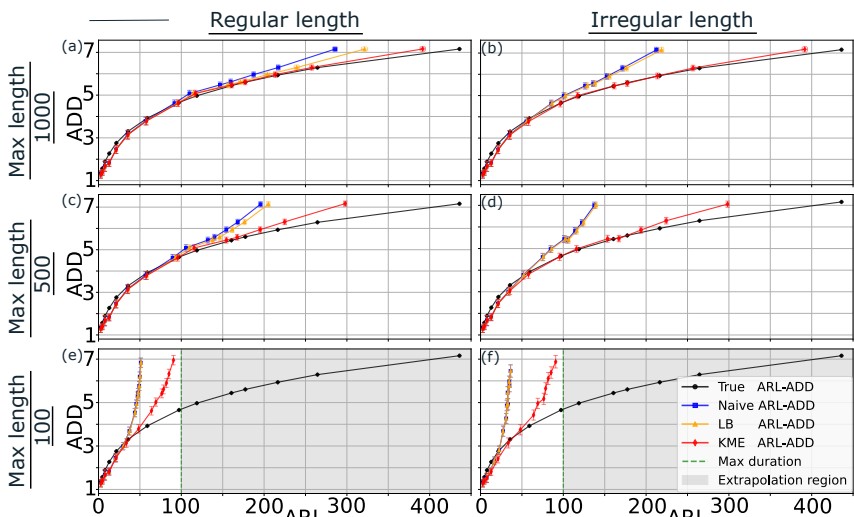

Figure 7: **Complete ablation study of Fig. 4.** The KM-ARL and KM-ADD provide more accurate estimates of the true ARL-ADD curve, even when sequence lengths are limited and irregular. The experiments use the Gaussian process dataset, comprising 10000 sequences. The employed QCD algorithm is the GSR procedure using ground-truth statistics. Changepoint locations are sampled uniformly. 50% of sequences contain a changepoint. Error bars represent the standard error of the mean. The gray areas indicate regions of extrapolation (excluding the true ARL), where ARLs cannot be estimated due to the absence of data, unless additional assumptions are imposed on the underlying distribution. **(a) Sequence length is** 1000. **(b) Sequence lengths vary irregularly in the range** [100, 1000]. **(c) Sequence length is** 500. **(d) Sequence lengths vary irregularly in the range** [50, 500]. **(e) Sequence length is** 100. **(f) Sequence lengths vary irregularly in the range** [10, 100].

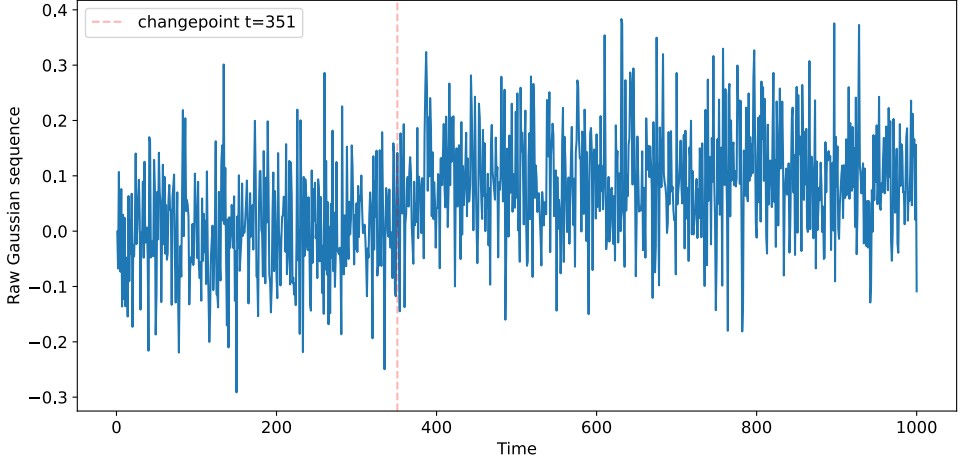

Figure 8: **Example sequence in Gaussian process dataset.**

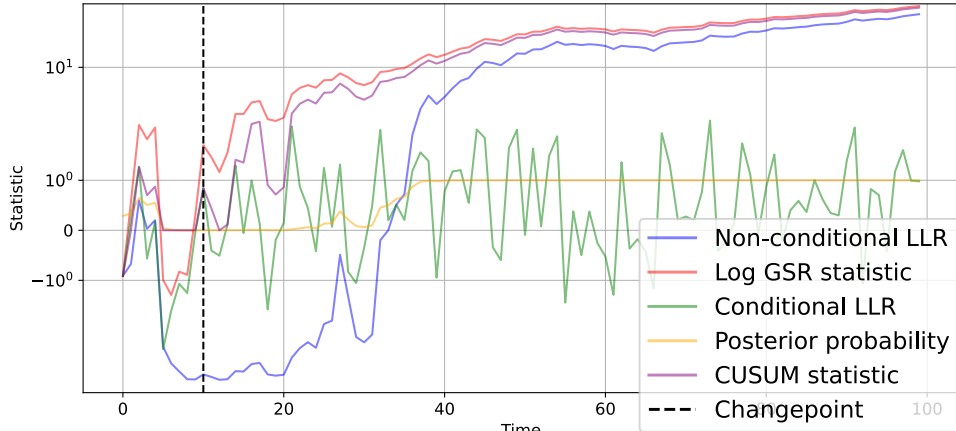

Figure 9: **Example statistics in Gaussian process dataset.** The observation interval is $\Delta t = 1$. LLR is short for log-likelihood ratio $\log(f(X^{(0,t)})/g(X^{(0,t)}))$, where $f$ and $g$ are the pre- and post-change density, respectively. Non-conditional LLR is given by $\log(f(X^{(0,t)})/g(X^{(0,t)}))$, while Conditional LLR is given by $\log(f(X^{(t)} \mid X^{(0,t-1)}))/g(X^{(t)} \mid X^{(0,t-1)}))$. Posterior probability means $g(X^{(0,t)})$.

### C.5 WISDM ACTITRACKER DATASET

We provide additional experimental results on the WISDM Actitracker dataset. Fig. 14 is the original figure of Fig. 1.

While we use the machine-labeled subset in the main text, Fig. 15 presents the ARL-ADD curves of QCD models evaluated on the user-labeled subset ("labeled" subset) of the WISDM Actitracker dataset, which is about $10^3 (= 51326/83)$ times smaller than the machine-labeled subset ("unlabeled" subset) of the WISDM Actitracker dataset (see Tab. 2 for statistics). Evaluation is challenging on this subset because the number of sequences is limited to only 83. For such small datasets, we recommend using min-max-based metrics, such as the box plot, rather than average-based metrics, such as the ARL and ADD.

### C.6 ARL-ADD WITH CUSUM

While we use the GSR procedure in the main text, we provide additional experimental results obtained with the CUSUM procedure (Page, 1954) with ground-truth statistics. The ARL-ADD curves are provided in Fig. 16 and support our findings in the main text. The CUSUM procedure is evaluated under various thresholds.

### C.7 ARL-ADD WITH GEOMETRIC CHANGEPOINT DISTRIBUTION

We provide additional experimental results obtained with the geometric changepoint distribution, instead of the uniform distribution. We evaluate the ARL-ADD tradeoff curve on a Gaussian process dataset. The results are provided in Fig. 17 and support our findings in the main text.

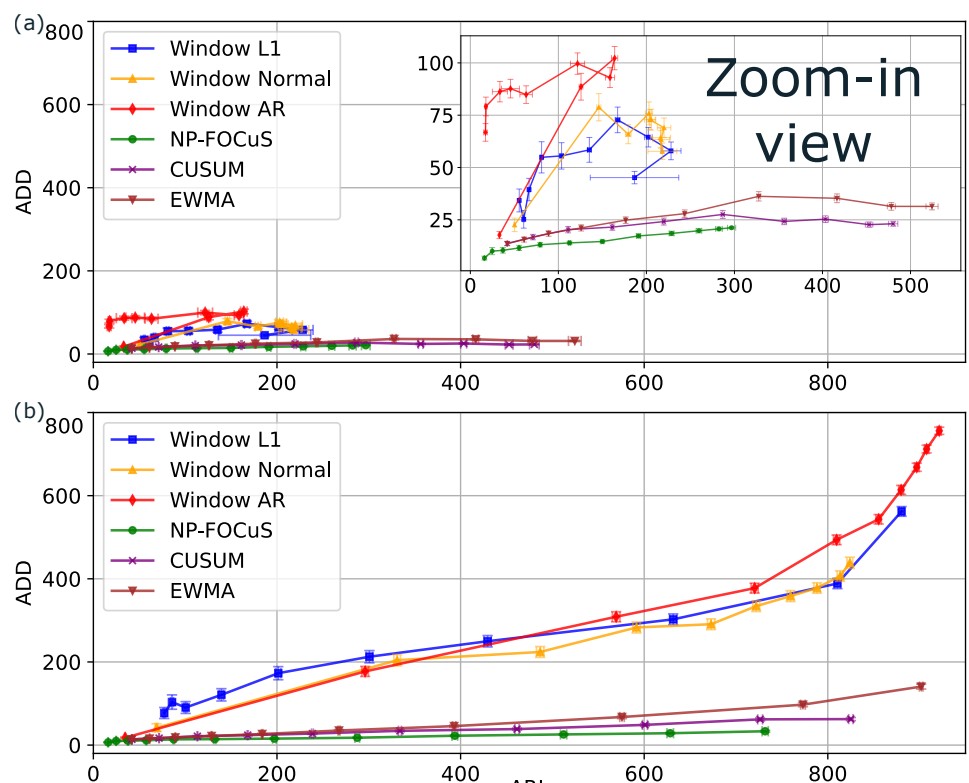

Figure 10: **Evaluation of QCD models on Gaussian process dataset. (a) LB-ARL and LB-ADD.** **(b) KM-ARL and KM-ADD.** Our KM-ARL and KM-ADD are more robust to irregular lengths than the conventional LB-ARL and LB-ADD. The experiments use the Gaussian process dataset, comprising $10000$ sequences. Changepoint locations are sampled uniformly. $50\%$ of the sequences contain a changepoint. Sequence lengths vary irregularly in the range $[100, 1000]$. Note that some curves in (a) are non-monotonic because the sequences used to compute the LB-ARL and LB-ADD differ drastically across different thresholds, causing unstable estimates. For example, short sequences are excluded from the computation of the LB metrics when the threshold is high. This issue does not arise for our KME-based metrics, contributing to their robustness to finite and irregular sequence lengths.

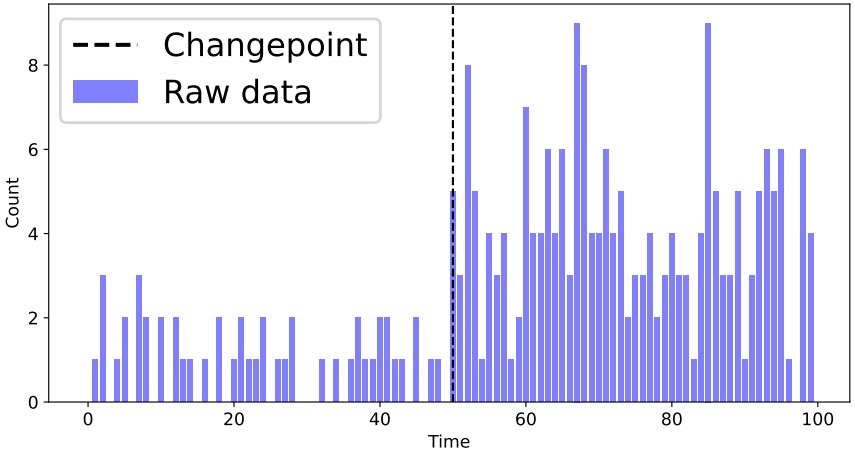

Figure 11: **Example sequence in Poisson process dataset.**

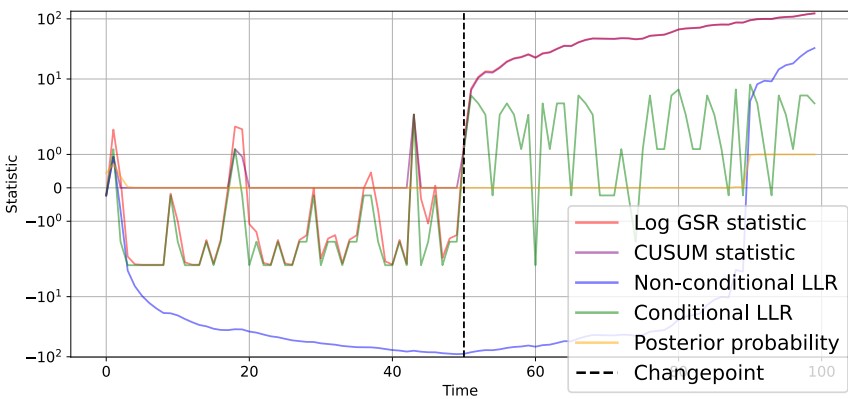

Figure 12: **Example statistics in Gaussian process dataset.** Observation interval is $\Delta t = 1$. LLR is short for log-likelihood ratio $\log(f(X^{(0,t)})/g(X^{(0,t)}))$, where $f$ and $g$ are the pre- and post-change density, respectively. Non-conditional LLR is given by $\log(f(X^{(0,t)})/g(X^{(0,t)}))$, while Conditional LLR is given by $\log(f(X^{(t)} \mid X^{(0,t-1)}))/g(X^{(t)} \mid X^{(0,t-1)}))$. Posterior probability means $g(X^{(0,t)})$.

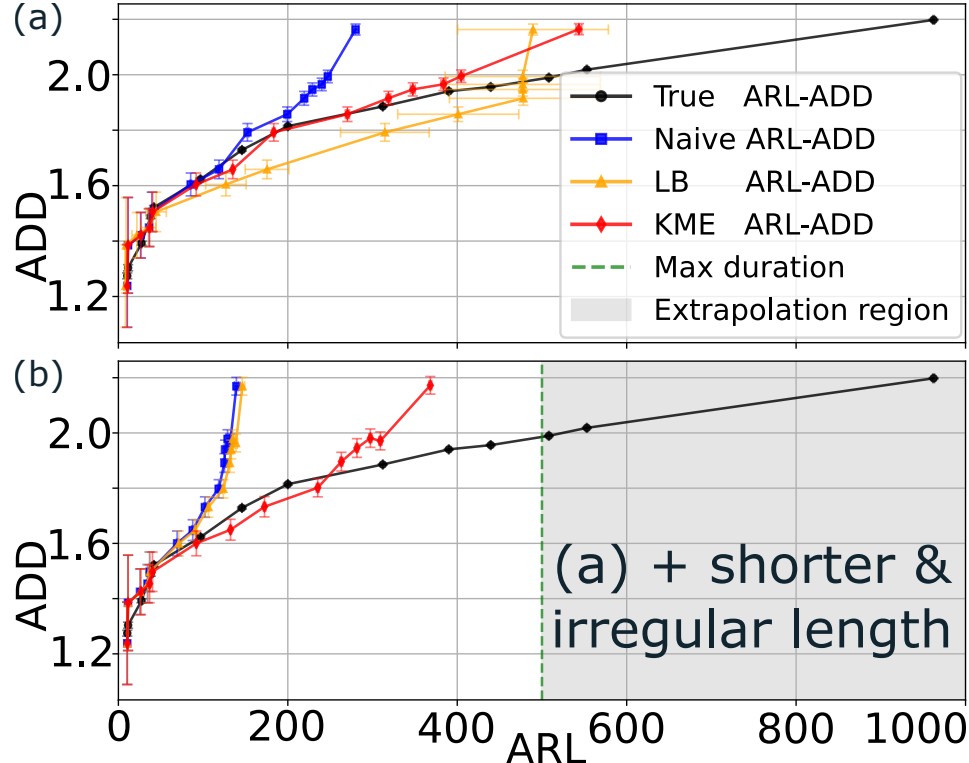

Figure 13: **ARL-ADD tradeoff curves on Poisson process dataset.** The KM-ARL and KM-ADD provide more accurate estimates of the true ARL-ADD curve, even when sequence lengths are limited and irregular. **(a) Sequence length is** $1000$. **(b) Sequence lengths vary irregularly in the range** $[50, 500]$. The experiments use the Gaussian process dataset, comprising 10000 sequences. The employed QCD algorithm is the GSR procedure using ground-truth statistics. Changepoint locations are sampled uniformly. $50\%$ of sequences contain a changepoint. Error bars represent the standard error of the mean. In (b), $\mathrm{ARL} > 500$ (gray area) indicates a region of extrapolation (excluding the true ARL), where ARLs cannot be estimated due to the absence of data, unless additional assumptions are imposed on the underlying distribution.

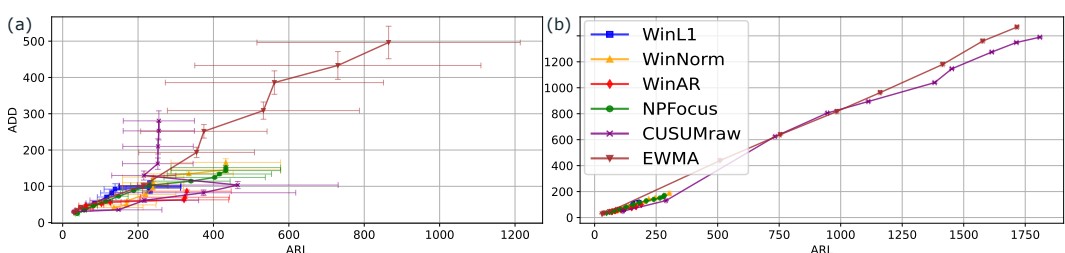

Figure 14: **Original figure of Fig. 1.** Our KM-ARL and KM-ADD are more interpretable than the conventional LB-ARL and LB-ADD.

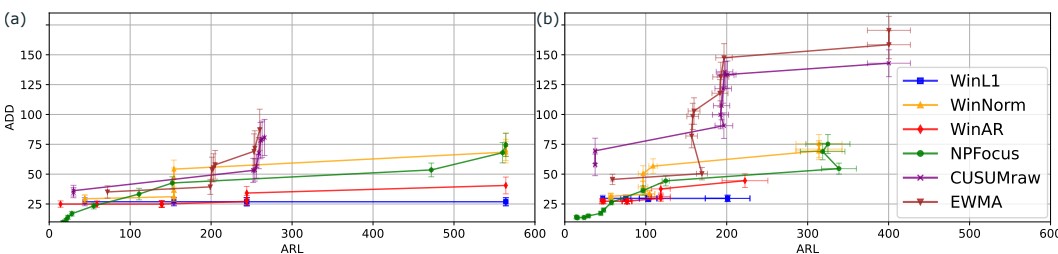

Figure 15: **Evaluation of QCD models on real-world dataset (user-labeled subset of WISDM Actitracker).** The user-labeled subset ("labeled" subset) of the WISDM Actitracker is used, which is about $10^3 (= 51326/83)$ times smaller than the machine-labeled subset ("unlabeled" subset) of the WISDM Actitracker dataset (see Tab. 2 for statistics). Evaluation is challenging on this subset because the number of sequences is limited to only 83. For such small datasets, we recommend using min-max-based metrics, such as the box plot, rather than average-based metrics, such as the ARL and ADD.

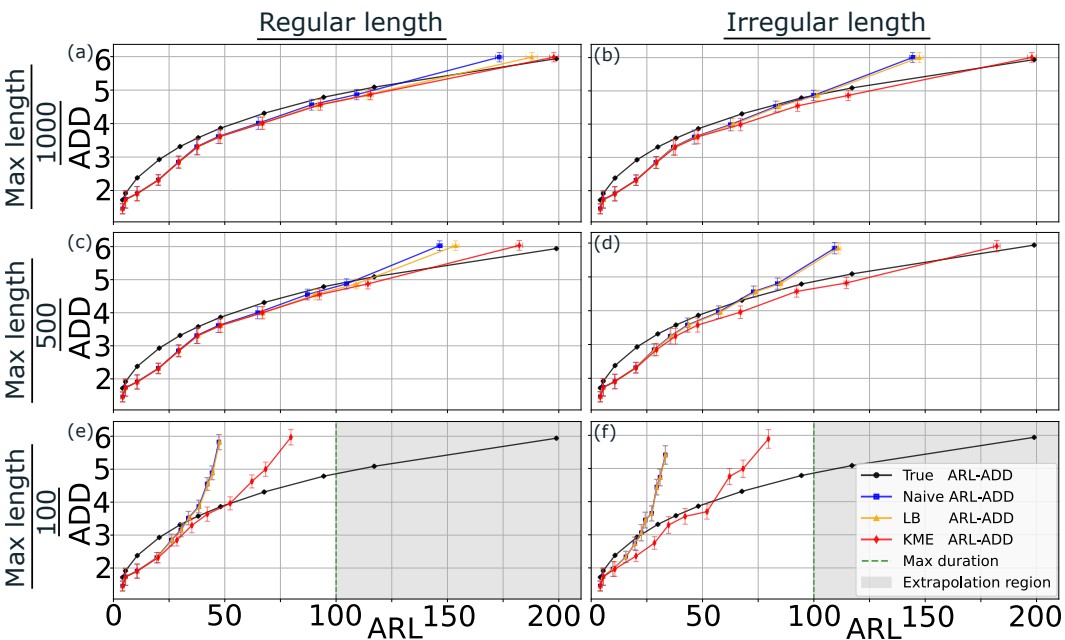

Figure 16: **ARL-ADD curve of CUSUM procedure with ground-truth statistics.** The results closely parallel those obtained with the GSR procedure in the main text: The KM-ARL and KM-ADD provide more accurate estimates of the true ARL-ADD curve, even when sequence lengths are limited and irregular. CUSUM is evaluated under various thresholds. The experiments use the Gaussian process dataset, comprising 10000 sequences. Changepoint locations are sampled uniformly. 50% of sequences contain a changepoint. Error bars represent the standard error of the mean. The gray areas indicate regions of extrapolation (excluding the true ARL), where ARLs cannot be estimated due to the absence of data, unless additional assumptions are imposed on the underlying distribution. **(a) Sequence length is** $1000$**. (b) Sequence lengths vary irregularly in the range** $[100, 1000]$**. (c) Sequence length is** $500$**. (d) Sequence lengths vary irregularly in the range** $[50, 500]$**. (e) Sequence length is** $100$**. (f) Sequence lengths vary irregularly in the range** $[10, 100]$**.**

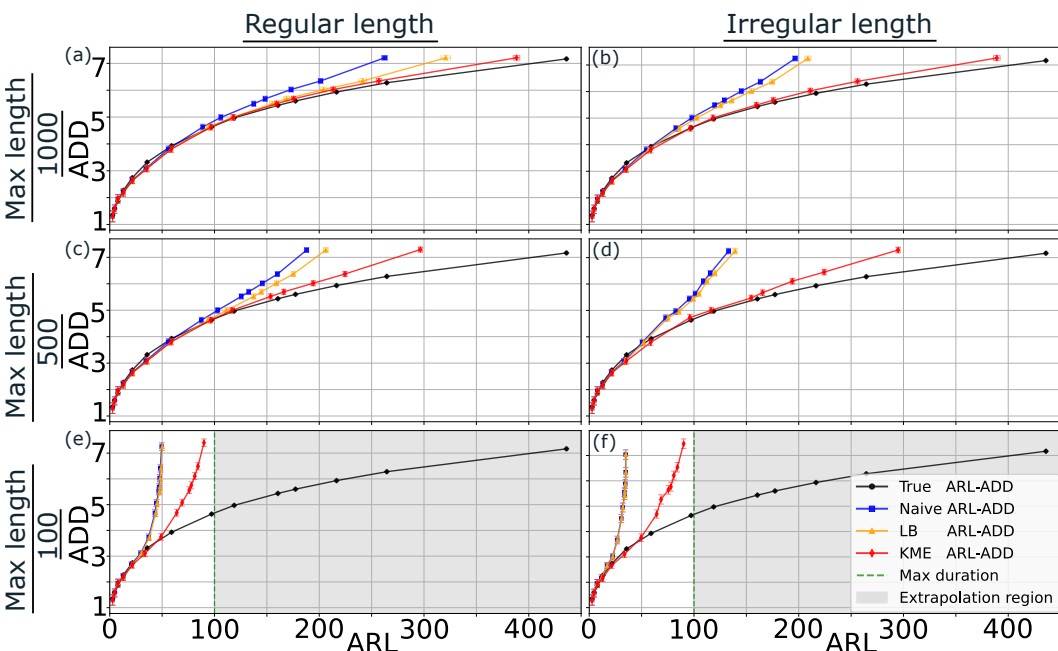

Figure 17: **ARL-ADD curve on Gaussian process dataset with geometric changepoint distribution.** The results closely parallel those obtained with the uniform changepoint distribution in the main text: The KM-ARL and KM-ADD provide more accurate estimates of the true ARL-ADD curve, even when sequence lengths are limited and irregular. The experiments use the Gaussian process dataset, comprising 10000 sequences. The employed QCD algorithm is the GSR procedure using ground-truth statistics. Changepoint locations are sampled from a geometric distribution with the success probability $p = 0.001$. Error bars represent the standard error of the mean. The gray areas indicate regions of extrapolation (excluding the true ARL), where ARLs cannot be estimated due to the absence of data, unless additional assumptions are imposed on the underlying distribution. **(a) Sequence length is** 1000**. (b) Sequence lengths vary irregularly in the range** $[100, 1000]$**. (c) Sequence length is** 500**. (d) Sequence lengths vary irregularly in the range** $[50, 500]$**. (e) Sequence length is** 100**. (f) Sequence lengths vary irregularly in the range** $[10, 100]$**.**

## D    DETAILED EXPERIMENTAL SETTINGS

**Runtime.**    The computational costs of metric evaluation, including KM-ARL and KM-ADD, are negligible in our experiments compared with the time required to run the QCD algorithms. Computing a single data point in the figures takes from a few seconds to several hours, depending on the sequence length, the number of sequences, and the QCD algorithm used.

**Computing Infrastructure.**    We use Python 3.11.9 (Van Rossum & Drake, 2009), Numpy 1.26.4 (Harris et al., 2020), lifelines 0.30.0 (Davidson-Pilon, 2019), changepoint-online 1.2.1 (Romano et al., 2024b), ruptures 1.1.9 (Truong et al., 2020), ocpdet 0.0.6 (Khamesi, 2022) (we pick up relevant implementations only (`ocpdet/CUSUM.py` and `ocpdet/EWMA.py`), and our implementation does not require TensorFlow (Abadi et al., 2015), which is required when installing ocpdet.). PyTorch 2.6.0 (Paszke et al., 2019) (used only for saving and loading the Gaussian and Poisson datasets and can be replaced with Numpy etc.), The operating system (OS) is Ubuntu 20.04. Intel(R) Core(TM) i9-7980XE CPU @ 2.60 GHz (18 cores) are used. The total random access memory (RAM) size of our server is 125 GBs. We do not use GPUs.

### D.1    ONLINE QCD MODELS

To see a practical relevance of our KME-based metrics, we evaluate six QCD models on the simulated Gaussian process dataset (App. C.3) and real-world WISDM Actitracker dataset (main text and App. C.5). We use the following online QCD models: Window L1 (Bai, 1995), Window Normal (Lavielle, 1999; Lavielle & Teyssiere, 2006), Window AR (Bai, 2000), non-parametric focused changepoint detection (NP-FOCuS) (Romano et al., 2024a), CUSUM (Page, 1954), and exponentially weighted moving average (EWMA) (Roberts, 1959). For Window L1, Window Normal, Window AR, the discrepancy measures for detection are derived from *cost functions* as specified in (Truong et al., 2020).

Window L1 detects changes in the median and uses the $L^1$ norm for the cost function. Window Normal detects changes in the mean and covariance matrix of a sequence of multivariate Gaussian random variables. It uses the log likelihood of empirical Gaussian distribution, for the computation of the cost function. Window AR estimates the least-squares estimates of the break dates obtained from a piecewise autoregressive (AR) model. NP-FOCuS is an online, non-parametric changepoint detection algorithm designed to efficiently detect distributional changes in real-time data streams by leveraging functional pruning techniques. For multivariate time-series, we instantiate $n$ detectors and take the minimum detection time, where $n$ is the number of features. CUSUM is a well-known QCD algorithm, which detects persistent shifts in the mean of a sequence. For multivariate time-series, we use the norm of the feature vector for the CUSUM chart. EWMA computes a running average of datapoints, placing exponentially decreasing weights on older observations.

Window L1, Window Normal, Window AR are implemented with `ruptures` (Truong et al., 2020), NP-FOCuS is implemented with `changepoint-online` (Romano et al., 2024b), and CUSUM and EWMA are implemented with `ocpdet` (Khamesi, 2022). We adapt window-based models from `ruptures`, originally designed for offline changepoint detection, to online QCD.

We fix both the window size and the burn-in interval at 30 frames, which is about three times smaller than the minimum maximum sequence length in our simulation experiments (100). It is also much smaller than the average lengths in the WISDM Actitracker. Most other hyperparameters remain at their default values; otherwise, they are specified in our code.

### D.2    PREPROCESSES FOR WISDM ACTITRACKER

Our preprocesses for the WISDM Actitracker dataset for both "labeled" and "unlabeled" subsets are comprised of the following procedures:

1. Extract user IDs ("user"), features ("X0", "X1", "X2", ..., "RESULTANT"), and frame labels (class").

2. Convert the string labels ("Jobbing", "Walking", "Stairs", "Sitting", "Standing", "Lying-Down") to binary numeric labels (0 for pre-change and 1 for post-change). In the "labeled"

|  | User-labeled set | Machine-labeled set |
|---|---|---|
| #Sequences | 83 | 51,326 |
| #Frames | 5,435 | 1,369,349 |
| #Seqs. w/ 100% positive labels | 37 | 76 |
| #Seqs. w/ 0% positive labels | 17 | 61 |
| #Seqs. w/ mixed labels | 29 | 51,189 |
| Positive frame ratio | 0.741 | 0.684 |
| Mean length | 65.5 | 26.7 |
| Min length | 1 | 1 |
| Max length | 565 | 54,401 |

Table 2: **Statistics of WISDM Actitracker after preprocesses.** A positive label here means that a frame (timestamp) is sampled from the post-change distribution. The user-labeled set and the machine-labeled set corresponds to the "labeled" set and the "unlabeled" set in the original WISDM Actitracker dataset. Note that the "unlabeled" set is actually labeled with a system developed in the WISDM Lab.

      subset, "Sitting" is mapped to 0 and all other activities to 1. In the "unlabeled" subset, "Walking" and "Sitting" are mapped to 1, while the remaining activities are mapped to 0.

3. Split sequences that have a label transition from 1 to 0 to remove turn-back sequences.

4. Zero-pad outlier features (value $> 10^{12}$) (although some huge values such as $10^{10}$ still remain).

5. Normalize each feature to the range $[-1, 1]$.

6. Save feature vectors, labels, and changepoints to a file.

All the preprocesses are detailed in our code (`WISDMactitracker.ipynb`). Finally, we manually exclude the sequences of length 1 when evaluating QCD algorithms (in `calc_esARL_WISDM_cpmodels.py` and `calc_esADD_WISDM_cpmodels.py` of our code).

### D.3 STATISTICS OF WISDM ACTITRACKER

Tab. 2 summarizes the statistics of the WISDM Actitracker dataset used in our experiments. The user-labeled and the machine-labeled subsets correspond to the "labeled" and the "unlabeled" subsets, respectively, in the original WISDM Actitracker dataset. Note that the "unlabeled" subset is, in fact, labeled with a system developed in the WISDM Lab (Kwapisz et al., 2011).

Fig. 18 shows histograms of sequence lengths of the user-labeled and machine-labeled subsets of the WISDM Actitracker dataset after the preprocesses, which are detailed in App. D.2. The sequence lengths exhibit substantial irregularity, and estimating the ARL and ADD is challenging.

Fig. 19 shows histograms of changepoint indices (timestamps) of the user-labeled and machine-labeled subsets of the WISDM Actitracker dataset after the preprocesses, which are detailed in App. D.2. They contain a variety of pre-change lengths, but the number of without-change sequences are limited, and estimating the ARL is challenging.

### D.4 NUMERICAL COMPUTATION OF VARIANCE

We describe the computation methods of the restricted variance of the detection time, $\hat{V} := \widehat{\text{Var}}[\min\{\tau, a\}]$, used in our experiments.

The restricted variance of LB-ARL is defined as the naive empirical variance: $\hat{V} = \widehat{\text{Var}}[\min\{\tau, T\}]$, which is consistent with the mean of LB-ARL ($\mathbb{E}[\min\{\tau, T\}]$). Given a dataset with size $N$, it is computed as $\frac{1}{N_{\text{LB}}} \sum_{i=1}^{N_{\text{LB}}} (\tau_i - \bar{\tau})^2$. $\bar{\tau}$ is the empirical LB-ARL, i.e., the detection time averaged over the subset in which $\tau_i \leq$ sequence length. The subset size is denoted by $N_{\text{LB}}(\leq N)$.

There are several methods to numerically compute the variance of RMST (KM-ARL). We adopt `lifelines` (Davidson-Pilon, 2019), a standard Python library for survival analysis, to compute the KME and its variance. Specifically, we use

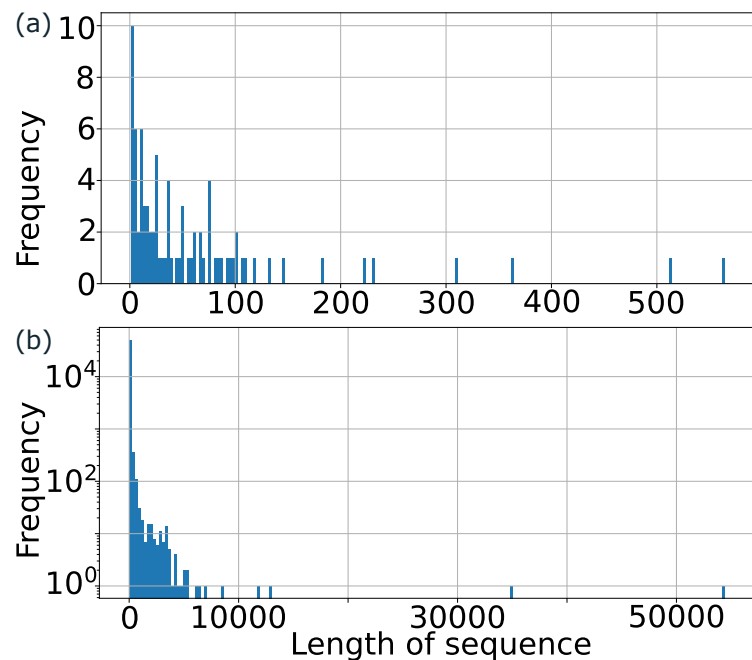

Figure 18: **Lengths of WISDM Actitracker sequences.** The sequence lengths exhibit substantial irregularity, and estimating the ARL and ADD is challenging. **(a) User-labeled subset ($y$-linear scale). (b) Machine-labeled subset ($y$-log scale).** The user-labeled and the machine-labeled subsets correspond to the "labeled" and the "unlabeled" subsets, respectively, in the original WISDM Actitracker dataset. Note that the "unlabeled" subset is, in fact, labeled with a system developed in the WISDM Lab (Kwapisz et al., 2011).

`lifelines.utils.restricted_mean_survival_time` with *return_variance=True*, in which $\hat{V} = 2\int_0^a t\hat{S}(t)dt - (\int_0^a \hat{S}(t)dt)^2$, where $\hat{S}(t)$ denotes the empirical survival function, computed from the given dataset with size $N$. The derivation is given in the appendix in (Royston & Parmar, 2013), as is mentioned in the documentation[1].

**Why do KM-ARL/ADD reduce variance compared to LB-ARL/ADD?** We elaborate on the discussion of our experimental results on the real-world dataset. In short, $\hat{V}$ of LB-ARL tends to larger than that of KM-ARL because the former can use only $N_{\text{LB}} \leq N$, while the latter can exploit all $N$ sequences. LB-ARL/ADD become unstable and exhibit high variance, particularly at large QCD thresholds, where $N_{\text{LB}}$ tends to be small because the detector often fails to raise an alarm within the sequence length. In contrast, KM-ARL/ADD do not suffer from this issue because they are calculated from a constant number of sequences $N$, regardless of the threshold, which enhances their robustness against censoring.

---

[1]`https://lifelines.readthedocs.io/en/latest/lifelines.utils.html#lifelines.utils.restricted_mean_survival_time.`

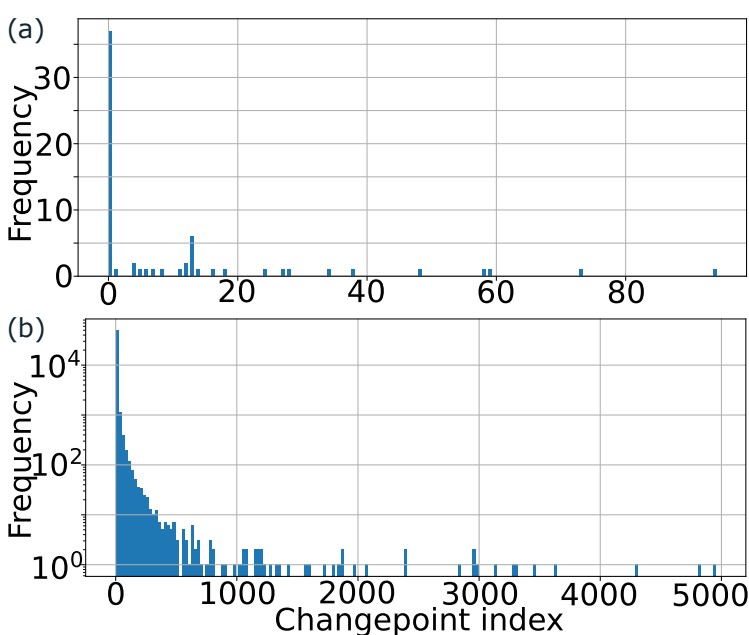

Figure 19: **Changepoint indices of WISDM Actitracker sequences.** Changepoint index here means the timestamp at which a change occurs, i.e., the pre-change length. They contain a variety of pre-change lengths, but the number of without-change sequences are limited, and estimating the ARL is challenging. **(a) User-labeled subset ($y$-linear scale).** The number of sequences without a changepoint is 17 out of 83. **(b) Machine-labeled subset ($y$-log scale).** The number of sequences without a changepoint is 61 out of 51326. The user-labeled and the machine-labeled subsets correspond to the "labeled" and the "unlabeled" subsets, respectively, in the original WISDM Actitracker dataset. Note that the "unlabeled" subset is, in fact, labeled with a system developed in the WISDM Lab (Kwapisz et al., 2011).

# E SUPPLEMENTARY RELATED WORK

**QCD metrics.** General QCD metircs include the Shiryaev's ARL (Shiryaev, 1963), Lorden's worst-case ADD (Lorden, 1971), Pollak's supremum conditional ADD (Pollak, 1985), exponential penalty detection delay (Poor, 1998), probability of false alarm (PFA) (Tartakovsky, 2019), precision-recall (Van den Burg & Williams, 2020), Jaccard index (Van den Burg & Williams, 2020), $F_\beta$ score (Van den Burg & Williams, 2020), range-based precision-recall (Tatbul et al., 2018), NAB (Lavin & Ahmad, 2015), SoftED (Salles et al., 2024), Hausdorff distance (Van den Burg & Williams, 2020), adjusted Rand index (Hubert & Arabie, 1985), variation of information (Arabie & Boorman, 1973), and segmentation covering metric, (Everingham et al., 2010; Arbelaez et al., 2010). In this paper, we focus on the standard ARL and Bayesian ADD (expectation is taken over the changepoint) because they are intuitive for practical evaluation and are widely used in proving the optimality of many QCD algorithms (Tartakovsky, 2019).

To our knowledge, no existing metric explicitly addresses random irregular sequence lengths in QCD. Some metrics consider sequence truncation, such as the inverse of the false alarm rate over a finite time interval $\Delta t$ (Alakent & Mutlu, 2018), which approximates the ARL for large $\Delta t$. The PFA in horizon (Huang & Veeravalli, 2024) is defined as the PFA measured on truncated sequences. However, these metrics are designed for fixed, finite-length sequences and introduce significant bias when applied to irregular-length sequences.

**ARL and ADD estimation.** There are many studies that derive the ARL and ADD of specific control charts or distributions: (Reynolds, 1975; Crowder, 1987; Saccucci & Lucas, 1990; Knoth & Knoth, 1998; Fu et al., 2002; Areepong & Peerajit, 2022; Haq & Woodall, 2022; Sunthornwat et al., 2023). Li et al. (2014) review simulation methods for computing the ARL and ADD of CUSUM (Page, 1954) and EWMA (Roberts, 1959), on simulation data. The references therein analyze Markov chain methods and integral equation methods. Alakent & Mutlu (2018) define the ARL of the Shewhart chart as the inverse of the alarm probability. In contrast, we propose the estimators of the ARL and ADD for arbitrary control charts, QCD models, and underlying distributions. Qiu (2013) defines a maximum length of the sequence for the ARL computation and excludes sequences that fail to detect changepoints by the maximum length. This estimator is similar to the LB-ARL and LB-ADD and exhibits a more substantial negative bias than our ARL estimator due to truncation, as proven in Sec.4 and demonstrated in Sec.5.

**Is ARL always informative?** (Mei, 2008) is the first paper in which the author systematically questions the appropriateness of the ARL. He claims that (i) a detection scheme with finite detection delay can have infinite ARL, where the detector can raise false alarms with probability 1, and (ii) under the standard minimax formulation with the ARL, we are in danger of finding a detection scheme that focuses on detecting larger changes instead of smaller changes. Claim (i) is theoretically relevant but practically irrelevant. It points out, in theory, that if the ARL is not assumed to be finite, one can construct a detector that raise false alarms. In practice, however, a detector with a small finite detection delay and an (approximately or numerically) infinite ARL is considered to perform well. This is not problematic in real-world applications. We agree with Claim (ii) because the minimax formulation of QCD, or more generally, the min and max operations are sensitive to outliers and fail to capture the average behavior of the model. For this reason, when used, they are typically complemented by average-based metrics such as the ARL and ADD. In summary, we encourage researchers to choose evaluation metrics with a clear understanding of their strengths and limitations.

**QCD model for finite sequence length.** To our knowledge, (Huang & Veeravalli, 2024) is the first and only work on the QCD problem on sequences of finite length. They propose a CUSUM variant for a fixed finite sequence length. If the model fails to detect the change before the horizon, they set the delay to $T - \nu$, causing a downward bias in the ADD.

**QCD for survival analysis.** There are many studies on the application of changepoint detection to survival analysis. In (Biswas & Kalbfleisch, 2008; Sego et al., 2009; GANDY et al., 2010; Phinikettos & Gandy, 2014; Gomon et al., 2024; Sasikumr & Sujatha, 2025), they propose variants of CUSUM to estimate survival time on right-censored data, which is orthogonal to our focus: accurately estimating ARLs and ADDs of arbitrary QCD algorithms on right-censored sequential data with irregular lengths. Gierz (2020) consider changepoint problems concerning a shift in a distribution for a set of

time-ordered observations under censoring or truncation. Polunchenko (2016) derive a closed-form formula for the survival function of the generalized Shiryaev-Roberts (GSR) detection time under Brownian motion.

**Parametric and closed-form approaches to ARL and ADD.** The exponential approximation of the underlying distribution provides accurate estimation of ARL and ADD in some scenarios. Also, closed-form expressions for ARL are known for some limited cases (e.g., (Fu et al., 2002; Areepong & Peerajit, 2022; Sunthornwat et al., 2023). However, it has been also known that: (1) the exponential approximation does not work well if the distribution deviates from exponential (Borror et al., 2003; Pehlivan & Testik, 2010); and (2) closed-form expressions are not always available—it depends on distributions and detectors. This is why we emphasize that our estimators are applicable to *arbitrary* underlying distributions (=non-parametric) and *arbitrary* QCD models, a key contributions that clearly distinguishes our work from prior approaches.

## F SUPPLEMENTARY DISCUSSION

**Terminology.** In general, the terms "average", "mean", and "expected" are used interchangeably when referring to the ARL and ADD. The ARL is also referred to as the ARL to false alarm (Tartakovsky, 2019) or the average time to signal (Li et al., 2014). The ADD with a fixed changepoint is also referred to as the conditional expected delay to detection (CEDD) (Tartakovsky, 2019). The ADD with taking expectation over the changepoint is referred to as the Bayesian ADD or, simply, ADD (Tartakovsky, 2019). In our paper, we refer the Bayesian ADD as the ADD (with $T = \infty$). In the statistical theory of control charts, including change detection in survival monitoring (or monitoring time-to-event data), the ARL and ADD are sometimes referred to as the in-control or zero-state ARL and the out-of-control or steady-state ARL, respectively (Saccucci & Lucas, 1990; Sasikumr & Sujatha, 2025; Lim & Lee, 2025). Strictly speaking, the steady-state ARL is defined as $\mathbb{E}[\tau \mid \nu = 0]$, not $\mathbb{E}[\tau - \nu \mid \tau \geq \nu, \nu < \infty]$.

**How to define $T$ and $T_{\max}^*$.** We clarify how to define the distribution of the truncation length (denoted by $F_T$ here), focusing on ARL, $T$ (truncation length), and $T_{\max}^*$ (the least upper bound for the support of $F_T$, i.e., the maximum possible sequence length). A parallel discussion applies to ADD. In brief, given an evaluation dataset, we can arbitrarily define $F_T$. In practice, once a test dataset is collected for model evaluation, $F_T$ may be taken as the empirical CDF of the test dataset (or a mollified (smoothed) version of it). Accordingly, $T_{\max}^*$ can be defined as the maximum sequence length in the test dataset, denoted by $T_{\max}$ in the main text.

**Tighter bound.** In Cor. 1.1 in (Stute, 1994), an additional tighter *upper* bound is available, which potentially can be used to derive tighter bounds for the KM-ARL and KM-ADD.

**Theoretical computational efficiency.** The computational complexity of KM-ARL is $\mathcal{O}(N) + \mathcal{O}(T_{\max})$ (plus event time sorting if necessary), where $N$ is the number of sequences in the dataset and $T_{\max}$ is the max sequence length. Bucketing the sequences into $T_{\max}$ bins takes $\mathcal{O}(N)$, counting $n_j^{\mathrm{ARL}}$ and $d_j^{\mathrm{ARL}}$ takes $\mathcal{O}(T_{\max})$, computing $\hat{S}^{\mathrm{ARL}}$ takes $\mathcal{O}(T_{\max})$, and integrating the step-like function $\hat{S}^{\mathrm{ARL}}$ takes $\mathcal{O}(T_{\max})$.

**Empirical computational efficiency.** As mentioned in App. D, the computational costs of metric evaluation, including KM-ARL and KM-ADD, are negligible in our experiments compared with the time required to run the QCD algorithms. For significantly longer sequences (e.g., $\gtrsim 10^3$, which is our maximum length) and larger datasets (e.g., $\gtrsim 10^4$, which is our maximum size), we recommend splitting the dataset and aggregating the ARLs and ADDs, although it can degrade evaluation accuracy.

**Evaluation on significantly small datasets.** Evaluation of ARLs and ADDs on significantly small datasets such as the user-labeled subset of the WISDM Actitracker is challenging, as shown in Fig. 15 in App. C.5. In such cases, consider applying finite-sample bias correction methods (e.g., bootstrapping) or increasing samples.

**Implementation in Python.** We implement our estimators based on Python (Van Rossum & Drake, 2009) because it is more compatible with machine learning development than R (R Core Team, 2021).

**Variance estimation.** While our primary focus is on bias estimation, we also provide a rough estimate of variance. The variance of LB-ARL (naive, standard definition of variance) is given by

$$\int_0^a (t - \bar{t})^2 dF(t), \tag{111}$$

where $a$ denotes the cut-off time, $F$ the CDF of the event time, and $\bar{t}$ the mean event time. On the other hand, Akritas (2000) provides the asymptotic variance of the RMST—analogous to the variance of KM-ARL in our setting:

$$\int_0^a \frac{S(t)}{1 - H(t)} (t - \bar{t})^2 dF(t) \tag{112}$$

(Eq. (4) in (Akritas, 2000)), where $S$ denotes the survival function and $H$ the CDF of $\min\{\text{event time}, \text{censoring}\}$. Compared with LB-ARL (111), the variance of KM-ARL (112) is computed as a weighted expectation of $(t - \bar{t})^2$, with its scale determined by the weight $\frac{S(t)}{1-H(t)}$. Because $S(t) \in [0, 1]$, this weight can grow large when $1 - H(t)$ is small unless $S(t)$ decreases sufficiently in the same region. Consequently, the variance integral in (112) can be dominated by the interval where $t$ is large and close to the least upper bound of $H(t)$. Therefore, the variance of KM-ARL can exceed that of LB-ARL when both event time and censoring have heavy tails and $a$ is large; otherwise, it may be smaller.

Finally, the assumptions and background theory in Akritas (2000) are intricate and require careful examination to validate this expression in QCD—one of the main challenges in developing our bias theory. A full treatment would warrant a separate paper.

**More about Heavy Censoring.** Demonstrating the numerical effect of bias under violations of the independent censoring assumption would be helpful for practitioners. According to Fig. 2 in (Malmquist, 2025), under a constant hazard rate and with the number of subjects (sequences) = 150, it is reported that the mean squared error of the survival function over time is approximately given by:

- For independent censoring (uniform censoring):
  - $< 0.01$ for 10% censoring.
  - $< 0.01$ for 50% censoring.
  - $< 0.01$ for 90% censoring.
- For dependent censoring (occurring just before event time (detection)):
  - $< 0.01$ for 10% censoring.
  - $\approx 0.4$ for 50% censoring.
  - $\approx 2.1$ for 90% censoring.

This result shows that heavy, dependent censoring inflates the bias. It aligns with our statement in Sec. 6, where we clarify the appropriate use cases for KM-ARL and KM-ADD.

### F.1 MORE ABOUT INDEPENDENT CENSORING ASSUMPTION

We extend our discussion in Sec. 6 about potential alleviation of the independent censoring assumption.

### F.1.1 BACKGROUND: RESTRICTED MEAN SURVIVAL TIME (RMST)

To mitigate the bias arising from dependent censoring and finite time interval, we must quantify the effect of the bias on the survival function or its integral within a finite time interval, known as the restricted mean survival time (RMST) in survival analysis. It is the counterpart of KM-ARL and KM-ADD in survival analysis and is defined as the expected survival time up to a specified horizon (denoted by $h$ here), equivalently the area under the survival curve from 0 to $h$. It has been known that the RMST is systematically biased under dependent censoring (Klein & Moeschberger, 1984; Rivest & Wells, 2001; Ebrahimi & Molefe, 2003).

### F.1.2 SOLUTIONS

Many studies have developed mitigation techniques for bias arising from dependent censoring. The following works, among others, suggest promising directions for deriving KM-ARL/ADD under dependent (informative) censoring or alleviating the resulting bias.

- (Ebrahimi & Molefe, 2003): Develops a consistent estimator of the survival function under dependent censoring, which may extend to estimating ARL and ADD in QCD.

- (Hsu & Taylor, 2010): Proposes a robust weighted KME that uses baseline prognostic covariates to correct bias in the marginal survival function when censoring is dependent.

- (Lin et al., 2023): Applies inverse probability of censoring weighting (IPCW); at each time $t$, each subject (patient) still under observation is weighted by the inverse of their estimated probability of remaining uncensored up to $t$.

- (Crommen et al., 2025): Summarizes several bias-correction strategies:
  - Nonparametric marginal estimation under a known copula: Given a specified copula for $(\tau, C)$, observable probabilities uniquely determine the marginals. Step-function estimators are constructed that reduce to the KME under independence and yield a consistent estimator under dependent censoring.
  - Likelihood estimation in copula models: Corrects bias by modeling the joint distribution of $(\tau, C)$; parametric copulas enable joint likelihood maximization with identifiable parameters and consistent, asymptotically normal estimates.
  - Correction of regression functionals: Embeds dependent censoring in copula-based Cox and related semi-parametric models to adjust hazard ratios, covariate effects, and causal effects.
  - Machine-learning and partial-identification approaches: Addresses bias when flexible modeling or weaker assumptions are required, using both deep and non-deep learning methods.

**Conclusion.** To avoid technical complications, we adopt the independent censoring assumption, a standard assumption in survival analysis, examine the conditions under which it holds in QCD, and specify them in the main text and this appendix. A full analysis of the above studies and their integration into QCD would warrant a separate paper and is indeed an ongoing research direction for the authors. Nonetheless, we are happy to present our current preliminary insights here, as we believe they offer useful starting points for future work and provide essential background for developing the emerging link between survival analysis and QCD.

### F.2 RELEVANCE OF REQUIRING DATASETS WITH MULTIPLE SEQUENCES WITH CHANGEPOINT LABELS

In this paper, we focus on the situations where datasets of multiple sequences with changepoint labels are available. This problem setting is common in machine learning. Although there are many studies of QCD on unlabeled datasets or pure simulations, there are also many studies on labeled datasets.

There are a wide variety of labeled datasets with multiple sequences and labeled changepoints (or annotations that can be seen as changepoints).

- Human sensing: WISDM Actitracker (Kwapisz et al., 2011) includes both human- and machine-labeled changepoints. We use this real-world sensing data in our experiments.

- Twitter sentiment-change detection: Twitter Data Stream (US Airline Sentiment) (Makone, 2016), a collection of tweets annotated with the tags of positive/negative/neutral, is used to detect sentiment changes in social media data stream (Bouchikhi et al., 2019): e.g., a sudden increase in negative tweets after an event.

- Speaker change detection: Many speech datasets, such as DIHARD dataset (Ryant et al., 2018) and IITG-MV phase 3 dataset (Haris et al., 2012), have speaker or language annotations, which can be used for speaker change detection (Mishra & Prasanna, 2024).

- Temporal action localization: THUMOS14 Jiang et al. (2014) and ActivityNet (Heilbron et al., 2015) are standard benchmark datasets in temporal action localization (Wang et al.,

2023a). They contain a number of videos with temporal annotations of human actions (e.g., Horseback riding, Diving, Long jump, etc.), where timestamps with action changes can be seen as changepoints.

- Earthquake detection: STEAD (Mousavi et al., 2019) and several datasets in SeisBench (Woollam et al., 2022) offer datasets of seismic signals with annotation of event times.

Furthermore, we can create labeled datasets from unlabeled real-world dataset by injecting change-points:

- Sound anomaly detection: Normal and abnormal audio data recorded from industry machine are concatenated to synthesize changepoints (Gopalan et al., 2021). The base dataset is the MIMI Dataset (Purohit et al., 2019).
- Social event detection: Based on the unlabeled MIT Cellphone Data (Eagle & Pentland, 2006), annotations are defined by linking changepoints to calender events (e.g., sponsor meeting, Presidents Day, spring break, etc.) (Kei et al., 2025).
- Social event detection: Similarly to (Kei et al., 2025), annotations are provided for Enron Email Data (Klimt & Yang, 2004) based on actual corporate events (e.g., Federal Energy Regulatory Commission's decisions, the CEO's public protests, the bankruptcy filing, etc.) (Wang et al., 2023b).

Of course, we can create labeled datasets via simulation (e.g., KDD Cup 1999 intrusion detection dataset (Stolfo et al., 1999)).

Thus, real-world applications include: human sensing and action recognition (Bishop, 2006; Wang et al., 2023a), sentiment-change detection on social media data stream (Bouchikhi et al., 2019), speaker/language change detection (Mishra & Prasanna, 2024), earthquake detection (Woollam et al., 2022), sound anomaly detection (Gopalan et al., 2021), event detection from cellphone data or email data (Kei et al., 2025; Wang et al., 2023b), and intrusion detection (Stolfo et al., 1999), among others.

In summary, although our estimators cannot be used for unlabeled datasets, as noted in the main text, they apply to many real-world tasks listed above, among others.

**Our contributions to machine learning community.** Finally, we would like to note that analyzing how we evaluate performance metrics has been of general interest in machine learning (Hernàndez-Orallo et al., 2004; Int, 2005; 2006; Pardo-Fernández & Castro, 2025). Recent top-tier conferences themselves emphasize this by requiring careful reporting of metrics in their Author Instructions and Reviewer Instructions, underscoring that reproducible and interpretable evaluation is a first-class concern, not an afterthought. Our work advances more robust, interpretable evaluation of QCD models, enabling empirical, intuitive model selection, as stated in the Abstract and Experiment sections.

## G   USE OF LARGE LANGUAGE MODELS

During our research, we use LLMs to polish writing, finding related work, and verify our proof.

