# OpenReview forum: "Accurate Evaluation of Quickest Changepoint Detectors via Non-parametric Survival Analysis"
_ICLR.cc/2026/Conference — Submitted to ICLR 2026_

### Official Review · Reviewer_jxVp · 2025-10-17

**Soundness:** 4
**Presentation:** 3
**Contribution:** 4
**Rating:** 6
**Confidence:** 3

**Summary:**

This paper proposes nonparametric estimators for average run length (ARL) and average detection delay (ADD) in online quickest changepoint detection (QCD) settings with finite, irregular-length real-world sequences. By adapting the Kaplan–Meier estimator from survival analysis, the authors introduce the KM-ARL and KM-ADD estimators, derive their bias properties, and show they are asymptotically unbiased under standard regularity conditions.

**Strengths:**

The paper addresses a critical and practical challenge in QCD evaluation: accurately estimating metrics in the presence of sequence truncation and irregularity, which is common in real-world applications. The theoretical contributions are robust, including finite-sample and truncation bias bounds, and conditions for asymptotic unbiasedness.

**Weaknesses:**

While the theoretical contributions are nice, my concern is how much interest this work will attract from the ICLR community. As the conference name implies, this conference is primarily for researchers working on the theories, models, and applications of learning representations. However, I would still like to support this work based on its strong theoretical contributions.

**Questions:**

1. In Theorem 4.1, the stopping time $\tau$ and censoring time are assumed to be independent. Under what scenarios will this hold? Wouldn't that mean that the detector is a "weak" detector?

1. How can the KM-ARL in (2) and KM-ADD in (3) be computed numerically?

1. The authors note that if "dependent censoring occurring just before event time (detection), estimation bias increases sharply". It would be interesting to see how much is the bias effect numerically to give practitioners a sense of the limitations of the approach.

---

> ### Author Response · Authors · 2025-11-23
>
> We deeply appreciate the reviewer for recognizing our paper's strengths and supporting acceptance.
> We would like to address all concerns and questions in our response.
>
> For later discussion, please let us summarize the weakness and questions the reviewer raised:
>
> - W1: How much interest this work attracts from the ICLR community.
> - Q1: In Theorem 4.1, the stopping time $\tau$ and censoring time $C^{\mathrm{ARL}}$ are assumed to be independent (i.e., independent censoring assumption for KM-ARL). Under what scenarios will this hold? Wouldn't that mean that the detector is a "weak" detector?
> - Q2: How can the KM-ARL in (2) and KM-ADD in (3) be computed numerically?
> - Q3: The authors note that if "dependent censoring occurring just before event time (detection), estimation bias increases sharply" (line 483 of the original submission). It would be interesting to see how much is the bias effect numerically to give practitioners a sense of the limitations of the approach.

---

> > ### Author Response · Authors · 2025-11-23
> >
> > ## W1
> >
> > The reviewer is concerned about the level of interest this work will attract from the ICLR community; nevertheless, we appreciate your support for its acceptance based on our strong theoretical contributions.
> >
> > We submit our work to ICLR because it has published papers on (1) statistical properties of performance metrics, (2) statistical bias analysis, and (3) changepoint detection. These three constitute the main topic of our paper. We provide examples of such ICLR papers, among others:
> >
> > 1. Statistical properties of performance metrics (theoretical and empirical):
> >    - [jxVp6] focuses on the evaluation of generative models. It analyzes average log-likelihood, Parzen window log-likelihood estimates, and sample visual fidelity and argues that these three evaluation criteria are largely independent in high dimensions.
> >    - [jxVp7] questions how reliable calibration metrics of deep neural networks are and provides practical takeaways.
> >    - [jxVp8] formalizes top-label calibration error (calibration of the predicted class' probability) and introduces multiclass-to-binary reduction schemes, proving distribution-free calibration guarantees.
> > 2. Statistical bias analysis:
> >    - [jxVp21] dhows that mini-batch quadratic estimators are systematically biased relative to their full-batch counterparts.
> >    - [jxVp22] derives an unbiased gradient estimator based on sampling without replacement.
> >    - [jxVp23] formalizes the statistical bias of the usual empirical risk when data are selected adaptively rather than i.i.d.
> >    - [jxVp24] gives an explicit asymptotic analysis showing that standard cross-validation risk estimates are biased in high-dimensional ridge.
> > 3. Changepoint detection:
> >    - [jxVp14] addresses segmented, piece-wise models for sequential data with changepoints. The paper proposes a way to make sequence segmentation fully differentiable so that both segment boundaries and model parameters can be learned jointly with gradient descent.
> >    - [jxVp15] proposes KL-CPD, a kernel learning framework for changepoint detection that optimizes a lower bound of test power via an auxiliary generative model as a surrogate to the abnormal distribution.
> >    - [jxVp16] proposes the Anomaly Transformer, which introduces an association discrepancy criterion for detecting anomalies in time-series data via a Transformer architecture.
> >    - [jxVp17] addresses unsupervised anomaly detection in multivariate streaming data by combining normalizing flows with a learnable dependency structure.
> >    - [jxVp18] tackles the meta-problem of choosing the best anomaly detector for a given time-series dataset without labels.
> >    - [jxVp19] is a framework designed for detecting anomalies in high-dimensional multivariate time series by capturing fine-grained frequency information and inter-channel relationships.
> >    - [jxVp20] focuses on adaptive hypothesis testing in the context of large language models (LLMs).
> >
> > Although we found no ICLR papers that directly address statistical bias estimation for QCD performance metrics, many papers discuss individual components related to our work.
> > Given these circumstances, submitting our work to ICLR was a natural choice for us (and, by the way, we recognize that ICML is also appropriate).
> >
> > In addition, we show the practical relevance and broad applicability of our estimators in our General Response (2-1. Relevance of requiring datasets with multiple sequences with changepoint labels).
> >
> > Finally, we would like to note that analyzing how we evaluate performance metrics has been of general interest in machine learning [jxVp1] [jxVp2] [jxVp3] [jxVp13]. Top-tier conferences themselves emphasize this by requiring careful reporting of metrics in their Author Instructions (https://iclr.cc/Conferences/2026/AuthorGuide) and Reviewer Instructions (https://iclr.cc/Conferences/2026/ReviewerGuide), underscoring that reproducible and interpretable evaluation is a first-class concern, not an afterthought. Our work advances more robust, interpretable evaluation of QCD models, enabling empirical, intuitive model selection, as stated in the Abstract and Experiment sections.
> >
> > We added a contribution statement for the community in Appendix F (Supplementary Discussion) of the revised version.
> > We hope our clarification is helpful.

---

> > > ### Author Response · Authors · 2025-11-23
> > >
> > > ## Q1
> > >
> > > > In Theorem 4.1, the stopping time $\tau$ and censoring time $C^{\mathrm{ARL}}$ are assumed to be independent (i.e., independent censoring assumption for KM-ARL). Under what scenarios will this hold?
> > >
> > > - Note: This assumption (Assumption (ii)) is referred to the *independent censoring assumption* for KM-ARL, where $\tau$ is referred to as the stopping time, event time, or detection time. $C^{\mathrm{ARL}}$ represents the random truncation length of the sequence.
> > > - Answer: We state the condition under which this assumption holds/does not hold in lines 215-220 of the original submission (below Theorem 4.1):
> > > "Assumption (ii) holds for online QCD models that do not look ahead the input sequence, which are the focus in this paper.
> > > This is because: detection points $\tau_i$ prior to censoring can be regarded as samples from $P(\tau \mid \nu=\infty)$; $P(\tau \mid \nu=\infty)$, $P(\nu)$, and $P(T)$ are independent; and thus, $P(\tau \mid \nu=\infty)$ and $P(C:= \min\{\nu, T\})$ are also independent.
> > > In contrast, Assumption (ii) does not hold for offline changepoint detection models because $\tau$ depends on $X^{(0, T)}$, which in turn depends on $\nu$ and $T$."
> > > - Please let us know if additional clarification is needed. We warmly welcome any suggestions that could improve our manuscript.
> > >
> > > > Wouldn't that mean that the detector is a "weak" detector?
> > >
> > > - Answer: No, it does not.
> > > - Rationale: $\tau$ and $C^{\mathrm{ARL}}$ denote the detection time and truncation length, respectively. They are assumed to be independent because the sequence length is usually uncontrollable. By contrast, if the detection time $\tau$ *were* independent of the *changepoint* $\nu$—rather than of $C^{\mathrm{ARL}}$—the detector would be weak, which we do not assume here. We suspect the reviewer may be considering this scenario.
> > > - Note: For the scenario in which the detection time $\tau$ may depend on the truncation length $C^{\mathrm{ARL}}$—an arrangement that is theoretically possible—please refer to our response to Reviewers FXnt (W1) and jxVp (Q3), where we address potential violations of the independent censoring assumption in Theorem 4.2 (KM-ADD) and expand the supplementary discussion for future work. The same reasoning applies to Theorem 4.1.
> > >
> > > ## Q2
> > >
> > > The numerical computation of Eqs. (2) and (3) (KM-ARL and KM-ADD) is simple and efficient. The survival functions $\hat{S}^{\mathrm{ARL}}(t)$ and $\hat{S}^{\mathrm{ADD}}(t)$ follow directly from the detection outcomes of the dataset through simple arithmetic (lines 151 and 175 of the original submission). Moreover, the numerical integrals $\int_0^a \hat{S}^{\mathrm{ARL}}(t) dt$ and $\int_0^b \hat{S}^{\mathrm{ADD}}(t) dt$ are easily computed because the survival functions are piecewise constant (sums of step functions), as illustrated in Figure 5 of the original submission.
> > >
> > > For the exact computation code, please see the `calc_KME` and `calc_KME_ADD` functions in our code (`./statistic_tools.py` in the Supplementary Materials).
> > > We provide user-friendly, ready-to-use code (see `README.md` and `quick_start.ipynb`) and will be released on GitHub in addition to OpenReview.
> > >
> > > Additional Note: The computational complexity of their numerical computation is provided in Appendix F ($O(N) + O(T_{\mathrm{max}})$, where $N$ and $T_{\mathrm{max}}$ denote the dataset size and the max sequence length, respectively: lines 1908-1912 of the original submission), which is negligible in our experiments (lines 1914-1919 of the original submission). For significantly long sequences and large datasets (e.g., $T_{\mathrm{max}} \gg 10^3$ and $N \gg 10^4$), we recommend splitting the dataset and aggregate the ARLs and ADDs (lines 1916-1919 of the original submission), as is often the case for large test datasets.

---

> ### Author Response · Authors · 2025-11-23
>
> ## Q3
>
> We agree that demonstrating the numerical effect of bias under violations of the independent censoring assumption would be helpful for practitioners. Thank you for the feedback.
>
> We extend the subsection that the reviewer mentions (lines 478-485 in Section 6 of the original submission).
> According to Figure 2 in Malmquist (2025), under a constant hazard rate and with the number of subjects (sequences) = 150, it is reported that the mean squared error of the survival function over time is approximately given by:
>
> - For independent censoring (uniform censoring):
>   - < 0.01 for 10% censoring.
>   - < 0.01 for 50% censoring.
>   - < 0.01 for 90% censoring.
>
> - For dependent censoring (occurring just before event time (i.e., detection time in QCD settings)):
>   - < 0.01 for 10% censoring.
>   - ~ 0.4 for 50% censoring.
>   - ~ 2.1 for 90% censoring.
>
> This result shows that heavy, dependent censoring inflates the bias.
> It aligns with our statement in Section 6, where we clarify the appropriate use cases for KM-ARL and KM-ADD.
>
> Section 6 also outlines potential approaches to mitigate heavy, dependent censoring (lines 472–485 of the original submission).
> In addition, we have expanded Appendix F to provide further details on these approaches in response to the reviewers' feedback (Reviewers FXnt (W1) and jxVp (Q3)).
>
> ----------------------------
> Again, thank you for your feedback, which we believe has made our contributions clearer and more convincing.
> If there are specific points we could clarify or improve to merit a higher assessment, please let us know.
> We would be happy to continue and improve the manuscript.
>
> ~~**The revised manuscript will be submitted in a few days.**~~ **Uploaded.**

---

> > ### Author Response · Authors · 2025-11-23
> >
> > ## References
> >
> > - [jxVp1] Hernàndez-Orallo, José, et al. "The 1st workshop on ROC analysis in artificial intelligence (ROCAI-2004)." ACM SIGKDD Explorations Newsletter 6.2 (2004): 159-161. URL: https://dl.acm.org/doi/10.1145/1046456.1046489.
> > - [jxVp2] Lachiche, N., et al., editors. Proceedings of the Second Workshop on ROC Analysis in Machine Learning (ROCML'05). International Conference on Machine Learning (ICML'05), 2005, http://icube-mecaflu.unistra.fr/11-LFMR05.
> > - [jxVp3] Lachiche, N., et al., editors. Proceedings of the Third Workshop on ROC Analysis in Machine Learning (ROCML'06). International Conference on Machine Learning (ICML'06), 2006, http://publis.icube.unistra.fr/11-LFM06.
> > - [jxVp4] Drummond, Chris, et al. "Workshop summary: The fourth workshop on evaluation methods for machine learning." Proceedings of the 26th Annual International Conference on Machine Learning. 2009. URL: https://aaai.org/proceeding/ws06-06/
> > - [jxVp5] Drummond, Chris, William Elazmeh, and Nathalie Japkowicz, editors. Evaluation Methods for Machine Learning. AAAI Workshop Papers WS-06-06, 2006, https://aaai.org/proceeding/ws06-06/.
> > - [jxVp6] Theis, Lucas, Aäron van den Oord, and Matthias Bethge. A Note on the Evaluation of Generative Models. ICLR, 2016, arxiv.org/abs/1511.01844. URL: https://openreview.net/forum?id=XbBwu75Qb5.
> > - [jxVp7] Tao, Linwei, et al. A Benchmark Study on Calibration. The Twelfth International Conference on Learning Representations, 2024, https://openreview.net/forum?id=GzNhzX9kVa.
> > - [jxVp8] Gupta, Chirag, and Aaditya Ramdas. Top-Label Calibration and Multiclass-to-Binary Reductions. International Conference on Learning Representations, 2022, https://openreview.net/forum?id=WqoBaaPHS-.
> > - [jxVp13] ROC2025: International Workshop on ROC Analysis and Related Topics. Universidade de Vigo, 2025, https://roc2025.webs.uvigo.es/.
> > - [jxVp14] Scharwächter, Erik, Jonathan Lennartz, and Emmanuel Müller. Differentiable Segmentation of Sequences. International Conference on Learning Representations, 2021, https://openreview.net/forum?id=4T489T4yav.
> > - [jxVp15] Chang, Wei-Cheng, et al. Kernel Change-point Detection with Auxiliary Deep Generative Models. International Conference on Learning Representations, 2019, https://openreview.net/forum?id=r1GbfhRqF7.
> > - [jxVp16] Xu, Jiehui, et al. Anomaly Transformer: Time Series Anomaly Detection with Association Discrepancy. International Conference on Learning Representations, 2022, https://openreview.net/forum?id=LzQQ89U1qm.
> > - [jxVp17] Dai, Enyan, and Jie Chen. Graph-Augmented Normalizing Flows for Anomaly Detection of Multiple Time Series. International Conference on Learning Representations, 2022, https://openreview.net/forum?id=45L_dgP48Vd.
> > - [jxVp18] Goswami, Mononito, et al. Unsupervised Model Selection for Time Series Anomaly Detection. The Eleventh International Conference on Learning Representations, 2023, https://openreview.net/forum?id=gOZ_pKANaPW.
> > - [jxVp19] Wu, Xingjian, et al. CATCH: Channel-Aware Multivariate Time Series Anomaly Detection via Frequency Patching. The Thirteenth International Conference on Learning Representations, 2025, https://openreview.net/forum?id=m08aK3xxdJ.
> > - [jxVp20] Richter, Leo, et al. An Auditing Test to Detect Behavioral Shift in Language Models. The Thirteenth International Conference on Learning Representations, 2025, https://openreview.net/forum?id=h0jdAboh0o.
> > - [jxVp21] Tatzel, Lukas, et al. Debiasing Mini-Batch Quadratics for Applications in Deep Learning. The Thirteenth International Conference on Learning Representations, 2025, https://openreview.net/forum?id=Q0TEVKV2cp.
> > - [jxVp22] Kool, Wouter, et al. Estimating Gradients for Discrete Random Variables by Sampling without Replacement. International Conference on Learning Representations, 2020, https://openreview.net/forum?id=rklEj2EFvB.
> > - [jxVp23] Farquhar, Sebastian, et al. On Statistical Bias in Active Learning: How and When to Fix It. International Conference on Learning Representations, 2021, https://openreview.net/forum?id=JiYq3eqTKY.
> > - [jxVp24] Liu, Sifan, and Edgar Dobriban. Ridge Regression: Structure, Cross-Validation, and Sketching. International Conference on Learning Representations, 2020, https://openreview.net/forum?id=HklRwaEKwB.

---

> > > ### Comment · Reviewer_jxVp · 2025-11-25
> > >
> > > I thank the authors for the detailed rebuttal that have clarified my doubts. I am happy to keep my current positive score.

---

### Official Review · Reviewer_dkXY · 2025-10-31

**Soundness:** 2
**Presentation:** 3
**Contribution:** 2
**Rating:** 4
**Confidence:** 3

**Summary:**

The paper addresses the challenge of evaluating the performance of online change‑point detection methods. The authors are interested of two key metrics: the average running length (ARL) and the average detection delay (ADD). Inspired by survival analysis (and, in particular the Kaplan‑Meier (KM) survival function estimator), the authors introduce new estimators called KM‑ARL and KM‑ADD. The suggested methods aim to estimate ARL and ADD, respectively. They thoroughly study biases of KM‑ARL and KM‑ADD estimates. They represent the total bias into a sum of truncation and finite-sample biases. According to Theorems 4.1 and 4.2, the finite‑sample biases of KM‑ARL and KM‑ADD decay exponentially as the sample size approaches infinity. In addition, Theorems 4.3 and 4.4 demonstrate that the truncation biases associated with KM‑ARL and KM‑ADD are smaller in absolute value compared to those of conventional LB‑ARL and LB‑ADD methods. To further validate their approach, the researchers present numerical experiments using both synthetic and real‑world datasets. These experiments clearly illustrate the superiority of KM‑ARL and KM‑ADD estimates when compared to LB‑ARL, LB‑ADD, as well as to naive ARL and ADD estimation methods.

**Strengths:**

Strong theoretical guarantees on biases of the KM‑ARL and KM‑ADD estimates.

**Weaknesses:**

1. The variances of KM-ARL and KM-ADD remain unexplored. Could the authors comment on comparison of the variances of KM-ARL and KM-ADD with the ones of LB-ARL and LB-ADD?

2. According to Theorems 4.3 and 4.4, a statistician must know $T_{\max}^\star$ and $\Delta T_{\max}^\star$ to ensure that the KM-ARL and KM-ADD have smaller biases than their LB competitors. However, the setup does not describe the distribution of $T_i$'s, so it is unclear how to find $T_{\max}^\star$ and $\Delta T_{\max}^\star$ in practice.

3. The experiments in Section 5 were designed in such a way that the values of $T_{\max}^\star$ and $\Delta T_{\max}^\star$ were available. For this reason, it is not surprising that the KM estimators performed better than their LB counterparts. It is not clear how the KM estimators will behave if a statistician chooses wrong $T_{\max}^\star$ and $\Delta T_{\max}^\star$.

4. The last sentence in the proof of Theorem 4.4 (restated as Theorem B.5 in Appendix B.4): it is not clear why function $\mathbb E [ \Delta \tau \,\vert\, \nu < 0, 0 \leq \Delta \tau \leq \Delta T_{\max}] - \mathbb E [ \Delta \tau \,\vert\, \nu < 0, 0 \leq \Delta \tau \leq \Delta T] \geq 0$. I would prefer to see a rigorous proof of this inequality.

**Questions:**

1. Can the authors comment on the variances of KM-ARL and KM-ADD? How do they compare with the variances of LB-ARL and LB-ADD in the numerical experiments? Is it possible to prove rigorous theoretical results relating variances of KM-ARL and KM-ADD with the ones of LB-ARL and LB-ADD?

2. What happens with the KM estimators if a statistician chooses wrong $T_{\max}^\star$ and $\Delta T_{\max}^\star$?

---

> ### Author Response · Authors · 2025-11-23
>
> We appreciate the reviewer for their detailed, insightful feedback.
> We would like to address all concerns and questions below.
>
> For later discussion, please let us summarize the weaknesses and questions the reviewer raised:
>
> - W1: The variances of KM-ARL and KM-ADD remain unexplored. Could the authors comment on comparison of the variances of KM-ARL and KM-ADD with the ones of LB-ARL and LB-ADD?
> - W2: According to Theorems 4.3 and 4.4, a statistician must know $T_{\mathrm{max}}^{\*}$ and $\Delta T_{\mathrm{max}}^{\*}$ to ensure that the KM-ARL and KM-ADD have smaller biases than their LB competitors. However, the setup does not describe the distribution of $T_i$'s, so it is unclear how to find $T\_{\mathrm{max}}^{\*}$ and $\Delta T\_{\mathrm{max}}^{\*}$ in practice.
> - W3-1: The experiments in Section 5 were designed in such a way that the values of $T\_{\mathrm{max}}^{\*}$ and $\Delta T\_{\mathrm{max}}^{\*}$ were available. For this reason, it is not surprising that the KM estimators performed better than their LB counterparts.
> - W3-2: It is not clear how the KM estimators will behave if a statistician chooses wrong $T_{\mathrm{max}}^{\*}$ and $\Delta T_{\mathrm{max}}^{\*}$.
> - W4: The last sentence in the proof of Theorem 4.4 (restated as Theorem B.5 in Appendix B.4): it is not clear why function $\mathbb{E}[ \Delta \tau \mid \nu < \infty, 0 \leq \Delta \tau \leq \Delta  T_\mathrm{max}^*] - \mathbb{E}[\Delta \tau \mid \nu < \infty, 0 \leq \Delta \tau \leq \Delta T] \geq 0$. I would prefer to see a rigorous proof of this inequality.
> - Q1-1: *(Same as W1)* Can the authors comment on the variances of KM-ARL and KM-ADD?
> - Q1-2: How do they compare with the variances of LB-ARL and LB-ADD in the numerical experiments?
> - Q1-3: Is it possible to prove rigorous theoretical results relating variances of KM-ARL and KM-ADD with the ones of LB-ARL and LB-ADD?
> - Q2: *(Same as W3-2)* What happens with the KM estimators if a statistician chooses wrong $T_{\mathrm{max}}^{\*}$ and $\Delta T_{\mathrm{max}}^{\*}$?

---

> > ### Author Response · Authors · 2025-11-23
> >
> > ## W1 (Q1-1)
> >
> > We discuss variance estimation in Appendix F (lines 1928–1938 of the original submission), as Reviewer FXnt acknowledges.
> > Nonetheless, to address the reviewer's concern, we would like to provide additional explanation from both empirical (Q1-2) and theoretical (Q1-3) perspectives below.
> > We focus on LB-ARL and KM-ARL; a similar discussion applies to LB-ADD and KM-ADD.
> >
> > ## Q1-2 and Q-1-3
> >
> > ### 1. Theoretical Perspective
> >
> > We add more details of our brief discussion in Appendix F.
> > The variance of LB-ARL (naive, standard definition of variance) is given by
> >
> > $\int_0^a (t - \bar{t})^2  dF(t)$, $\cdots$ (A)
> >
> > where $a$ denotes the cut-off time, $F$ the CDF of the event time, and $\bar{t}$ the mean event time.
> > On the other hand, Akritas (2000) provides the asymptotic variance of the RMST—analogous to the variance of KM-ARL in our setting:
> >
> > $\int_0^a \frac{S(t)}{1 - H(t)} (t - \bar{t})^2  dF(t)$ $\cdots$ (B)
> >
> > (Eq. (4) in (Akritas, 2000)), where $S$ denotes the survival function and $H$ the CDF of $\min \{\mathrm{event\,time}, \mathrm{censoring} \}$.
> > Compared with LB-ARL (A), the variance of KM-ARL (B) is computed as a weighted expectation of $(t - \bar{t})^2$, with its scale determined by the weight $\frac{S(t)}{1 - H(t)}$.
> > Because $S(t) \in [0, 1]$, this weight can grow large when $1-H(t)$ is small unless $S(t)$ decreases sufficiently in the same region. Consequently, the variance integral in (B) can be dominated by the interval where $t$ is large and close to the least upper bound of $H(t)$. Therefore, the variance of KM-ARL can exceed that of LB-ARL when both event time and censoring have heavy tails and $a$ is large; otherwise, it may be smaller.
> >
> > Finally, the assumptions and background theory in Akritas (2000) are intricate and require careful examination to validate this expression in QCD—one of the main challenges in developing our bias theory. A full treatment would warrant a separate paper.
> >
> > We have expanded our variance discussion in the revised manuscript (Appendix F).

---

> > > ### Author Response · Authors · 2025-11-23
> > >
> > > ### 2. Empirical Perspective
> > >
> > > We describe the empirical, practical computation of the restricted variance of the detection time, $\hat{V} := \widehat{\mathrm{Var}} \[ \min \\{\tau, a\\} \]$, in our experiments.
> > >
> > > - LB-ARL:
> > >   - The restricted variance of LB-ARL is defined as the naive empirical variance: $\hat{V} = \widehat{\mathrm{Var}} \[ \min \\{\tau, T \\} \]$, which is a consistent definition with the mean of LB-ARL ($\mathbb{E}[\min \\{\tau, T \\} ]$). Given a dataset with size $N$, it is computed as $\frac{1}{N_{\mathrm{LB}}} \sum_{i=1}^{N_{\mathrm{LB}}} (\tau_i - \bar{\tau})^2$. $\bar{\tau}$ is the empirical LB-ARL, i.e., the detection time averaged over the subset in which $\tau_i \leq$ sequence length. The subset size is denoted by $N_{\mathrm{LB}} (\leq N)$.
> > >
> > > - KM-ARL:
> > >   - There are several methods to numerically compute the variance of RMST (KM-ARL). As stated in Appendix D (Detailed Experimental Settings), we adopt `lifelines` (Davidson-Pilon, 2019), a standard Python library for survival analysis, to compute the KME and its variance.
> > >   - Specifically, we use `lifelines.utils.restricted_mean_survival_time` with `return_variance=True`, in which $\hat{V} = 2 \int_{0}^{a} t \hat{S}(t) d t - (\int_0^{a} \hat{S}(t) d t )^2$, where $\hat{S}(t)$ denotes the empirical survival function, computed from the given dataset with size $N$. The derivation is given in the appendix in [dkXY1], as is mentioned in [dkXY2].
> > >
> > > - Why do KM-ARL/ADD reduce variance compared to LB-ARL/ADD?
> > >   - We elaborate on the discussion of our experimental results on the real-world dataset (lines 438-446 of the original submission).
> > >   - In short, $\hat{V}$ of LB-ARL tends to larger than that of KM-ARL because the former can use only $N_{\mathrm{LB}} \leq N$, while the latter can exploit all $N$ sequences.
> > >     - LB-ARL/ADD become unstable and exhibit high variance, particularly at large QCD thresholds, where $N_{\mathrm{LB}}$ tends to be small because the detector often fails to raise an alarm within the sequence length (Figure 1(a)).
> > >     - In contrast, KM-ARL/ADD do not suffer from this issue because they are calculated from a constant number of sequences $N$, regardless of the threshold, which enhances their robustness against censoring.
> > >
> > > We have expanded our discussion on experimental results and settings in the revised manuscript (Section 5 and Appendix D).
> > >
> > > Thank you for your feedback, and we hope our clarification and additional analysis of variance address the reviewer's concern.
> > > Please let us know if further information is needed for improving our manuscript.

---

> > > > ### Author Response · Authors · 2025-11-23
> > > >
> > > > ## W2
> > > >
> > > > Thank you for the suggestion.
> > > > We would like to clarify how to define the distribution of the truncation length (denoted by $F_T$ here), focusing on ARL, $T$ (truncation length), and $T_{\mathrm{max}}^{*}$ (the least upper bound for the support of $F_T$, i.e., the maximum possible sequence length).
> > > > A parallel discussion applies to ADD.
> > > >
> > > > In brief, given an evaluation dataset, we can arbitrarily define $F_T$.
> > > > In practice, once a test dataset is collected for model evaluation, $F_T$ may be taken as the empirical CDF of the test dataset (or a mollified (smoothed) version of it).
> > > > Accordingly, $T_{\mathrm{max}}^{*}$ can be defined as the maximum sequence length in the test dataset, denoted by $T_{\mathrm{max}}$ in the main text (line 162 of the original submission).
> > > >
> > > > ## W3-1
> > > >
> > > > As noted in the response to W2, $T_{\mathrm{max}}^{\*}$ and $\Delta T_{\mathrm{max}}^{\*}$ are known in all cases, so it poses no limitation.
> > > >
> > > > ## W3-2 (Q2)
> > > >
> > > > Although we have shown that $T_{\mathrm{max}}^{\*}$ and $\Delta T_{\mathrm{max}}^{\*}$ are straightforwardly defined from the test dataset, we here clarify what happens if a statistician chooses wrong $T_{\mathrm{max}}^{\*}$ in Theorem 4.3 (truncation bias of KM-ARL).
> > > > A similar discussion applies to $\Delta T_{\mathrm{max}}^{*}$.
> > > >
> > > > According to Eq. (57) in the proof of Theorem 4.3, replacing $T_{\mathrm{max}}$ with $a$, we can see that:
> > > >
> > > > - If we choose $a > T_{\mathrm{max}}^{*}$, the truncation-bias bound still holds.
> > > >   - However, the finite-sample-bias bound may become looser, as noted in lines 225-228 of the original submission.
> > > > - If we choose $a < T_{\mathrm{max}}^{*}$, the truncation-bias bound may violate.
> > > >   - However, the finite-sample-bias bound becomes tighter.
> > > >
> > > > ### Note (just in case)
> > > >
> > > > Please do not confuse $a$ and $b$ with $T_{\mathrm{max}}^{\*}$ and $\Delta T_{\max}^{\*}$.
> > > > In our experiments, the upper bounds of the integrals in Eq. (2) and Eq. (3) (definitions of KM-ARL and KM-ADD, respectively), denoted by $a$ and $b$, are set to $T_{\max}$ and $\Delta T_{\max}$, respectively.
> > > > This is clarified in the main text along with the corresponding rationales.
> > > > Specifically:
> > > >
> > > > - (lines 162-165 of the original submission) we explain how choosing $a$ too large or too small affects the finite-sample bias.
> > > > - (lines 260-264 of the original submission) we show that setting $a = T_{\max}^{*}$ is necessary to ensure the truncation-bias bound in Theorem 4.3.
> > > > - (lines 279-292 of the original submission) we analyze how the total estimation bias behaves when $a$ is taken to be large or small.

---

> > > > > ### Author Response · Authors · 2025-11-23
> > > > >
> > > > > ## W4
> > > > >
> > > > > Thank you very much for your careful reading and insightful comment.
> > > > > The last sentence in the proof of Theorem 4.4 was incorrect, and we have updated the manuscript accordingly while preserving the overall conclusion and logical coherence:
> > > > >
> > > > > - We added the independent censoring assumption to Theorem 4.4 (Truncation bias of KM-ADD), similarly to Theorem 4.2 (Finite-sample bias of KM-ADD).
> > > > > - We minimally revised the proof of Theorem 4.4 by introducing Lemmas B.5, B.6, and B.8.
> > > > >   - Lemma B.8 refines the proof of Theorem 4.4.
> > > > >   - Lemmas B.5 and B.6 are additional and further clarify and ensure coherence with the proof of Theorem 4.3.
> > > > >
> > > > > We believe the revised argument is now clearer and more rigorous.
> > > > >
> > > > > ### Revised Proof for the last sentence of Theorem 4.4
> > > > >
> > > > > We provide the revised proof for the last sentence of Theorem 4.4: $\mathbb{E}[ \Delta \tau \mid \nu < \infty, 0 \leq \Delta \tau \leq \Delta  T_\mathrm{max}^{\*}] - \mathbb{E}[\Delta \tau \mid \nu < \infty, 0 \leq \Delta \tau \leq \Delta T] \geq 0$.
> > > > > Without loss of generality, we will show that
> > > > > $\mathbb{E}[ \Delta \tau \mid 0 \leq \Delta \tau \leq \Delta  T_\mathrm{max}^{\*}] - \mathbb{E}[\Delta \tau \mid 0 \leq \Delta \tau \leq \Delta T] \geq 0$, where $\Delta T \leq \Delta T_{\mathrm{max}}^{{\*}} (\in \mathbb{R})$ holds almost surely. $\Delta \tau$ and $\Delta T (= C^{\mathrm{ADD}})$ are independent because of the independent censoring assumption.
> > > > > For notational simplicity, we rewrite this inequality as
> > > > >
> > > > > $\mathbb{E}\_{X,Y} [X | X \leq y_{\mathrm{max}}] - \mathbb{E}\_{X,Y} [X | X \leq Y] \geq 0$,
> > > > >
> > > > > where $X$ is a non-negative random variable, $Y$ is a non-negative random variable with $0 \leq Y \leq y_{\mathrm{max}}$ almost surely, $y_{\mathrm{max}} (\geq 0)$ is a real constant, and $X$ and $Y$ are independent.
> > > > >
> > > > > Write $F_X$ for the CDF of $X$, and let
> > > > >
> > > > > $g(x) := P(Y \geq x), \quad 0 \leq x \leq y_{\mathrm{max}}$.
> > > > >
> > > > > Because $Y \in\left[0, y_{\mathrm{max}}\right], g$ is non-increasing in $x$, while $g(x)=0$ for $x > y_{\mathrm{max}}$.
> > > > > Thus, we have
> > > > >
> > > > > $P(X \leq Y) = \mathbb{E}\left[1 (X \leq Y) \right] = \int_0^{\infty} P(Y \geq x) d F_X(x)=\int_0^{y_{\mathrm{max}}} g(x) d F_X(x)$,
> > > > >
> > > > > where $1$ denotes the indicator function, and we used the independence of $X$ and $Y$. Additionally, we have
> > > > >
> > > > > $\mathbb{E}\left[X  1 (X \leq Y)\right] = \int_0^{\infty} x P(Y \geq x) d F_X(x) =\int_0^{y_{\mathrm{max }}} x g(x) d F_X(x)$.
> > > > >
> > > > > Therefore,
> > > > >
> > > > > $\mathbb{E}[X \mid X \leq Y] = \frac{\mathbb{E}\left[X  1 (X \leq Y)\right]}{P(X \leq Y)} = \frac{\int_0^{y_{\mathrm{max}}} x g(x) d F_X(x)}{\int_0^{y_{\mathrm{max}}} g(x) d F_X(x)}.$ $\cdots$ (A)
> > > > >
> > > > > On the other hand, we can similarly derive
> > > > >
> > > > > $\mathbb{E}\left[X \mid X \leq y_{\mathrm{max}}\right]=\frac{\int_0^{y_{\mathrm{max}}} x d F_X(x)}{\int_0^{y_{\mathrm{max}}} d F_X(x)}.$ $\cdots$ (B)
> > > > >
> > > > > Next, let $\nu$ be the finite measure on $\left[0, y_{\mathrm{max}} \right]$ given by $d \nu(x)=d F_X(x)$, and define another measure $\mu$ by
> > > > >
> > > > > $d \mu(x)=g(x) d \nu(x), \quad 0 \leq x \leq y_{\mathrm{max}}$.
> > > > >
> > > > > Note that $g(x)$ is non-increasing in $x$, and $x$ is increasing in $x$.
> > > > > Thus, Eqs. (A) and (B) can be rewritten as
> > > > >
> > > > > $\mathbb{E}[X \mid X \leq Y]=\frac{\int x d \mu(x)}{\int d \mu(x)}, \quad \mathbb{E}\left[X \mid X \leq y_{\mathrm{max}}\right]=\frac{\int x d \nu(x)}{\int d \nu(x)}$.
> > > > >
> > > > > Consider the covariance under the normalized measure proportional to $\nu$:
> > > > >
> > > > > $\mathrm{Cov}\_\nu(x, g(x)) = \mathbb{E}\_{\nu} [X g(X)] - \mathbb{E}\_{\nu} [X]  \mathbb{E}\_{\nu} [g(X)] \leq 0$,
> > > > >
> > > > > because $x$ is increasing and $g(x)$ is decreasing (see Lemma B.6).
> > > > > Therefore,
> > > > >
> > > > > $(\frac{\int x g(x) d \nu(x)}{\int d \nu(x)}) \leq ( \frac{\int x d \nu(x)}{\int d \nu(x)} ) (\frac{\int g(x) d \nu(x)}{\int d \nu(x)})$.
> > > > >
> > > > > This is equivalent to
> > > > >
> > > > > $\frac{\int x g(x) d \nu(x)}{\int g(x) d \nu(x)} \leq \frac{\int x d \nu(x)}{\int d \nu(x)}$.
> > > > >
> > > > > In other words,
> > > > > $\mathbb{E}[X \mid X \leq Y] \leq \mathbb{E}\left[X \mid X \leq y_{\mathrm{max}}\right]$.
> > > > >
> > > > > This concludes the proof.

---

> ### Author Response · Authors · 2025-11-23
>
> We are grateful for the reviewer's comments, which have clarified our contributions and improved our proof.
> If the reviewer has further suggestions for revisiting the score, please let us know.
> We would be happy to continue and improve the manuscript.
> Please also refer to the Strengths we summarized in our General Response, which the reviewer may have overlooked.
>
> ~~**The revised manuscript will be submitted in a few days.**~~ **Uploaded.**
>
> ## References
>
> - [idXY1] Royston, Patrick, and Mahesh KB Parmar. "Restricted mean survival time: an alternative to the hazard ratio for the design and analysis of randomized trials with a time-to-event outcome." BMC medical research methodology 13.1 (2013): 152.
> - [idXY2] lifelines documentation: https://lifelines.readthedocs.io/en/latest/lifelines.utils.html#lifelines.utils.restricted_mean_survival_time

---

### Official Review · Reviewer_FXnt · 2025-11-01

**Soundness:** 2
**Presentation:** 3
**Contribution:** 2
**Rating:** 6
**Confidence:** 4

**Summary:**

This paper addresses a significant practical gap in the evaluation of quickest changepoint detection (QCD) models. While the Average Run Length (ARL) and Average Detection Delay (ADD) are standard theoretical metrics, their application to real-world datasets is hindered because data sequences are often of finite and irregular lengths. This "truncation" or "censoring" leads to conventional estimators being highly biased and unreliable.

The authors propose two new non-parametric estimators, KM-ARL and KM-ADD, which solve this problem by drawing a novel analogy between changepoint detection and survival analysis. The authors provide a theoretical analysis of the estimators, deriving bounds for the estimation bias and proving they are asymptotically unbiased unless extrapolation is required (i.e., estimating an ARL far beyond the longest observed sequence).

**Strengths:**

- Solves a Practical Problem: The paper tackles a clear and important real-world challenge. Anyone who has tried to apply QCD models outside of a perfect simulation environment will have faced the problem of evaluating them on finite, messy data. This work provides a principled and ready-to-use solution.

- Novel and Clever Approach: The analogy to survival analysis is insightful and allows the authors to leverage a well-established, powerful statistical tool (the KME) in a new domain.

- Theoretically Sound: The proposal is not just a heuristic. The authors provide a solid theoretical foundation by deriving bias bounds and proving asymptotic unbiasedness. Decomposing the bias into "finite-sample" and "truncation" components clearly explains why the estimators work.

- Non-Parametric: The KME-based approach makes no assumptions about the underlying data distributions or how detection times are distributed (e.g., exponentially). This makes the estimators broadly applicable.

**Weaknesses:**

- The bias analysis for the KM-ADD (Thm 4.2) relies on an "independent censoring" assumption. The authors justify this with an approximation, but it remains a simplification that may not perfectly hold in all scenarios.
- The proposed estimators require a dataset with ground-truth changepoint labels ($\nu_i$) to be calculated. This is a significant practical limitation, as many real-world QCD problems involve unlabeled data. The method is therefore excellent for evaluating models on a labeled test set but cannot be used to estimate the ARL/ADD of a model in a general, unlabeled operational setting (where simulations are traditionally used).
- The paper's theoretical analysis focuses almost entirely on proving the estimators are unbiased. A rigorous analysis of the variance of the KM-ARL and KM-ADD estimators is explicitly noted as being out of scope. While empirical results suggest the variance is lower than baseline methods, a full theoretical understanding of the estimator's variance is a missing piece for a complete evaluation.

**Questions:**

- Though in Section 3, the analogy between changepoint models and survival analysis is well-explained. But the motivation and intuition of the original KM estimator, and the idea of the choices of $d_j$ and $n_j$ in the changepoint model, are unclear.

---

> ### Author Response · Authors · 2025-11-23
>
> We deeply appreciate the reviewer for their careful reading and detailed comments, as well as for recognizing the strengths of our paper and supporting its acceptance.
> Below, we would like to address all concerns and questions the reviewer raised.
>
> For later discussions, please let us summarize the weaknesses and questions the reviewer raised:
>
> - W1:  The independent censoring assumption in Theorem 4.2 (finite-sample bias bound for KM-ADD) is justified with an approximation, but it remains a simplification that may not perfectly hold in all scenarios.
> - W2: The proposed estimators require a dataset with ground-truth changepoint labels.
> - W3: A rigorous analysis of the variance of the KM-ARL and KM-ADD estimators is explicitly noted as being out of scope. While empirical results suggest the variance is lower than baseline methods, a full theoretical understanding of the estimator's variance is a missing piece for a complete evaluation.
> - Q1: Though in Section 3, the analogy between changepoint models and survival analysis is well-explained. But the motivation and intuition of the original KM estimator, and the idea of the choices of $d_j$ and $n_j$ in the changepoint model, are unclear.

---

> ### Author Response · Authors · 2025-11-23
>
> ## W1
>
> We agree that "the independent censoring assumption in Theorem 4.2 is a simplification that may not *perfectly* hold in *all* scenarios."
> Indeed, we clarify the impact of its violation in Section 6: "for dependent censoring occurring just before event time (detection), estimation bias increases sharply", citing (Malmquist, 2025) (line 483 of the original submission), which has been quantified in our response to Reviewer jxVp (Q3), providing more insights for practitioners.
>
> To address the reviewer's concern, we would like to extend our potential solutions presented in Section 6:
> > (Lines 474-477 of the original submission) This assumption can potentially be relaxed by leveraging extensive research on dependent censoring in survival analysis (Hsu & Taylor, 2010; Lin et al., 2023; Crommen et al., 2025). Doing so would not only eliminate the independent censoring assumption but also extend our bias analysis to offline QCD, where censoring depends on detection.
>
> ### Background: Restricted Mean Survival Time (RMST)
>
> To mitigate the bias arising from dependent censoring and finite time interval, we must quantify the effect of the bias on the survival function or its integral within a finite time interval, known as the restricted mean survival time (RMST) in survival analysis.
> It is the counterpart of KM-ARL and KM-ADD in survival analysis and is defined as the expected survival time up to a specified horizon (denoted by $h$ here), equivalently the area under the survival curve from $0$ to $h$.
> It has been known that the RMST is systematically biased under dependent censoring [FXnt1] [FXnt2] [FXnt3].
>
> ### Solutions
>
> Many studies have developed mitigation techniques for bias arising from dependent censoring.
> The following works, among others, suggest promising directions for deriving KM-ARL/ADD under dependent (informative) censoring or alleviating the resulting bias:
>
> - [FXnt3]: Develops a consistent estimator of the survival function under dependent censoring, which may extend to estimating ARL and ADD in QCD.
> - (Hsu & Taylor, 2010): Proposes a robust weighted KME that uses baseline prognostic covariates to correct bias in the marginal survival function when censoring is dependent.
> - (Lin et al., 2023): Applies inverse probability of censoring weighting (IPCW); at each time $t$, each subject (patient) still under observation is weighted by the inverse of their estimated probability of remaining uncensored up to $t$.
> - (Crommen et al., 2025): Summarizes several bias-correction strategies:
>   - *Nonparametric marginal estimation under a known copula*: Given a specified copula for $(\tau, C)$, observable probabilities uniquely determine the marginals. Step-function estimators are constructed that reduce to the KME under independence and yield a consistent estimator under dependent censoring.
>   - *Likelihood estimation in copula models*: Corrects bias by modeling the joint distribution of $(\tau, C)$; parametric copulas enable joint likelihood maximization with identifiable parameters and consistent, asymptotically normal estimates.
>   - Correction of regression functionals: Embeds dependent censoring in copula-based Cox and related semi-parametric models to adjust hazard ratios, covariate effects, and causal effects.
>   - Machine-learning and partial-identification approaches: Addresses bias when flexible modeling or weaker assumptions are required, using both deep and non-deep learning methods.
>
> ### Conclusion of W1
>
> To avoid technical complications, we adopt the independent censoring assumption, a standard assumption in survival analysis, examine the conditions under which it holds in QCD, and specify them (lines 215-221 and 246-251 of the original submission).
> In this response, we also expand the discussion and references on strategies for relaxing this assumption and for correcting the resulting bias.
> These additions have been incorporated in Appendix F (Supplementary Discussion).
>
> A full analysis of the above studies and their integration into QCD would warrant a separate paper and is indeed an ongoing research direction for the authors.
> Nonetheless, we are happy to share our current preliminary insights here, as we believe they offer useful starting points for future work and provide essential background for developing the emerging link between survival analysis and QCD.

---

> > ### Author Response · Authors · 2025-11-23
> >
> > ## W2
> >
> > Thank you for your feedback. The reviewer's comment is valuable, as our work is interdisciplinary and requires careful clarification for a broad range of readers.
> >
> > In short, this problem setting is common in machine learning.
> > Although there are many studies of QCD on unlabeled datasets or pure simulations, there are also many studies on labeled datasets (please see our General Response for the references [1]-[17] below):
> >
> > - **There are a wide variety of labeled datasets** with multiple sequences and labeled changepoints (or annotations that can be seen as changepoints):
> >   - **Human sensing**: WISDM Actitracker [Kwapisz et al., 2011] includes both human- and machine-labeled changepoints. We use this real-world sensing data in our paper.
> >   - **Twitter sentiment-change detection**: Twitter Data Stream (US Airline Sentiment) [7], a collection of tweets annotated with the tags positive/negative/neutral, is used to detect sentiment changes in social media data stream [6]: e.g., a sudden increase in negative tweets after an event.
> >   - **Speaker change detection**: Many speech datasets, such as DIHARD dataset [11] and IITG-MV phase 3 dataset [12], have speaker or language annotations, which can be used for speaker change detection [10].
> >   - **Temporal action localization**: THUMOS14 [13] and ActivityNet [14] are standard benchmark datasets in temporal action localization [15]. They contain a number of videos with temporal annotations of human actions (e.g., Horseback riding, Diving, Long jump, etc.), where timestamps with action changes can be seen as changepoints.
> >   - **Earthquake detection**: STEAD [16] and several datasets in SeisBench offer datasets [17] of seismic signals with annotation of event times.
> > - Furthermore, we can **create labeled datasets from unlabeled real-world dataset** by injecting changepoints:
> >   - **Sound anomaly detection**: Normal and abnormal audio data recorded from industry machine are concatenated to synthesize changepoints [1]. The base dataset is the MIMI Dataset [2].
> >   - **Social event detection**: Based on the unlabeled MIT Cellphone Data [4], annotations are defined by linking changepoints to calender events (e.g., sponsor meeting, Presidents Day, spring break, etc.) [3].
> >   - **Social event detection**: Similarly to [3], annotations are provided for Enron Email Data [9] based on actual corporate events (e.g., Federal Energy Regulatory Commission's decisions, the CEO's public protests, the bankruptcy filing, etc.) [8].
> > - Of course, we can **create labeled datasets via simulation** (e.g., KDD Cup 1999 intrusion detection dataset [5]).
> > - Thus, real-world applications include:
> >   - human sensing and action recognition [Kwapisz et al., 2021] [15]
> >   - sentiment-change detection on social media data stream [6]
> >   - speaker/language change detection [10]
> >   - earthquake detection [17]
> >   - sound anomaly detection [1]
> >   - event detection from cellphone data or email data [3] [8]
> >   - intrusion detection [5], among others.
> >
> > In summary, a short answer to the reviewer's comment
> >
> > > The proposed estimators require a dataset with ground-truth changepoint labels ($\nu_i$) to be calculated. This is a significant practical limitation, as many real-world QCD problems involve unlabeled data.
> >
> > is as follows: It is correct that "many real-world QCD problems involve unlabeled data", but requiring a dataset with changepoint labels is not "a significant practical limitation", as shown above.
> >
> > We hope our response clarifies the practical relevance and broad applicability of our estimators. We have extended Appendix F (Supplementary Discussion) to incorporate the discussion above, which we believe further clarify our contribution for a broad range of readers. Thank you again for the feedback.

---

> > > ### Author Response · Authors · 2025-11-23
> > >
> > > ## W3
> > >
> > > We discuss variance estimation of KM-ARL/ADD in Appendix F (lines 1928–1938 of the original submission), as the reviewer notes (thank you for reading). Because a complete variance analysis requires additional assumptions to examine the transfer of KME to QCD—one of the main challenges in developing our bias theory—we provide only a brief discussion in Appendix F.
> > >
> > > Below, to address the reviewer's concern, we add further explanation from both theoretical and empirical perspectives.
> > > We focus on LB-ARL and KM-ARL; a similar discussion applies to LB-ADD and KM-ADD.
> > >
> > > ### 1. Theoretical Perspective
> > >
> > > We add more details of our brief discussion in Appendix F.
> > > The variance of LB-ARL (naive, standard definition of variance) is given by
> > >
> > > $\int_0^a (t - \bar{t})^2  dF(t)$, $\cdots$ (A)
> > >
> > > where $a$ denotes the cut-off time, $F$ the CDF of the event time, and $\bar{t}$ the mean event time.
> > > On the other hand, Akritas (2000) provides the asymptotic variance of the RMST—analogous to the variance of KM-ARL in our setting:
> > >
> > > $\int_0^a \frac{S(t)}{1 - H(t)} (t - \bar{t})^2  dF(t)$ $\cdots$ (B)
> > >
> > > (Eq. (4) in (Akritas, 2000)), where $S$ denotes the survival function and $H$ the CDF of $\min \{\mathrm{event\,time}, \mathrm{censoring} \}$.
> > > Compared with LB-ARL (A), the variance of KM-ARL (B) is computed as a weighted expectation of $(t - \bar{t})^2$, with its scale determined by the weight $\frac{S(t)}{1 - H(t)}$.
> > > Because $S(t) \in [0, 1]$, this weight can grow large when $1-H(t)$ is small unless $S(t)$ decreases sufficiently in the same region. Consequently, the variance integral in (B) can be dominated by the interval where $t$ is large and close to the least upper bound of $H(t)$. Therefore, the variance of KM-ARL can exceed that of LB-ARL when both event time and censoring have heavy tails and $a$ is large; otherwise, it may be smaller.
> > >
> > > Finally, the assumptions and background theory in Akritas (2000) are intricate and require careful examination to validate this expression in QCD—one of the main challenges in developing our bias theory. A full treatment would warrant a separate paper.
> > >
> > > We have expanded our variance discussion in the revised manuscript (Appendix F).
> > >
> > > ### 2. Empirical Perspective
> > >
> > > We describe the empirical, practical computation of the restricted variance of the detection time, $\hat{V} := \widehat{\mathrm{Var}} [ \min \{\tau, a\} ]$, in our experiments.
> > >
> > > - LB-ARL:
> > >   - The restricted variance of LB-ARL is defined as the naive empirical variance: $\hat{V} = \widehat{\mathrm{Var}} [ \min \{\tau, T \} ]$, which is consistent with the mean of LB-ARL ($\mathbb{E}[\min \{\tau, T \} ]$). Given a dataset with size $N$, it is computed as $\frac{1}{N_{\mathrm{LB}}} \sum_{i=1}^{N_{\mathrm{LB}}} (\tau_i - \bar{\tau})^2$. $\bar{\tau}$ is the empirical LB-ARL, i.e., the detection time averaged over the subset in which $\tau_i \leq$ sequence length. The subset size is denoted by $N_{\mathrm{LB}} (\leq N)$.
> > >
> > > - KM-ARL:
> > >   - There are several methods to numerically compute the variance of RMST (KM-ARL). As stated in Appendix D (Detailed Experimental Settings), we adopt `lifelines` (Davidson-Pilon, 2019), a standard Python library for survival analysis, to compute the KME and its variance.
> > >   - Specifically, we use `lifelines.utils.restricted_mean_survival_time` with `return_variance=True`, in which $\hat{V} = 2 \int_{0}^{a} t \hat{S}(t) d t - (\int_0^{a} \hat{S}(t) d t )^2$, where $\hat{S}(t)$ denotes the empirical survival function, computed from the given dataset with size $N$. The derivation is given in the appendix in [FXnt4], as is mentioned in [FXnt5].
> > >
> > > - Why do KM-ARL/ADD reduce variance compared to LB-ARL/ADD?
> > >   - We elaborate on the discussion of our experimental results on the real-world dataset (lines 438-446 of the original submission).
> > >   - In short, $\hat{V}$ of LB-ARL tends to larger than that of KM-ARL because the former can use only $N_{\mathrm{LB}} \leq N$, while the latter can exploit all $N$ sequences.
> > >     - LB-ARL/ADD become unstable and exhibit high variance, particularly at large QCD thresholds, where $N_{\mathrm{LB}}$ tends to be small because the detector often fails to raise an alarm within the sequence length (Figure 1(a)).
> > >     - In contrast, KM-ARL/ADD do not suffer from this issue because they are calculated from a constant number of sequences $N$, regardless of the threshold, which enhances their robustness against censoring.
> > >
> > > We have expanded our discussion on experimental results and settings in the revised manuscript (Section 5 and Appendix D).
> > >
> > > ### Final Remark
> > >
> > > Thank you for your feedback. We hope that our clarification and additional variance analysis address the reviewer’s concern.
> > > Given the broad range of strengths of our interdisciplinary work (summarized in the Strengths section of our General Response), we do not believe that the absence of a *complete* analysis of *both* bias and variance constitutes sufficient grounds for rejection.

---

> ### Author Response · Authors · 2025-11-23
>
> ## Q1
>
> Thank you very much for the actionable suggestion.
> To clarify the motivation and intuition of the original KME, and the roles of $d_j$ and $n_j$, we have extended Appendix A (QCD-Survival Analysis Correspondence) to include:
>
> - an illustrative explanation of the original KME, KM-ARL, and the computation of $d_j$ and $n_j$ (Figure 5 of the revised submission), and
> - a new background subsection (Appendix A.1: Background of KME in Survival Analysis of the revised submission) that clarifies the original KME and the rationale for choosing $d_j$ and $n_j$. This subsection contains:
>   - A.1.1 Introduction: Introduces relevant definitions and motivates the KME by introducing a naive non-parametric estimator of the survival function. Subsections A.1.2-A.1.5 are dedicated to the derivation of the KME formula to clarify its motivation.
>   - A.1.2 Decomposition via Conditional Survival Probabilities: Decomposes the survival function into the products of conditional probabilities.
>   - A.1.3 Risk Set $n_j$ and Number of Events $d_j$: Introduces $d_j$ and $n_j$ and explains their intuitions.
>   - A.1.4 Why $d_j/n_j$? Conditional Probability Viewpoint: Introduces the factor of $1 - d_j/n_j$ in the KME as the empirical estimate of the conditional survival probability at time $t_j$.
>   - A.1.5 KME as Product-limit Estimator: Finally defines the KME by putting everything together, and summarizes the roles of $d_j$ and $n_j$.
>
> ------------------
> Thank you again for your time and insightful feedback.
> If there are specific points we could clarify or improve to merit a higher assessment, please let us know.
> We would be happy to continue and improve our manuscript.
>
> ~~**The revised manuscript will be submitted in a few days.**~~ **Uploaded.**
>
> ## References
>
> - [FXnt1] Klein, J.P. & Moeschberger, M.L. (1984): “Asymptotic Bias of the Product Limit Estimator under Dependent Competing Risks.” Indian J. of Productivity, Reliability & Quality Control 9:1–7.
> - [FXnt2] Rivest, Louis-Paul, and Martin T. Wells. "A martingale approach to the copula-graphic estimator for the survival function under dependent censoring." Journal of Multivariate Analysis 79.1 (2001): 138-155.
> - [FXnt3] Ebrahimi, N. & Molefe, D. (2003): "Survival function estimation when lifetime and censoring time are dependent." J. Multivar. Anal. 87(1):101–132.
> - [FXnt4] Royston, Patrick, and Mahesh KB Parmar. "Restricted mean survival time: an alternative to the hazard ratio for the design and analysis of randomized trials with a time-to-event outcome." BMC medical research methodology 13.1 (2013): 152.
> - [FXnt5] lifelines documentation: https://lifelines.readthedocs.io/en/latest/lifelines.utils.html#lifelines.utils.restricted_mean_survival_time

---

### Official Review · Reviewer_DqTG · 2025-11-01

**Soundness:** 2
**Presentation:** 2
**Contribution:** 2
**Rating:** 2
**Confidence:** 3

**Summary:**

This paper proposes nonparametric estimators for the Average Run Length (ARL) and Average Detection Delay (ADD), termed KM-ARL and KM-ADD, respectively. By drawing an analogy between quickest change detection (QCD) and survival analysis, the authors apply the Kaplan–Meier estimator to model detection probabilities under sequence truncation, addressing the challenge of limited and irregular sequence lengths.

**Strengths:**

- The paper makes a conceptual link between QCD and survival analysis, and proposes a potential alternative to Monte Carlo–based ARL/ADD evaluations when sequences are truncated.
- The empirical studies include both synthetic and real data examples.

**Weaknesses:**

Overall, I find the contribution limited in two aspects. Firstly and most importantly, the problem itself is of limited interest to the community. While the QCD problem is very useful and widely studied in practice, it is unclear how useful the proposed ARL and ADD evaluation methods are in realistic scenarios. I can imagine that in simulation environments, the proposed estimators could be used to approximate the ARL without simulating until the actual stopping time (which can be very time-consuming, as ARL values are often on the order of 10⁴ or 10⁵). However, even in such settings, the exponential approximation often provides sufficiently accurate results, and in some cases, closed-form expressions for the ARL already exist—allowing the detection threshold to be determined simply by solving an analytical equation. In real-world scenarios, where finite sequence lengths are common due to data collection constraints, the proposed KM-ARL and KM-ADD require multiple sequences to obtain the Kaplan–Meier estimator. This substantially limits the applicability of the proposed method, as in many practical situations there is typically only a single data stream for which detection is performed, and multiple independent streams with the same distribution may not be available. Secondly, the methodology itself is based on the well-known Kaplan–Meier estimator (KME), a standard technique in survival analysis, and therefore offers limited methodological novelty.

Moreover, the presentation could be improved for clarity; definitions and assumptions are sometimes imprecise or ambiguous (e.g., in Eq. (1)), see questions below.

**Questions:**

I have a concern about the definition of ADD in Eq.(1). Typically, in the QCD literature, there are two common definitions of detection delay. One is the conditional average detection delay, which involves taking an additional supremum over all possible change-points $ \nu = 0, 1, \ldots $ in Eq.(1); the other is the Bayesian setting, which further takes an expectation over the change-point $\nu$ in Eq.(1). Both metrics do not depend on a specific change-point $\nu$, whereas the definition in Eq.(1) does if I understand correctly. Since the detection delay can indeed vary across different change-point locations—as has been well documented in the QCD literature—the current definition of ADD in Eq.(1) is, in a sense, not precise.
Moreover, as mentioned in Section 5 ("two types of change-point distributions: geometric and uniform"), I assume the authors are considering a Bayesian setting. If so, it would be more appropriate to use the commonly adopted Bayesian ADD definition. To my understanding, in Sec 5, the authors assign different (random) change-points for each sequence but estimate the ADD under the assumption that the detection delay distributions for different change-points $\nu$ are identical. This assumption does not generally hold for many detection algorithms.

For the real-data results, the authors conclude that the proposed methods “...reduce both bias and variance compared to baseline estimators...” While it is clear from Figure 1 that KM-ARL and KM-ADD can indeed reduce variance, it is not evident why they also reduce bias. In this real-world dataset with limited sequence lengths (and substantial censoring), there is no known ground-truth value for ARL or ADD to support such a conclusion.

Minor: It should be clarified in Eq.(1) whether the expectation is also taken over the randomness in the observed sequence length $T$, or if $T =\infty $ is assumed (i.e., no censoring).

---

> ### Author Response · Authors · 2025-11-23
>
> We appreciate the reviewer for their time and insightful comments.
> Below, we would like to address all concerns and resolve potential confusions one by one.
>
> ## Weaknesses
>
> For later discussion, please let us summarize the weaknesses raised by the reviewer:
>
> - W1: The topic of the paper is of limited interest.
> - W2: It is unclear how useful the proposed ARL and ADD evaluation methods are in realistic scenarios.
> - W3: The methodology is based on the standard, well-known KME and offers limited novelty.
>
> For the presentation issue that the reviewer mentions at the end of the Weaknesses section, please refer to our responses to the Questions section below.
>
> ### W1
>
> Thank you for the feedback; however, we respectfully disagree that the paper is of limited interest in the community.
> Analyzing how we evaluate performance metrics is of **general interest** in machine learning:
>
> - Top-tier conferences themselves emphasize this by requiring careful reporting of metrics in their Author Instructions (https://iclr.cc/Conferences/2026/AuthorGuide) and Reviewer Instructions (https://iclr.cc/Conferences/2026/ReviewerGuide), underscoring that **reproducible and interpretable evaluation is a first-class concern**, not an afterthought. Our work advances more robust, interpretable evaluation of QCD models, enabling empirical, intuitive model selection, as stated in the Abstract and Experiment sections.
> - Many ICLR papers discuss individual components related to our work: (1) statistical properties of performance metrics, (2) statistical bias analysis, and (3) changepoint detection. Examples of such ICLR papers include papers such as [jxVp6]-[jxVp8] and [jxVp14]-[jxVp24] (please see our response to Reviewer jxVp (W1) for these references).
> - Furthermore, we show the practical relevance and broad applicability of our estimators in our General Response (2-1. Relevance of requiring datasets with multiple sequences with changepoint labels).
>
> In response to the reviewer's feedback, we have added a contribution statement in Appendix F (Supplementary Discussion), clarifying the relevance and applications of our method.

---

> > ### Author Response · Authors · 2025-11-23
> >
> > ### W2
> >
> > Thank you for the insightful suggestion and clarifications.
> > We respond to the comments individually to avoid potential confusion.
> >
> > > While the QCD problem is very useful and widely studied in practice, it is unclear how useful the proposed ARL and ADD evaluation methods are in realistic scenarios.
> >
> > We would like to elaborate on our practical contributions in the following.
> > The reviewer's comment is valuable, as our work is interdisciplinary and requires careful clarification for a broad range of readers.
> >
> > > I can imagine that in simulation environments, the proposed estimators could be used to approximate the ARL without simulating until the actual stopping time (which can be very time-consuming, as ARL values are often on the order of 10⁴ or 10⁵).
> >
> > Yes, the proposed estimators can be used in simulation environments without simulating until the actual stopping time, which is often computationally expensive. As a result, they allow substantial savings in computational budget. Thank you for highlighting this advantage. We have added this contribution to Appendix F (Supplementary Discussion).
> >
> > > However, even in such settings, the exponential approximation often provides sufficiently accurate results, and in some cases, closed-form expressions for the ARL already exist—allowing the detection threshold to be determined simply by solving an analytical equation.
> >
> > We agree that:
> >
> > - the exponential approximation provides accurate performances in **some scenarios**, and
> > - closed-form expressions for the ARL are known for **some limited cases** (e.g., (Fu et al., 2002; Areepong & Peerajit, 2022; Sunthornwat et al., 2023), cited in Appendix E).
> >
> > However, it has been also known that:
> >
> > - the exponential approximation does not work well if the distribution deviates from exponential [DqTG18] [DqTG19], and
> > - closed-form expressions are not always available—it depends on distributions and detectors.
> >
> > This is why we emphasize that our estimators are applicable to **arbitrary** underlying distributions (=non-parametric) and **arbitrary** QCD models (line 1861 etc. of the original submission), a key contributions that clearly distinguishes our work from prior approaches.
> >
> > (continued)

---

> ### Author Response · Authors · 2025-11-23
>
> > In real-world scenarios, where finite sequence lengths are common due to data collection constraints, the proposed KM-ARL and KM-ADD **require multiple sequences** to obtain the Kaplan–Meier estimator. This substantially **limits the applicability** of the proposed method, **as in many practical situations there is typically only a single data stream** for which detection is performed, and multiple independent streams with the same distribution may not be available.
>
> It is correct that our proposed estimators require a test dataset of multiple sequences with changepoint labels, as is stated at the first sentence in Introduction; however, we do not believe this can be the ground for rejection because:
>
> - **There are a wide variety of labeled datasets** with multiple sequences and labeled changepoints (or annotations that can be seen as changepoints):
>   - **Human sensing**: WISDM Actitracker [Kwapisz et al., 2011] includes both human- and machine-labeled changepoints. We use this real-world sensing data in our paper.
>   - **Twitter sentiment-change detection**: Twitter Data Stream (US Airline Sentiment) [DqTG7], a collection of tweets annotated with the tags positive/negative/neutral, is used to detect sentiment changes in social media data stream [DqTG6]: e.g., a sudden increase in negative tweets after an event.
>   - **Speaker change detection**: Many speech datasets, such as DIHARD dataset [DqTG11] and IITG-MV phase 3 dataset [DqTG12], have speaker or language annotations, which can be used for speaker change detection [DqTG10].
>   - **Temporal action localization**: THUMOS14 DqTG[DqTG13] and ActivityNet [DqTG14] are standard benchmark datasets in temporal action localization [DqTG15]. They contain a number of videos with temporal annotations of human actions (e.g., Horseback riding, Diving, Long jump, etc.), where timestamps with action changes can be seen as changepoints.
>   - **Earthquake detection**: STEAD [DqTG16] and several datasets in SeisBench offer datasets [DqTG17] of seismic signals with annotation of event times.
> - Furthermore, we can **create labeled datasets from unlabeled real-world dataset** by injecting changepoints:
>   - **Sound anomaly detection**: Normal and abnormal audio data recorded from industry machine are concatenated to synthesize changepoints [DqTG1]. The base dataset is the MIMI Dataset [DqTG2].
>   - **Social event detection**: Based on the unlabeled MIT Cellphone Data [DqTG4], annotations are defined by linking changepoints to calender events (e.g., sponsor meeting, Presidents Day, spring break, etc.) [DqTG3].
>   - **Social event detection**: Similarly to [DqTG3], annotations are provided for Enron Email Data [DqTG9] based on actual corporate events (e.g., Federal Energy Regulatory Commission's decisions, the CEO's public protests, the bankruptcy filing, etc.) [DqTG8].
> - Of course, we can **create labeled datasets via simulation** (e.g., KDD Cup 1999 intrusion detection dataset [DqTG5]).
> - Thus, real-world applications include:
>   - human sensing and action recognition [Kwapisz et al., 2021] [DqTG15]
>   - sentiment-change detection on social media data stream [DqTG6]
>   - speaker/language change detection [DqTG10]
>   - earthquake detection [DqTG17]
>   - sound anomaly detection [DqTG1]
>   - event detection from cellphone data or email data [DqTG3] [DqTG8]
>   - intrusion detection [DqTG5], among others.
>
> We hope our response clarifies the practical relevance and broad applicability of our estimators, as acknowledged by Reviewers FXnt and jxVp (please see the Strengths summarized in our General Response).
>
> In response to the reviewer's comment, we have expanded Appendix E (Supplementary Related Work) and Appendix F (Supplementary Discussion) to address these points, which we believe further clarify our contributions for a broad range of readers. We appreciate the reviewer's the feedback.

---

> > ### Author Response · Authors · 2025-11-23
> >
> > ### W3
> >
> > We respectfully disagree with the claim that "the novelty is limited because the methodology is based on a standard, well-known technique (KME)".
> > A primary novelty of our work lies in the **conceptual link between QCD and survival analysis**, as noted by Reviewers DqTG and FXnt.
> > This connection allows us to **leverage a well-established statistical tool (KME) in a new domain (QCD)**, as highlighted by Reviewer FXnt.
> > As a result, we enable the evaluation of ARL and ADD on real-world datasets with limited and irregular-length sequences—an important practical challenge faced by anyone attempting to apply QCD models outside a perfect simulation environment (noted by Reviewers FXnt and jxVp).
> > In addition, we demonstrate practical utility through experiments on synthetic and real-world data (Reviewer DqTG) and provide ready-to-use Python implementations (Reviewer FXnt).
> > Furthermore, we also derive estimation bias bounds and prove that the estimators are asymptotically unbiased under certain conditions, which was appreciated by Reviewers FXnt, dkXY, and jxVp.
> >
> > In line with the ICLR 2026 Reviewer Guidelines (https://iclr.cc/Conferences/2026/ReviewerGuide), our goal is to present novel findings that "better address a known application or problem, draw attention to a new application or problem, or introduce and/or explain a new theoretical finding".
> > Novelty in research often arises from combining established techniques, which may seem obvious in retrospect.
> > Although from a different venue (NeurIPS 2025 Reviewer Guidelines: https://neurips.cc/Conferences/2025/ReviewerGuidelines), our work "offers a novel combination of existing techniques," and we note that "originality does not necessarily require introducing an entirely new method".
> >
> > Again, we greatly value the reviewer’s perspective and feedback, and we hope our response clarifies our contributions and assists in reassessing the overall contribution, novelty, practical relevance, and effectiveness of our work.

---

> > > ### Author Response · Authors · 2025-11-23
> > >
> > > ## Questions
> > >
> > > We would like to summarize the questions the reviewer raised:
> > >
> > > - Q1: The definition in Eq. (1) (ADD $M_\infty$) depends on a specific changepoint, which is uncommon and conflicts with the standard definitions of the sup ADD (ADD obtained by taking sup over the changepoint) and the Bayesian ADD (ADD obtained by taking the expectation over the changepoint).
> > > - Q2: In the Experiment section (Section 5), the reviewer assumes a Bayesian setting is considered. If so, it would be more appropriate to use the commonly adopted Bayesian ADD definition.
> > > - Q3: In the Experiment section (Section 5), the detection delay distributions for different changepoints are assumed to be identical, which does not generally hold.
> > > - Q4: In the real-data experiment, the statement "These metrics (=proposed estimators) reduce both bias and variance compared to baseline estimators..." (lines 442-443 in Section 5.2) is imprecise because, although variance is obviously reduced (Figure 1), the bias cannot be assessed on real-world data, as no ground truth for ARL or ADD exists in such data.
> > > - Q5: (Minor) In Eq.(1), expectation is also taken over $T$, or $T=\infty$ is assumed?
> > >
> > > ### Q1
> > >
> > > We apologize for any confusion, which we believe arises from a notational convention.
> > > Our definition of ADD **is** standard and follows the QCD literature (e.g., Tartakovskii (2019)).
> > >
> > > - We refer to the Bayesian ADD as ADD, which takes an expectation over the changepoint.
> > > - This definition aligns with Eq. (2.27) in Tartakovskii (2019), the standard definition of the Bayesian average delay to detection.
> > > - It is also is consistent with our proof and with the definitions given in the main text ("the changepoint $\nu$ is a random variable": lines 107-108 of the original submission).
> > >
> > > We have clarified the definition in Eq. (1), in which we explicitly state the expectation is also taken over $\nu$.
> > > Additionally, while preparing this response, we recognized that the original sentences in lines 1898–1899 of the submission were incorrect. We have corrected them in the revised version.
> > > Thank you very much for the helpful suggestion.
> > >
> > > We hope our response to Q1 addresses Q2 and Q3; nevertheless, we would like to answer them below for completeness.

---

> ### Author Response · Authors · 2025-11-23
>
> ### Q2
>
> We adopt the Bayesian ADD, as the reviewer assumes, and thus our setting is valid.
>
> ### Q3
>
> It is a consequence of the independent censoring assumption for KM-ADD (Theorem 4.2, lines 249-251 of the original submission).
> We extend the discussion of this validity condition and potential alleviation in our response to Reviewer FXnt (W1).
>
> Or, do you refer to the assumption mentioned in the files `calc_gtADD_Gauss.py` and `calc_gtADD_Poisson.py` in our code?
> It is made to make the numerical experiment tractable.
> Technically, this assumption drastically reduce the computational time for no-censoring detection delays (illustrated as black "True" curves in Figures 3, 4, 6, 12, 15, and 16 of the original submission), as they otherwise require effectively infinite lengths of sequences.
> Without this assumption, it would take $>$ several weeks to compute no-censoring detection delays in several settings (e.g., geometric changepoint distribution with small success probabilities ($p$)).
> We have now added our comments above in Appendix D (Detailed Experimental Settings), not only in the code.
>
> Please feel free to request further clarification or restate the question if additional information is needed, or kindly let us know if the concern is resolved.
> We would be happy for any additional suggestions that could help with improving our manuscript.
>
> ### Q4
>
> The reviewer's comment is correct: "although variance is obviously reduced (Figure 1), the bias cannot be assessed on real-world data, as no ground truth for ARL or ADD exists in such data."
> We have therefore removed the reference to "bias" from these lines.
> This revision does not affect the overall conclusion or the logic coherence of this section.
> Thank you for the suggestion.
>
> ### Q5
>
> $T=\infty$ is assumed. We have clarified the definition in Eq. (1) in the revised version.
>
> -----------------
> Once again, we would like to express our sincere appreciation for your thoughtful feedback, which we believe has further strengthened our manuscript.
> We would be grateful for any additional suggestions you may have that could help with reconsideration of the current score.
> We remain open to further discussion and warmly welcome any additional comments.
> Please also refer to the Strengths we summarized in our General Response, which the reviewer may have overlooked.
>
> ~~**The revised manuscript will be submitted in a few days.**~~ **Uploaded.**

---

> > ### Author Response · Authors · 2025-11-23
> >
> > ## References
> >
> > - [DqTG1] Gopalan, Aditya, Braghadeesh Lakshminarayanan, and Venkatesh Saligrama. "Bandit quickest changepoint detection." Advances in Neural Information Processing Systems 34 (2021): 29064-29073. URL: https://arxiv.org/abs/2107.10492.
> > - [DqTG2] Harsh Purohit, Ryo Tanabe, Kenji Ichige, Takashi Endo, Yuki Nikaido, Kaori Suefusa, and Yohei Kawaguchi. MIMII Dataset: Sound Dataset for Malfunctioning Industrial Machine Investigation and Inspection. Version public 1.0. Zenodo, Sept. 2019. DOI:
> > 10.5281/zenodo.3384388. URL: https://doi.org/10.5281/zenodo.3384388.
> > - [DqTG3] Kei, Yik Lun, et al. "Change Point Detection in Dynamic Graphs with Decoder-Only Latent Space Model." Transactions on Machine Learning Research, Jan. 2025. URL: https://openreview.net/forum?id=DVeFqV56Iz.
> >  URL: https://openreview.net/forum?id=DVeFqV56Iz.
> > - [DqTG4] MIT Cellphone Data: http://realitycommons.media.mit.edu/realitymining.html.
> > - [DqTG5] KDD-99 dataset: https://kdd.ics.uci.edu/databases/kddcup99/kddcup99.html.
> > - [DqTG6] Bouchikhi, Ikram, et al. “Kernel Based Online Change Point Detection.” 2019 27th European Signal Processing Conference (EUSIPCO), 2019, pp. 1–5. IEEE Xplore, https://doi.org/10.23919/EUSIPCO.2019.8902582.
> > - [DqTG7] Twitter Data Stream (US Airline Sentiment) from Kaggle: https://www.kaggle.com/datasets/crowdflower/twitter-airline-sentiment.
> > - [DqTG8] Wang, Xiuheng, et al. "Change point detection with neural online density-ratio estimator." ICASSP 2023-2023 IEEE International Conference on Acoustics, Speech and Signal Processing (ICASSP). IEEE, 2023. URL: https://ieeexplore.ieee.org/document/10095321.
> > - [DqTG9] Enron Email Data: https://www.cs.cmu.edu/~enron/.
> > - [DqTG10] Mishra, Jagabandhu, and SR Mahadeva Prasanna. "Spoken language change detection inspired by speaker change detection." Circuits, Systems, and Signal Processing 43.10 (2024): 6373-6398.
> > - [DqTG11] Ryant, N., Church, K., Cieri, C., Cristia, A., Du, J., Ganapathy, S., and Liberman, M. (2018). “First dihard challenge evaluation plan,” 2018, Tech. Rep.
> > - [DqTG12] Haris, B. C., Pradhan, G., Misra, A., Prasanna, S., Das, R. K., and Sinha, R. (2012). “Multivariability speaker recognition database in indian scenario,” International Journal of Speech Technology 15(4), 441–453.
> > - [DqTG13] THUMOS14: https://www.crcv.ucf.edu/THUMOS14/.
> > - [DqTG14] ActivityNet: http://activity-net.org/download.html.
> > - [DqTG15] Wang, Binglu, et al. "Temporal action localization in the deep learning era: A survey." IEEE Transactions on Pattern Analysis and Machine Intelligence 46.4 (2023): 2171-2190. URL: https://ieeexplore.ieee.org/document/10310147.
> > - [DqTG16] STanford EArthquake Dataset (STEAD):A Global Data Set of Seismic Signals for AI: https://github.com/smousavi05/STEAD.
> > - [DqTG17] SeisBench: https://github.com/seisbench/seisbench.
> > - [DqTG18] Borror, Connie M., J. Bert Keats, and Douglas C. Montgomery. "Robustness of the time between events CUSUM." International Journal of Production Research 41.15 (2003): 3435-3444.
> > - [DqTG19] Pehlivan, Canan, and Murat Caner Testik. "Impact of model misspecification on the exponential EWMA charts: a robustness study when the time‐between‐events are not exponential." Quality and Reliability Engineering International 26.2 (2010): 177-190.

---

### Author Response · Authors · 2025-11-23
**General Response by Authors**

First of all, we heartily appreciate all reviewers for their feedback and suggestions, all of which will be incorporated into our revision.

## 1. Strengths

We thank the reviewers for recognizing our paper's strengths. We summarize them below for later discussions:

- Insightful **conceptual link** between QCD and survival analysis (DqTG, FXnt).
- **Leverages a well-established powerful statistical tool (the KME)**  in a new domain (FXnt).
- Empirical evaluation including both **synthetic** and **real-world** data (DqTG).
- Tackles a **clear, important real-world challenge** that anyone who tried to apply QCD models outside a perfect simulation environment will have faced (FXnt, jxVp).
- Provides a principled, **ready-to-use solution** (FXnt).
- Not just heuristics, but **theoretically sound and robust,** including finite-sample and truncation bias bounds and asymptotic unbiasedness (FXnt, dkXY, jxVp).
- Requires no assumptions about the underlying data distribution or how detection times are distributed (e.g., exponentially), i.e., **non-parametric**, leading to a **broad applicability** of the proposed estimator (FXnt).
- The proposed estimators substantially reduce the notoriously time-consuming computation of ARL in simulation (DqTG).
- Excellent Soundness and Contribution (jxVp).

We are pleased to present our interdisciplinary idea that bridges QCD and survival analysis.

---

> ### Author Response · Authors · 2025-11-23
>
> ## 2. Selected Concerns from Multiple Reviewers
>
> ### 2-1. Relevance of requiring datasets with multiple sequences with changepoint labels
>
> We thank Reviewers DqTG and FXnt for raising the concerns about the availability of multiple sequences with labeled changepoints.
> Their comments are valuable, as our work is interdisciplinary and requires careful clarification for a variety of readers.
>
> In short, this problem setting is common in machine learning.
> Although there are many studies of QCD on unlabeled datasets or pure simulations, there are also many studies on labeled datasets:
>
> - **There are a wide variety of labeled datasets** with multiple sequences and labeled changepoints (or annotations that can be seen as changepoints):
>   - **Human sensing**: WISDM Actitracker [Kwapisz et al., 2011] includes both human- and machine-labeled changepoints. We use this real-world sensing data in our experiments.
>   - **Twitter sentiment-change detection**: Twitter Data Stream (US Airline Sentiment) [7], a collection of tweets annotated with the tags of positive/negative/neutral, is used to detect sentiment changes in social media data stream [6]: e.g., a sudden increase in negative tweets after an event.
>   - **Speaker change detection**: Many speech datasets, such as DIHARD dataset [11] and IITG-MV phase 3 dataset [12], have speaker or language annotations, which can be used for speaker change detection [10].
>   - **Temporal action localization**: THUMOS14 [13] and ActivityNet [14] are standard benchmark datasets in temporal action localization [15]. They contain a number of videos with temporal annotations of human actions (e.g., Horseback riding, Diving, Long jump, etc.), where timestamps with action changes can be seen as changepoints.
>   - **Earthquake detection**: STEAD [16] and several datasets in SeisBench [17] offer datasets of seismic signals with annotation of event times.
> - Furthermore, we can **create labeled datasets from unlabeled real-world dataset** by injecting changepoints:
>   - **Sound anomaly detection**: Normal and abnormal audio data recorded from industry machine are concatenated to synthesize changepoints [1]. The base dataset is the MIMI Dataset [2].
>   - **Social event detection**: Based on the unlabeled MIT Cellphone Data [4], annotations are defined by linking changepoints to calender events (e.g., sponsor meeting, Presidents Day, spring break, etc.) [3].
>   - **Social event detection**: Similarly to [3], annotations are provided for Enron Email Data [9] based on actual corporate events (e.g., Federal Energy Regulatory Commission's decisions, the CEO's public protests, the bankruptcy filing, etc.) [8].
> - Of course, we can **create labeled datasets via simulation** (e.g., KDD Cup 1999 intrusion detection dataset [5]).
> - Thus, real-world applications include:
>   - human sensing and action recognition [Kwapisz et al., 2021] [15]
>   - sentiment-change detection on social media data stream [6]
>   - speaker/language change detection [10]
>   - earthquake detection [17]
>   - sound anomaly detection [1]
>   - event detection from cellphone data or email data [3] [8]
>   - intrusion detection [5], among others.
>
> We hope our response clarifies the practical relevance and broad applicability of our estimators. Although they cannot be used for unlabeled datasets (lines 29–30 of the original submission; Reviewer DqTG; Reviewer FXnt), they apply to many real-world tasks listed above, among others. We have extended Appendix F (Supplementary Discussion) to incorporate the points discussed above, which we believe further clarify our contribution for a broad range of readers.

---

> ### Author Response · Authors · 2025-11-23
>
> ### 2-2. Relevance for the ICLR Community
>
> Reviewers DqTG and jxVp expressed concerns about the level of interest this work will attract from the ICLR community.
>
> - We would like to note that analyzing how we evaluate performance metrics is of general interest in machine learning. Top-tier conferences themselves emphasize this by requiring careful reporting of metrics in their Author Instructions (https://iclr.cc/Conferences/2026/AuthorGuide) and Reviewer Instructions (https://iclr.cc/Conferences/2026/ReviewerGuide), underscoring that reproducible and interpretable evaluation is a first-class concern, not an afterthought. Our work advances more robust, interpretable evaluation of QCD models, enabling empirical, intuitive model selection, as stated in the Abstract and Experiment sections.
> - In addition, many ICLR papers discuss individual components closely related to our work (please see our response to Reviewer jxVp (W1)).
> - Furthermore, we have shown the practical relevance and broad applicability of our estimators above (2-1. Relevance of requiring datasets with multiple sequences with changepoint labels).
>
> Given these considerations, submitting our work to ICLR was a natural choice.
> In response to reviewers' feedback, we added a contribution statement for the community in Appendix F (Supplementary Discussion) to clarify the relevance of our work.
> Please see each thread for individual responses.
>
> ### 2-3. More about Variance Estimation
>
> Reviewers FXnt and dkXY mention variance analysis beyond our bias analysis.
>
> - Please note that in Appendix F (lines 1928-1938 of the original submission), we discuss the variance of the restricted mean survival time—analogous to the variances of ARL and ADD in our setting—based on (Akritas, 2000).
> - In response to reviewers' suggestions, we extended Appendix F to further elaborate on the variance analysis from both theoretical and empirical perspectives.
>
> For details, please refer to our responses for each reviewer in the individual threads.
>
> ------------------------
> We are happy to continue and improve the manuscript as suggested by the reviewers and the OpenReview community.
> **The revised manuscript will be submitted in a few days.**
>
> Sincerely,
>
> Authors

---

> > ### Author Response · Authors · 2025-11-23
> >
> > ## 3. References
> >
> > - [1] Gopalan, Aditya, Braghadeesh Lakshminarayanan, and Venkatesh Saligrama. "Bandit quickest changepoint detection." Advances in Neural Information Processing Systems 34 (2021): 29064-29073. URL: https://arxiv.org/abs/2107.10492.
> > - [2] Harsh Purohit, Ryo Tanabe, Kenji Ichige, Takashi Endo, Yuki Nikaido, Kaori Suefusa, and Yohei Kawaguchi. MIMII Dataset: Sound Dataset for Malfunctioning Industrial Machine Investigation and Inspection. Version public 1.0. Zenodo, Sept. 2019. DOI:
> > 10.5281/zenodo.3384388. URL: https://doi.org/10.5281/zenodo.3384388.
> > - [3] Kei, Yik Lun, et al. "Change Point Detection in Dynamic Graphs with Decoder-Only Latent Space Model." Transactions on Machine Learning Research, Jan. 2025. URL: https://openreview.net/forum?id=DVeFqV56Iz.
> >  URL: https://openreview.net/forum?id=DVeFqV56Iz.
> > - [4] MIT Cellphone Data: http://realitycommons.media.mit.edu/realitymining.html.
> > - [5] KDD-99 dataset: https://kdd.ics.uci.edu/databases/kddcup99/kddcup99.html.
> > - [6] Bouchikhi, Ikram, et al. “Kernel Based Online Change Point Detection.” 2019 27th European Signal Processing Conference (EUSIPCO), 2019, pp. 1–5. IEEE Xplore, https://doi.org/10.23919/EUSIPCO.2019.8902582.
> > - [7] Twitter Data Stream (US Airline Sentiment) from Kaggle: https://www.kaggle.com/datasets/crowdflower/twitter-airline-sentiment.
> > - [8] Wang, Xiuheng, et al. "Change point detection with neural online density-ratio estimator." ICASSP 2023-2023 IEEE International Conference on Acoustics, Speech and Signal Processing (ICASSP). IEEE, 2023. URL: https://ieeexplore.ieee.org/document/10095321.
> > - [9] Enron Email Data: https://www.cs.cmu.edu/~enron/.
> > - [10] Mishra, Jagabandhu, and SR Mahadeva Prasanna. "Spoken language change detection inspired by speaker change detection." Circuits, Systems, and Signal Processing 43.10 (2024): 6373-6398.
> > - [11] Ryant, N., Church, K., Cieri, C., Cristia, A., Du, J., Ganapathy, S., and Liberman, M. (2018). “First dihard challenge evaluation plan,” 2018, Tech. Rep.
> > - [12] Haris, B. C., Pradhan, G., Misra, A., Prasanna, S., Das, R. K., and Sinha, R. (2012). “Multivariability speaker recognition database in indian scenario,” International Journal of Speech Technology 15(4), 441–453.
> > - [13] THUMOS14: https://www.crcv.ucf.edu/THUMOS14/.
> > - [14] ActivityNet: http://activity-net.org/download.html.
> > - [15] Wang, Binglu, et al. "Temporal action localization in the deep learning era: A survey." IEEE Transactions on Pattern Analysis and Machine Intelligence 46.4 (2023): 2171-2190. URL: https://ieeexplore.ieee.org/document/10310147.
> > - [16] STanford EArthquake Dataset (STEAD):A Global Data Set of Seismic Signals for AI: https://github.com/smousavi05/STEAD.
> > - [17] SeisBench: https://github.com/seisbench/seisbench.

---

> > > ### Author Response · Authors · 2025-11-24
> > > **Manuscript Updated**
> > >
> > > Dear Reviewers, PCs, ACs, and SACs,
> > >
> > > We have uploaded our revised manuscript, incorporating all reviewer suggestions and feedback.
> > > We remain committed to further improving the manuscript and are happy to support the reviewers as they revisit their assessments.
> > >
> > > Sincerely,
> > >
> > > Authors

---

### Meta-Review · Area_Chair_CKRx · 2025-12-09

**Summary:**

The paper proposes non-parametric estimators for the average run length and average detection delay in changepoint detection.  They show the estimation bias bounds and prove that they are asymptotically unbiased.

**Reviewer Concerns:**

The contribution appears limited. The problem itself holds limited community interest. While the QCD problem is both practically useful and widely studied, the proposed ARL and ADD evaluation methods offer unclear utility in realistic scenarios. Their potential application seems confined to simulation environments, where they could approximate the ARL without running simulations to the actual stopping time. However, even in such settings, the established exponential approximation often yields sufficient accuracy, and in some cases, closed-form solutions for the ARL already exist—enabling detection thresholds to be set by simply solving an analytical equation.

The concern about the definition of ADD in Eq.(1).

**Reviewer Scores:**

Reviewer DqTG will not change the score.

 In Sec 5, the authors assign different (random) change-points for each sequence but estimate the ADD under the assumption that the detection delay distributions for different change-points are identical. This assumption does not generally hold for many detection algorithms. This question is not well adressed.

---

### Decision · Program_Chairs · 2026-01-26

Reject